# Neuroblastoma arises in early fetal development and its evolutionary duration predicts outcome

Verena Körber[1], Sabine A. Stainczyk [2,3], Roma Kurilov[4], Kai-Oliver Henrich[2,3], Barbara Hero[5], Benedikt Brors[4], Frank Westermann [2,3] ✉ & Thomas Höfer [1] ✉

Neuroblastoma, the most frequent solid tumor in infants, shows very diverse outcomes from spontaneous regression to fatal disease. When these different tumors originate and how they evolve are not known. Here we quantify the somatic evolution of neuroblastoma by deep whole-genome sequencing, molecular clock analysis and population-genetic modeling in a comprehensive cohort covering all subtypes. We find that tumors across the entire clinical spectrum begin to develop via aberrant mitoses as early as the first trimester of pregnancy. Neuroblastomas with favorable prognosis expand clonally after short evolution, whereas aggressive neuroblastomas show prolonged evolution during which they acquire telomere maintenance mechanisms. The initial aneuploidization events condition subsequent evolution, with aggressive neuroblastoma exhibiting early genomic instability. We find in the discovery cohort ($n = 100$), and validate in an independent cohort ($n = 86$), that the duration of evolution is an accurate predictor of outcome. Thus, insight into neuroblastoma evolution may prospectively guide treatment decisions.

Cancers result from the accumulation of oncogenic mutations[1]. Insights into tumor evolution—and, particularly, the temporal order of driver mutations—are beginning to support diagnosis and treatment[2]. Initially, age–incidence curves were used to estimate the number of rate-limiting mutations in carcinogenesis[3,4]. Recently, the order of driver mutations has been inferred from genome sequencing data[5–9], and mathematical approaches based on population genetics have been used to reconstruct the clonal evolution of cancers from the statistics of somatic variants[2,10]. Deep whole-genome sequencing (WGS) data are suited for such integrative analyses, because the allele frequencies of neutral somatic variants that hitchhike with driver mutations provide rich information on how drivers shape tumor growth[7,11–13].

Tumors of early childhood provide a paradigm for cancer evolution in the context of development. A key question is how driver mutations subvert the normal development of the tissue of origin[14,15]. The most common solid tumor in infants, neuroblastoma, arises in the sympathetic nervous system. A striking feature of neuroblastoma is the wide spectrum of clinical outcomes, ranging from low-risk cases requiring light or no treatment to high-risk disease that remains fatal for ~50% of patients[16]. Characteristic mutations, including *MYCN* amplification, gain of telomere maintenance mechanisms (TMMs) and gains (17q) or losses (1p, 11q) of chromosomal segments, have been associated with high-risk disease[17]. Nevertheless, the prospective stratification of patients into observation and different treatment groups remains a formidable challenge. In this Article, we study how neuroblastomas originate in development and evolve genetically and ask whether this understanding can provide insights into disease severity and outcomes.

[1]Division of Theoretical Systems Biology, German Cancer Research Center, Heidelberg, Germany. [2]Hopp Children's Cancer Center, Heidelberg, Germany. [3]Division of Neuroblastoma Genomics, German Cancer Research Center, Heidelberg, Germany. [4]Division of Applied Bioinformatics, German Cancer Research Center, Heidelberg, Germany. [5]Department of Pediatric Oncology and Hematology, University Children's Hospital of Cologne, Medical Faculty, Cologne, Germany. ✉e-mail: f.westermann@kitz-heidelberg.de; t.hoefer@dkfz-heidelberg.de

## Results

### Mutation patterns in a comprehensive neuroblastoma cohort

We assembled a discovery cohort of deep (~80×) WGS data of 100 neuroblastomas (Supplementary Table 1), covering all clinical stages of the disease according to the International Neuroblastoma Staging System (Fig. 1a). Sixty-seven samples were derived from initial diagnoses (including seven from metastases) and 33 from relapsed tumors. Median age at primary diagnosis was 0.7 years for stages 1, 2 and 4S, 3.5 years for stage 3 and 3.9 years for stage 4 (Extended Data Fig. 1a).

Median tumor purity was high (88%), which allowed for reliable estimation of tumor ploidy (Fig. 1b) as confirmed by direct measurement of DNA index (Extended Data Fig. 1b). Tumors had baseline ploidies between two and four, with 55, 33 and 12 samples being near-diploid, near-triploid and near-tetraploid, respectively. Each ploidy class contained tumors of all stages (Extended Data Fig. 1c). Relative to baseline ploidy, we detected in 96% of the tumors characteristic segmental (1q and 17q) or whole-chromosome (2, 7 and 17) gains, as well as segmental (1p, 11q) or whole-chromosome (11) losses (Extended Data Fig. 1d), all of which are probable drivers of neuroblastoma[18–23]. Based on tumor cell content and ploidy, the vast majority of these gains and losses were clonal and hence were present in the most recent common ancestor cell (MRCA) of the tumor (Fig. 1c). These data point to aneuploidy as an early feature of neuroblastoma.

The majority of tumors (69%) combined chromosomal gains or losses with candidate driver mutations[23] at smaller genomic scale (Fig. 1d and Supplementary Tables 2–5), including focal gene amplifications (for example, *MYCN*, *CDK4*, *ALK*), structural rearrangements (for example, in *TERT* and *ATRX*), large deletions (for example, *CDKN2A*, *ATRX*), small insertions/deletions (indels; for example, in *ATRX* and *NF1*) and somatic single-nucleotide variants (SSNVs which, among other genes, occurred in *ALK* and those coding for components of MAPK pathways: *HRAS*, *KRAS*, *NRAS* and *BRAF*). Most commonly, these additional oncogenic drivers support telomere maintenance by alternative lengthening of telomeres (ALT), *MYCN* amplification or rearrangement in the *TERT* locus, which, collectively, are a molecular predictor of poor outcome[24–26].

In contrast to driver mutations, neutral SSNVs and small indels are continuously accumulated and contain information on tumor evolution. Analysis of mutational signatures assigned the majority of SSNVs to clock-like signatures (SBS1, SBS5, SBS40). Overall the next most abundant signature was SBS18, associated with reactive oxygen species[27] and frequently found in neuroblastoma[28], followed by SBS3, associated with failure of homologous recombination in dividing cells (Fig. 1e). These signatures overall occurred at similar frequencies clonally and in subclones (Extended Data Fig. 1e).

### Neutral SSNVs distinguish sequential driver events

To understand when and how neuroblastomas evolve, we used neutral SSNVs as a molecular clock to time key events: (1) the emergence of the MRCA, from which the clonal sweep that defines the resected tumor emerges, and (2) the acquisition of clonal chromosomal gains (which may have occurred before or coincident with the MRCA). For each tumor, we computed the frequency distribution of somatic variants on all genomic segments of a given copy number. An example tumor, classified as near-triploid (Fig. 2a), had dominant copy numbers 2 and 3 and a smaller portion of segments at copy number 4—a typical configuration in neuroblastoma. On each copy number, we found a large clonal peak (Fig. 2b) comprising mutations occurring on a single chromosomal copy in each tumor cell. In addition, on copy numbers >2, we detected clonal mutations present on the two (and in some cases more) copies of a multiplied chromosome. Collectively, both single- and multiple-copy clonal SSNVs characterize the MRCA (Fig. 2c). Indeed, the density of clonal SSNVs was similar for the different copy numbers (Extended Data Fig. 2a), thus timing the MRCA by means of a molecular clock (Fig. 2d). To evaluate a potentially biasing effect of partial tumor sampling, which

may erroneously cause a subset of subclonal mutations to be classified as clonal, we analyzed pairs of primary and relapse samples and found that the vast majority (85 ± 5%) of clonal SSNVs were present in both samples, indicating that sampling did not introduce a strong bias. Nevertheless, to estimate MRCA density conservatively we performed all subsequent computations with the measured densities corrected by a factor of 0.85. In the example tumor, the mean SSNV density of the MRCA was one SSNV per 5 million base pairs (bp).

Next, we asked whether clonal chromosomal gains occurred coincident with, or ancestral to, the MRCA. To this end, we quantified SSNVs that were identical and clonal on two copies on trisomic and tetrasomic segments (termed amplified clonal SSNVs[6]; Fig. 2b). These mutations were acquired before a gain on the respective allele (Fig. 2c, dark green), and hence the number of these mutations correlates positively with the time at which the chromosomal gain occurred[6,29]. To make this timing measure independent of segment length, we use SSNV density. We compared these densities to those at the MRCA based on a negative binomial distribution of SSNVs across the genome (Methods). In the example tumor, nearly all gains had a mean density of one amplified clonal SSNV per 100 Mbp, which is significantly smaller (adjusted *P* < 0.01) than that of the MRCA (Fig. 2e). Hence, the molecular clock places these gains ancestral to the MRCA in an early common ancestor (ECA) of the tumor. The only exceptions were gains of Chr. 9 and 20q, which had a mutation density consistent with that of the MRCA and hence occurred later than the ECA. On those segments gained ancestral to the MRCA, the densities of amplified clonal SSNVs were statistically indistinguishable (Fig. 2e). Hence, near-triploidization occurred early and as a temporally confined event during the development of this tumor. Thereafter, further genetic evolution occurred before the clonal sweep from the MRCA commenced (Fig. 2f).

### Two evolutionary classes of neuroblastoma

We timed the MRCAs for all tumor samples, and the ECAs associated with chromosomal gains. First, we validated these timing approaches by comparison of mutation densities in samples taken following primary diagnosis with those of relapsed neuroblastomas. As expected, timing of the ECA, defining the putative early origin of tumorigenesis, was conserved in all sample types (Extended Data Fig. 2b). By contrast, the timing of the MRCA, defining the origin of a tumor sample, was significantly later in relapsed tumors, consistent with a bottleneck imposed by incomplete tumor resection or cytotoxic therapy (Supplementary Table 1) and, eventually, regrowth from a small number of surviving cells (Extended Data Fig. 2b,c). Interestingly, the MRCAs of metastases resected after initial diagnosis had SSNV densities indistinguishable from those of the primary tumors (Extended Data Fig. 2b,c), suggesting that metastases had originated around the time when the tumor started to grow from its MRCA.

To infer the early evolution of tumors up to the MRCA, we focused on samples taken at initial diagnosis (primary tumors and metastases, *n* = 67 in the discovery cohort). Remarkably, mutation densities of the MRCA showed a bimodal distribution (Fig. 3a). This finding suggests that there are two classes of neuroblastoma, one where growth of the resected tumor commenced early (early-MRCA neuroblastoma) and another where it occurred later (late-MRCA neuroblastoma); for mutational signatures, see Extended Data Fig. 2d,e. Of note, although we estimated the mutation density of the MRCA conservatively (Methods), the dichotomy between early and late MRCA emerged robustly.

These data raise the question of whether late-MRCA tumors began to develop later or developed early and evolved for a longer period of time. To address this question in terms of the molecular clock, we analyzed SSNV densities on chromosomal/segmental gains in both tumor classes that defined an ECA. All 20 early-MRCA tumors had such gains (with 70% showing near-triploidization, 35% harboring segmental gains and 25% displaying both features; Fig. 3b). The respective mutation densities timing chromosomal gains and MRCA were statistically

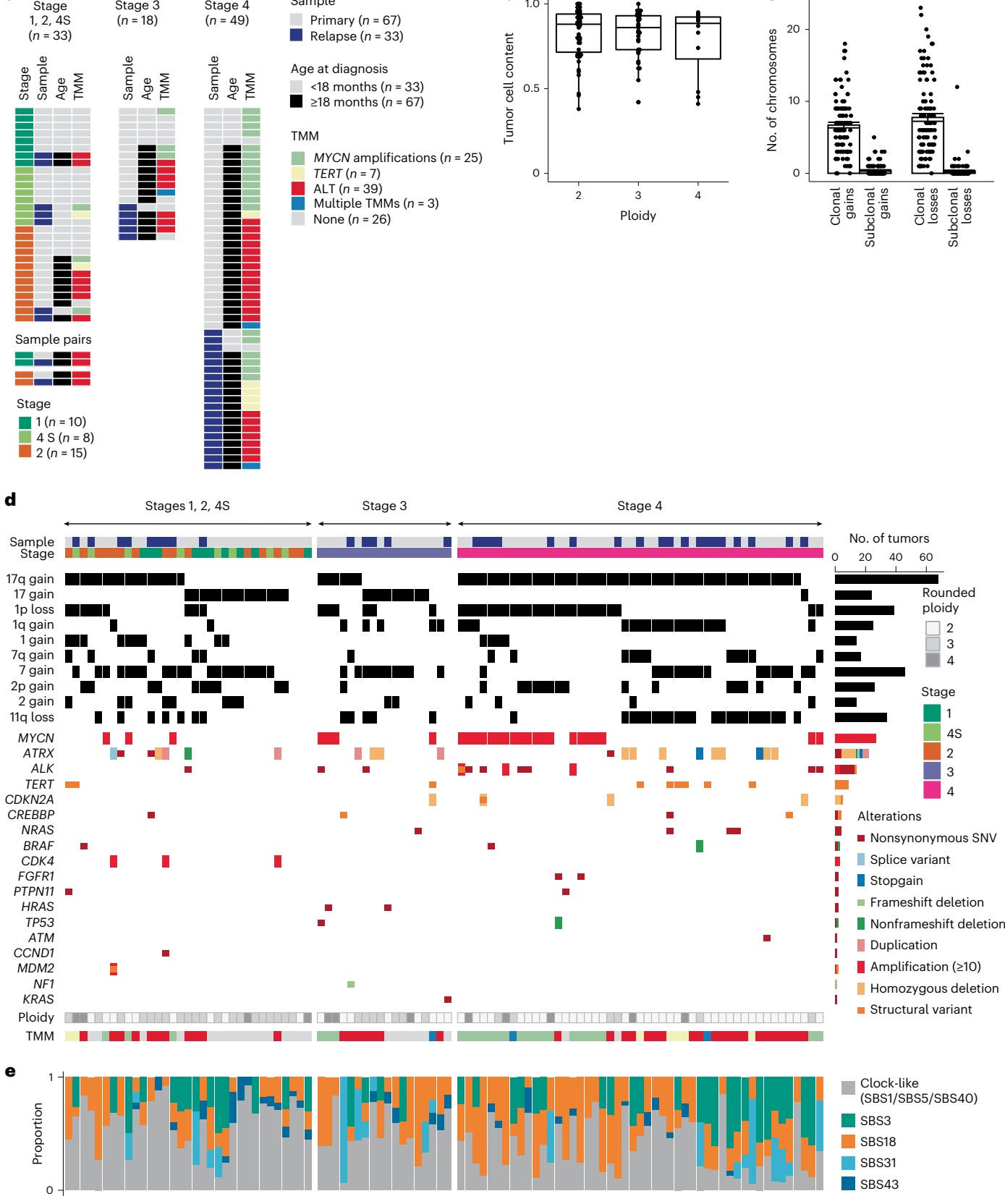

**Fig. 1 | Molecular subtypes and mutation spectrum of the discovery cohort.**
**a**, Clinical parameters and molecular characteristics. **b**, Tumor baseline ploidies
and tumor cell content. Boxes show median, 25 and 75% percentiles and whiskers
extend to the smallest and largest value within 1.5× interquartile range.
Shown are $n = 55$ near-diploid, $n = 33$ near-triploid and $n = 12$ near-tetraploid
tumors. **c**, Number of chromosomes harboring gains and losses ≥10⁶ bp. Shown

are mean and s.e. of the mean for $n = 100$ tumors. **d**, Copy number variants
and small-scale mutations (SSNVs, small insertions/deletions, amplifications,
homozygous deletions and structural rearrangements) in candidate driver genes.
**e**, Exposures of mutational signatures (COSMIC v.3.1) per sample. Signatures
SBS1, SBS5 and SBS40 were grouped into a single, clock-like mutations signature.

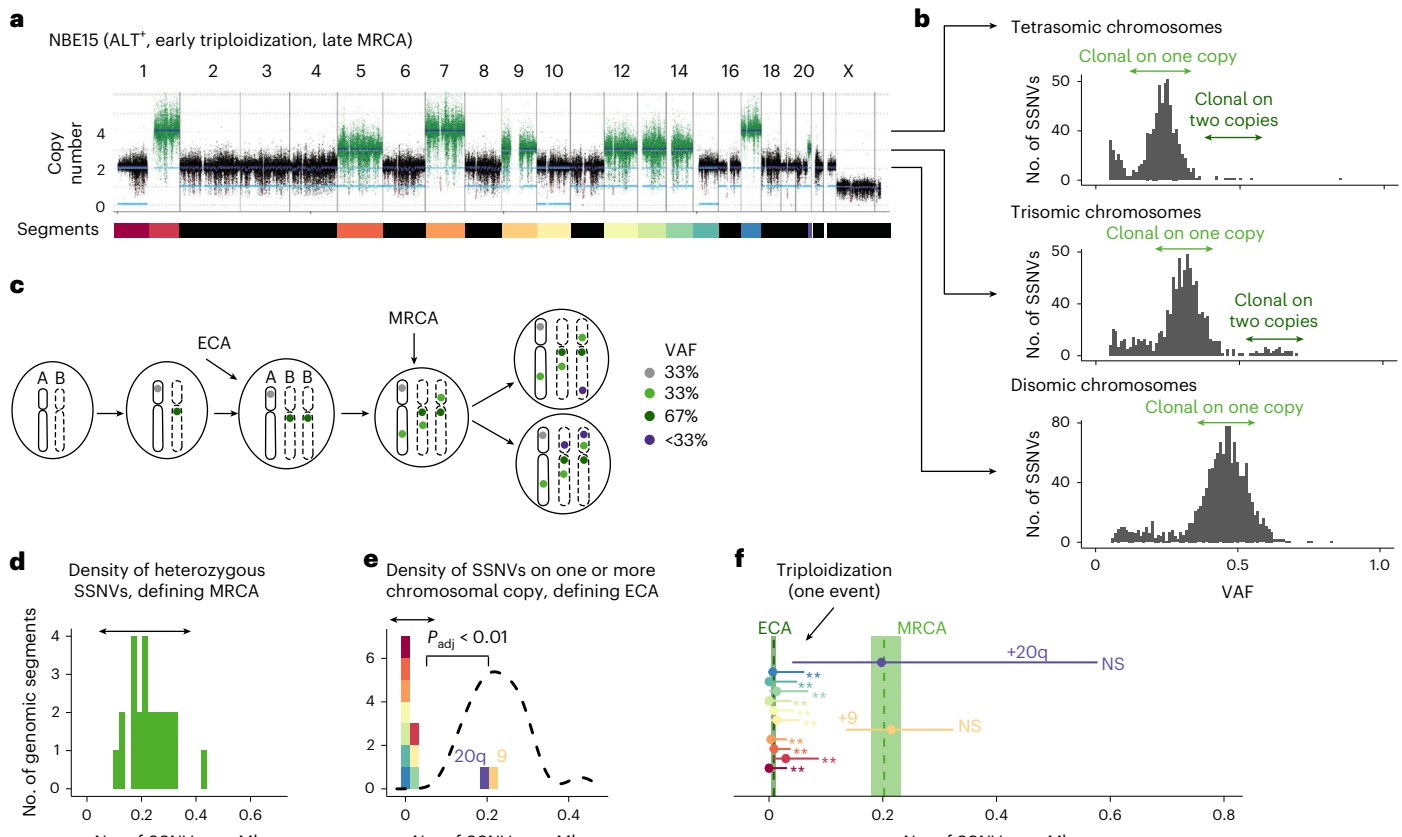

**Fig. 2 | Timing of chromosomal gains using SSNVs. a**, Copy number profile of a near-triploid tumor with ALT. Each segment of equal copy number is denoted by a color in the bar at the bottom; these colors are used below to mark the segments in **e,f**. **b**, VAF distribution of SSNVs stratified by copy number for the tumor shown in **a**. **c**, Schematic introducing the nomenclature based on VAF of SSNVs. **d,e**, Densities of non-amplified (**d**) and amplified (**e**) clonal mutations on genomic segments of length ≥10[7] bp for the tumor shown in **a**. **e**, Dashed line indicates kernel-density estimate of the distribution of non-amplified clonal mutations shown in **d**. **d,e**, Chromosomal segments of equal copy number

were combined into single genomic segments, and Holm-corrected, one-sided *P* values were computed based on negative binomial distribution (exact *P* values provided in Source Data). **f**, Mutation densities of ECA and MRCA, with 95% confidence bounds estimated by bootstrapping for the tumor shown in **a**. Horizontal lines represent mutation densities at gained segments, as in **e**, with 95% confidence bounds computed from $\chi^2$ distributions. Segments on which densities of amplified clonal mutations were significantly different from that at MRCA are marked by ** (adjusted *P* < 0.01); NS, not significant; test and *P* values as in **e**. **e,f**, Color coding for segments is in **a**.

indistinguishable in the vast majority (90%) of cases (Extended Data Fig. 3a shows an example), suggesting that aneuploidy was not followed by acquisition of further clonal drivers in these tumors. In the remaining 10% of tumors, separate ECAs and MRCAs were timed but these were very close in regard to mutation density (0.04 SSNVs per Mb between ECA and MRCA; Extended Data Fig. 3b;), implying that they were temporally close events. Indeed, 95% of early-MRCA tumors lacked additional small-scale drivers while a single tumor harbored a *MYCN* amplification and a mutation in *ALK*. Hence early-MRCA tumors appear to be driven predominantly by aneuploidization.

By contrast, in the majority (55%) of the 47 late-MRCA tumors, we distinguished two well-separated events, ECA and MRCA (Fig. 3c, left), with average distance 0.24 ± 0.1 SSNVs per Mb. In about one-third of these cases, a single early near-triplodization event defined an ECA (Fig. 2); whereas, in the remaining two-thirds, smaller-scale chromosomal gains defined the ECA (Extended Data Fig. 3c). Hence, the majority of late-MRCA tumors showed a signature of early gains followed by further genetic evolution to the MRCA. In the remaining 45% of late-MRCA cases, we could not reliably time a separate ECA (for example, if very short fragments were gained or if gains were found at copy number >4) (Fig. 3c (right) and Extended Data Fig. 3d). Hence, for these tumors we cannot time an early genetic event which, however, leaves open the possibility that small-scale mutations, chromosomal

losses or high-level amplifications (four or more copies), neither of which can be timed reliably, preceded the MRCA of the tumor. Consistent with this hypothesis, the mutation densities of the MRCA in tumors without timeable ECA were nearly as high as in those with two distinguishable events (Extended Data Fig. 3e). Moreover, near-tetraploid tumors without timeable ECA showed evidence of sequential events. Here, we found a 2:0 allelic configuration in 11 of 12 near-tetraploid tumors with 1p loss, suggesting that 1p deletion preceded genome doubling. Overall, the ECA of late-MRCA tumors had mutation density indistinguishable from the MRCAs of early-MRCA tumors (Fig. 3d), suggesting that the majority of late-MRCA tumors acquired aneuploidy as early as the early-MRCA tumors.

Finally, we analyzed the timing of gains in genome-wide events: near-triploidization of the genome or genome doubling. In all such tumors, the individual gains involved showed statistically indistinguishable timing, which is consistent with near-triploidization and genome doubling occurring as single catastrophic events (Extended Data Fig. 3f).

In sum, we find that the MRCAs in primary neuroblastoma fell into two evolutionary groups: in early-MRCA tumors, near-triploidization of the genome and/or gains of chromosomal arms or whole chromosomes coincided with the MRCA. Similarly, in more than one-half of late-MRCA tumors, an early ECA was defined by whole-chromosomal

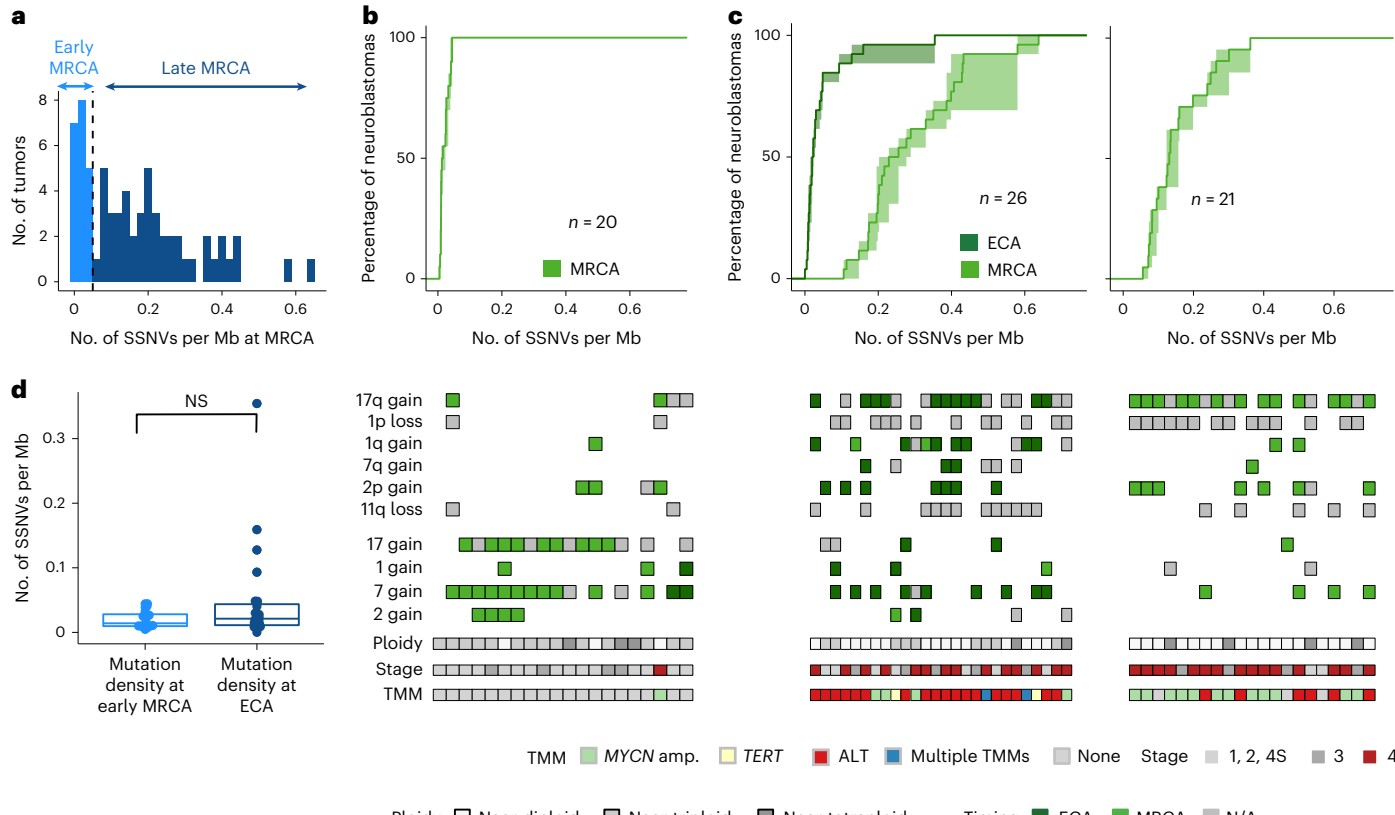

**Fig. 3 | Early- versus late-MRCA neuroblastoma. a**, Bimodal distribution of mutation densities of the MRCA for the discovery cohort; a threshold of 0.05 SSNVs per Mb separates the two modes (dashed line). **b,c**, Top, cumulative SSNV densities of the ECA (dark green) and MRCA (light green) of primary tumors classified as either early-MRCA neuroblastoma (**b**, $n = 20$ tumors) or late-MRCA neuroblastoma (**c**, $n = 26$ tumors with ECA and $n = 21$ tumors without ECA). **c**, Left and right panels correspond to tumors with and without timeable ECA, respectively. **b,c**, Solid lines represent maximum-likelihood estimates, while shaded areas represent 95% confidence intervals obtained by nonparametric

bootstrapping of chromosomal segments. Bottom, clonal chromosomal/segmental gains and losses implied in oncogenesis, ploidy, stage and presence of an acquired TMM. Gains were timed whenever possible as occurring in either the ECA (dark green) or MRCA (light green); gains that could not be timed are shown in gray. **d**, Mutation densities at the MRCA in 20 primary tumors classified as early MRCA, and at the ECA in 26 cases classified as late MRCA. Significance was tested using a two-sided Wilcoxon rank sum test/Mann–Whitney $U$-test ($\alpha = 0.05$). Boxes show median, 25 and 75% percentiles and whiskers extend to the smallest and largest value within 1.5× interquartile range. N/A, not applicable.

or arm-level gains. Hence, these late-MRCA neuroblastomas originated at a time similar to early-MRCA examples but then showed prolonged genetic evolution.

### Long evolution of neuroblastoma predicts unfavorable outcome

The early-MRCA class contained cases of stages 1, 2 and 4 S, typically having a prognosis, but only one case of stage 4 (Fig. 3a). Hence, we asked whether MRCA timing predicts outcome. Remarkably, early MRCA timing clearly identified cases with long event-free survival (Fig. 4a) and long overall survival (Fig. 4b). To further test this idea, we assembled an independent cohort of primary tumors and metastases, which was enriched for tumors with WGS at the customary coverage of 30× ($n = 86$; Supplementary Tables 6–10)[22,25]. We evaluated the timing of ECA and MRCA in each sample as described for the discovery cohort. The mutation density of the MRCA showed the same bimodal pattern as in the discovery cohort (Fig. 4c). Moreover, the three main patterns of ECA and MRCA occurrence were identical to the discovery cohort: coincidence of chromosomal/segmental gains and MRCA in the early-MRCA class (Fig. 4d; compare with Fig. 3b); in the late-MRCA class, chromosomal/segmental gains defining an early ECA that substantially preceded either a late MRCA (Fig. 4e, left; compare with Fig. 3c, left) or a late MRCA without timeable ECA (Fig. 4e, right; compare with Fig. 3c, right). Thus, the validation cohort

corroborates the scenarios of ECA and MRCA timing found in the discovery cohort.

In the validation cohort, MRCA timing was an accurate predictor of both event-free and overall survival (Fig. 5a,b). Merging the two cohorts ($n = 152$ primary tumors and metastases, excluding one case lacking survival data), we confirmed this result (Fig. 5c,d). To compare MRCA timing with other predictors of survival, we considered clinical variables used worldwide (stage, age), gain of a TMM (which improves on the clinically used criterion, *MYCN* amplification[26]) and a more recently proposed molecular predictor, the mutation status of the RAS/p53 pathway[25]. Early MRCA timing emerged as the most informative predictor of event-free survival (Fig. 5); overall survival was best explained by both MRCA timing and mutations in the RAS/p53 pathway (Extended Data Fig. 4). In sum, survival analyses suggest that extended evolution up to the founding cell of the primary tumor predicts unfavorable outcome.

### Genomic instability and telomere maintenance in late-MRCA tumors

Given that both early- and late-MRCA neuroblastomas begin to develop in the same time window (compare with Fig. 3d) but show markedly different durations of evolution and clinical outcome, we asked whether late-MRCA tumors have evolved further than early-MRCA examples simply by chance, or whether there are molecular factors that

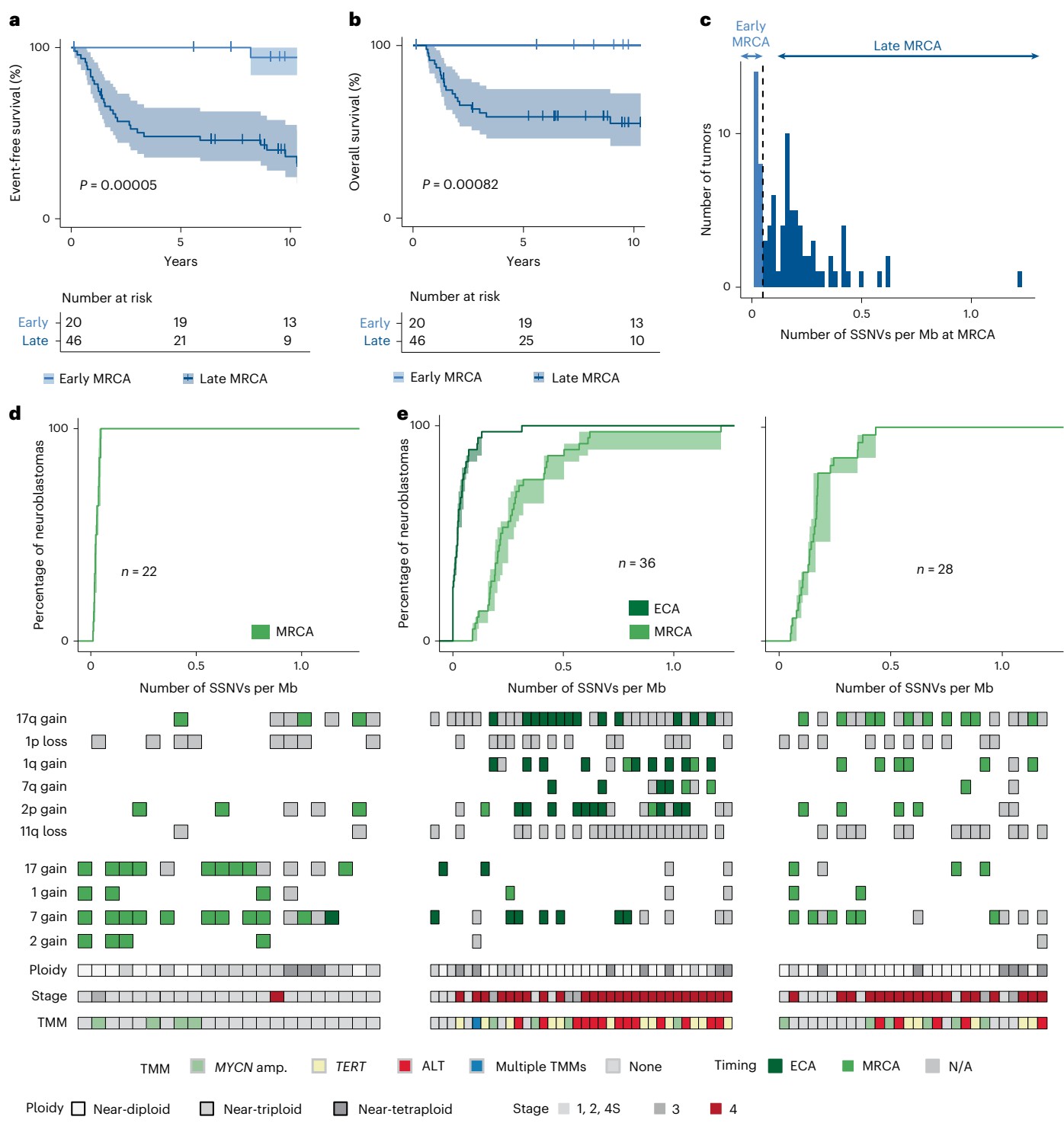

**Fig. 4 | Timing of the MRCA predicts outcome. a,b,** Event-free (**a**) and overall survival (**b**) stratified by mutation density of the MRCA of primary tumors and metastases from the discovery cohort (*n* = 66; a single tumor lacking survival information was excluded from the analysis). Survival is shown for up to 10 years. *P* values were computed using the log-rank test; error band represents 95% confidence interval. **c,** Distribution of mutation densities of the MRCA for the validation cohort. The threshold (0.05 SSNVs per Mb, dashed line) is identical to that of the discovery cohort (compare with Fig. 3a). **d,e,** Mutation densities of ECA and MRCA for the validation cohort. Top, cumulative SSNV densities of the ECA (dark green) and MRCA (light green) of primary tumors classified as either early-MRCA neuroblastoma (**d**, *n* = 22) or late-MRCA neuroblastoma (**e**, *n* = 36 with ECA, *n* = 28 without ECA). **e,** Left and right panels correspond to tumors with and without timeable ECA, respectively. **d,e,** Solid lines represent maximum-likelihood estimates and shaded areas represent 95% confidence intervals obtained by nonparametric bootstrapping of chromosomal segments. Bottom, timing of pervasive chromosomal gains; segments compatible with both ECA and MRCA were classified as early, with subclonal gains excluded.

predispose to ongoing evolution in the late-MRCA class. Characteristic oncogenic events (chromosomal or segmental gains or losses, TMMs) and standard prognostic features (stage and age at diagnosis) showed

strong separation between the two classes of neuroblastoma (Fig. 6a). Early-MRCA tumors had predominantly whole-chromosome aneuploidy (for example, chromosomes 17 and 7), whereas arm-level aneuploidy

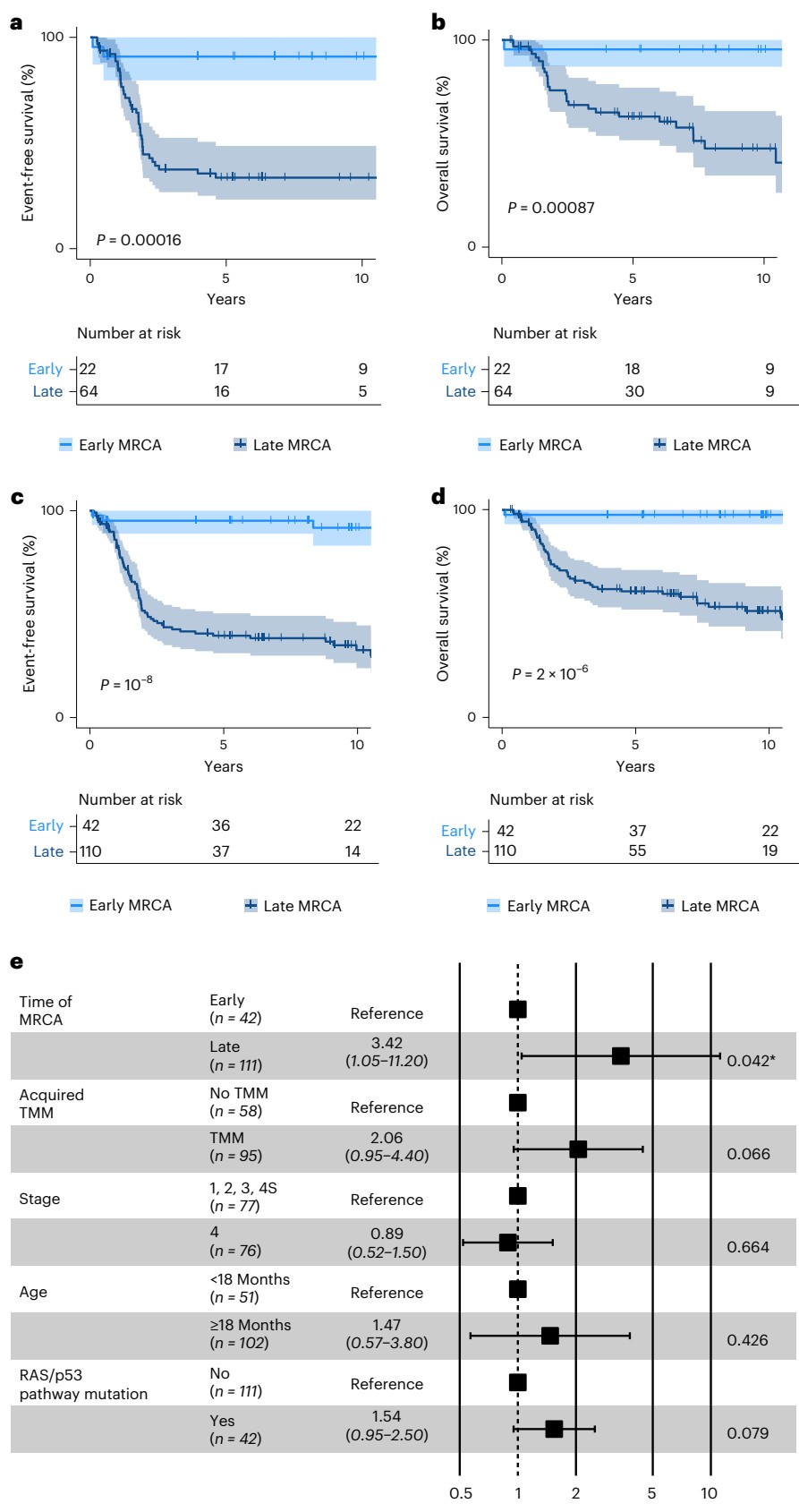

**Fig. 5 | Survival analysis. a–d**, Event-free (**a**,**c**) and overall survival (**b**,**d**) stratified by mutation density at the MRCA of primary tumors and metastases from the validation cohort (**a**,**b**, *n* = 86) and from both cohorts (**c**,**d**, *n* = 152). Survival is shown for up to 10 years; *P* values were computed using the log-rank test, and error band represents 95% confidence interval. **e**, Multivariate Cox regression analysis for event-free survival of both cohorts (*n* = 152), considering mutation densities at MRCA, acquired mechanisms of telomere maintenance, disease stage at diagnosis, age at diagnosis and functional mutations in the RAS/p53 pathway. Shown are mean hazard ratio, 95% confidence intervals and *P* values for each variable (two-sided Wald test; *, *P* < 0.05).

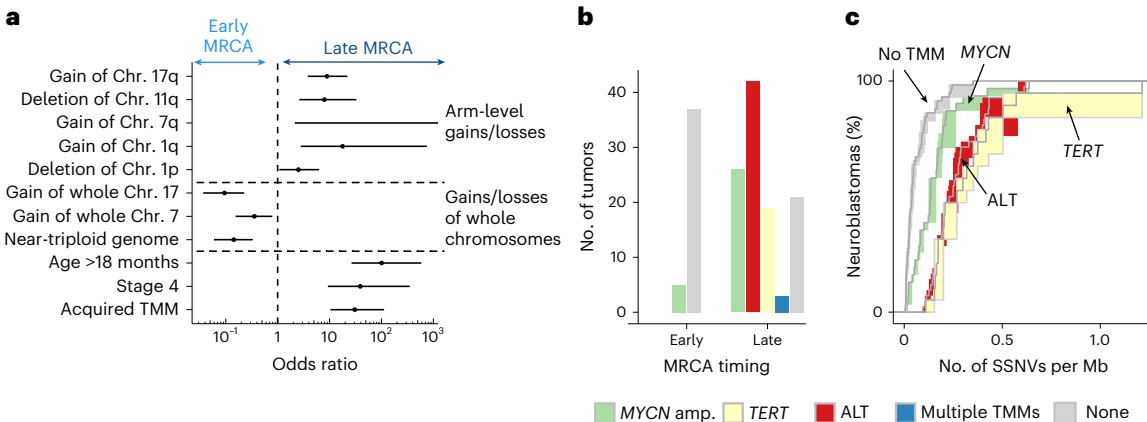

**Fig. 6 | Early genomic instability is associated with unfavorable outcome.**
**a**, Enrichment of chromosomal aberrations and clinical parameters in early- and
late-MRCA tumors predicting favorable and unfavorable outcome, respectively.
Shown are odds ratios (centers) with 95% confidence intervals (error bars)
between tumors with late and early MRCA for characteristics with significant
enrichment (*P* < 0.05 according to two-sided Fisher's exact test; exact P values
are provided in Source data). The odds ratio for 7q gain is infinite and hence only

the lower bound is displayed. **b**, Prevalence of TMMs across early- and late-MRCA
neuroblastomas. **c**, Cumulative mutation densities of the MRCA stratified by
TMM. Shown are 150 primary tumors and metastases of both cohorts, excluding
three tumors with multiple TMM. Solid lines represent maximum-likelihood
estimates and shaded areas represent 95% confidence intervals obtained by
bootstrapping.

(including gains of 17q, 7q and 1q, as well as loss of 11q and 1p) was preva-
lent in the late-MRCA group. In late-MRCA tumors with timeable ECA,
the vast majority of segmental gains that we could time were coinci-
dent with the ECA (Figs. 3c and 4e, dark green squares). Taken together,
different processes underlie the early acquisition of aneuploidy in the two
classes: mis-segregation of entire chromosomes in early-MRCA tumors
vis-à-vis genomic instability in late-MRCA tumors (Fig. 6a).

Acquired TMMs were strongly enriched in the late-MRCA class
(Fig. 6a); 12% of early-MRCA tumors but 81% of late-MRCA tumors
gained telomere maintenance—via *MYCN* amplification, ALT or *TERT*
rearrangement—in the combined discovery and validation cohorts
(Fig. 6b). Due to their small size, these structural rearrangements
cannot be timed using mutation densities and hence may have been
acquired before the emergence of arm-level aneuploidy, together with
aneuploidy or subsequently. Early timing of arm-level aneuploidy in
more than one-half of the late-MRCA cases (that is, tumors with an
ECA; left-hand panels in Figs. 3c and 4e) suggests that telomere main-
tenance was gained during a secondary event between ECA and MRCA
in these cases, which mainly included ALT and *TERT* rearrangement.
In the remaining cases, where arm-level aneuploidy was timed at the
MRCA and no ECA was identifiable (right-hand panels in Figs. 3c and
4e), telomere maintenance could also have been acquired in an onco-
genic event preceding the MRCA. Interestingly, those cases with *MYCN*
amplification fell predominantly within this latter group (Fisher's exact
test, *P* = 0.006055, odds ratio = 3.9 (1.4, 11.6)), suggesting that *MYCN*
amplification tends to occur earlier than ALT or *TERT* rearrangement.
Indeed, *MYCN*-amplified tumors generally had an earlier MRCA than
tumors with ALT or *TERT* rearrangement (Fig. 6c).

Collectively, these findings establish a link between genetic evo-
lution of neuroblastoma and the observation that neuroblastomas
with extensive arm-level aneuploidy tend to carry a poor prognosis[30].
The typically late-emerging MRCA in these cases indicates that the
underlying genomic instability predisposes these tumors to prolonged
evolution, including the acquisition of TMMs.

**Chromosomal gains occur during sympathetic neurogenesis**
Finally, we asked whether our genetic insights into neuroblastoma
development could be synthesized into an integrative model of tumor
evolution. To this end, we devised population-genetic models and

quantified key parameters by fitting the models to our measured data,
including the time-dependent incidences of ECA and MRCA and the
variant allele frequency (VAF) distribution of somatic variants. In addi-
tion, we required the models to reproduce the overall incidence of the
disease in the human population. For this reason, we focused on neuro-
blastomas with poor prognosis ('high-risk', enriched in the late-MRCA
class), which occur in around one in $10^5$ children (the frequency of
low-risk cases, enriched in the early-MRCA class, is not reliably known
due to incomplete diagnosis[31–34]).

The models describe a proliferative population of putative cells
of origin that are lost by either differentiation or cell death (Fig. 7). On
average, $\mu$ SSNVs occur per cell division. A rare subset of mutations
(including chromosomal gains/losses and smaller-scale events, such
as SSNVs or localized *MYCN* amplification) will be oncogenic drivers
and give rise to selected cell clones. As a minimal requirement for the
evolution of a high-risk tumor, we accounted for two oncogenic events,
defining ECA and MRCA, with mutation frequencies $\mu_1$ and $\mu_2$ per cell
division. To determine these parameters, the model was fit to the
experimental data using approximate Bayesian computation. Initially,
we assumed that cells of origin are generated during fetal develop-
ment and then remain available for neuroblastoma development. This
simple model consistently overestimated the overall incidence of
high-risk disease (Fig. 8a). By contrast, a model in which putative
cells of origin were available for only a limited time window (Fig. 8b)
accurately reproduced both the overall incidence and dynamics of the
emergence of ECA and MRCA (Fig. 8c–e and Extended Data Fig. 5a).

The inferred rate of SSNV acquisition ($\mu$) was 3.2 ± 0.4 SSNVs per
day, consistent with measurements of somatic mutation rate in the
developing central nervous system (5.1 (1.5, 9.0; 95% confidence interval)
SSNVs per day)[35]. The model further inferred that oncogenic driver
events occurred on average once per 1 million cell divisions (geometric
mean of $\mu_1$ and $\mu_2$), which is consistent with a global estimate of driver
mutation rate of $3.4 \times 10^{-5}$ per division in the human genome[13], given
that only a subset of all drivers will cause neuroblastoma. We used SSNV
rate to calibrate the molecular clock against real time, which placed
the ECA within the first trimester of pregnancy (Fig. 8c), at which time
rapidly dividing sympathetic neuroblasts are developing[36]. This first hit
(ECA) sustains a subclone in which the MRCA emerges due to a further
oncogenic event (Extended Data Fig. 5b) which, in the majority of cases,

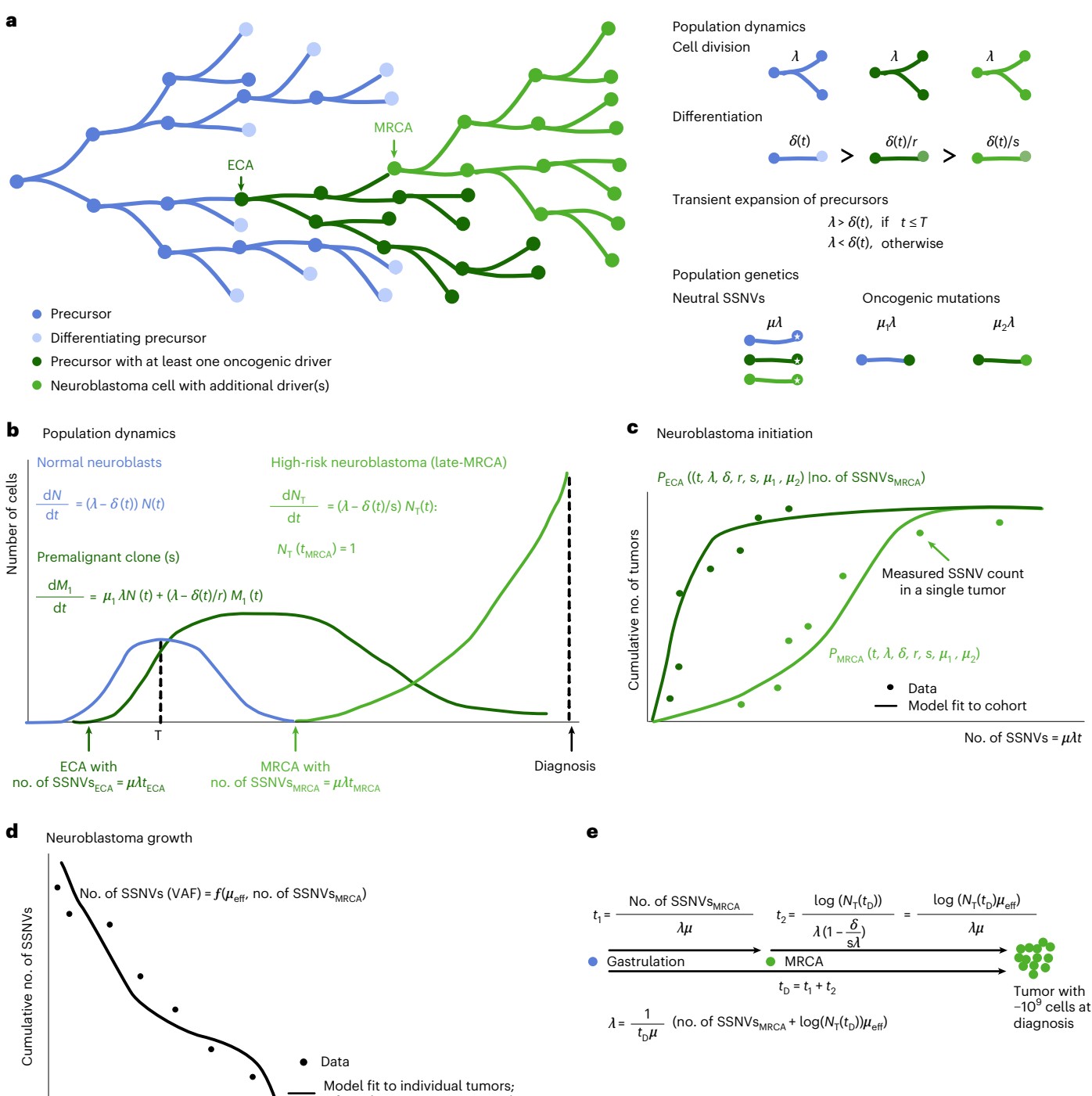

**Fig. 7 | Population genetics models of tumor initiation and growth. a,** Model scheme. Neuronal precursors divide and differentiate. Oncogenic events at ECA and MRCA cause the outgrowth of premalignant (ECA) and malignant (MRCA) clones. Cells divide at rate $\lambda$ and differentiate at rate $\delta$; oncogenic mutations reduce the loss rate by a factor $1/r$ (first event) and $1/s$ (second event). Neutral SSNVs are acquired at rate $\mu\lambda$ in each cell while driver mutations are acquired at lower rates $\mu_1\lambda$ (corresponding to the first oncogenic event and defining the ECA of the tumor) and $\mu_2\lambda$ (corresponding to the second oncogenic event and defining the MRCA of the tumor). **b,** Population dynamics of normal neuroblasts ($N$), premalignant clones (harboring one oncogenic event, $M_1$) and high-risk neuroblastomas (harboring two oncogenic events, $N_T$). Rates are defined in **a. c,** The model outlined in **a,b** yields a probability distribution for the time point at which the MRCA emerges ($P_{MRCA}(t)$); likewise, it also yields a conditional probability for the time point at which the ECA emerged, given the mutation density in the MRCA ($P_{ECA}(t|$no. of SSNVs$_{MRCA}$)). The parameters

of both probability distributions can be estimated from the measured SSNV counts at ECA and MRCA across the cohort. **d,** Model of mutation accumulation during neuroblastoma growth. Neutral SSNVs are continuously acquired at rate $\mu_{eff}$, defined as the number of neutral mutations per effective division (where one effective division produces two surviving daughter cells). By fitting the model to the measured VAF distribution, an estimate for $\mu_{eff}$ is obtained on the level of individual tumors (subsetting on cases with sufficient data quality and excluding tumors with evidence for subclonal selection during tumor growth). **e,** Estimation of division rate in actual time. The time between gastrulation and diagnosis ($t_D$) consists of a premalignant time span, up to the emergence of the MRCA ($t_1$), and the expansion of the tumor thereafter ($t_2$). Assuming exponential tumor growth and approximately $10^9$ tumor cells at diagnosis, this yields an estimate for the division rate per tumor for the subset used to estimate $\mu_{eff}$. To obtain an estimate for the population level, these estimates are subsequently averaged across the cohort.

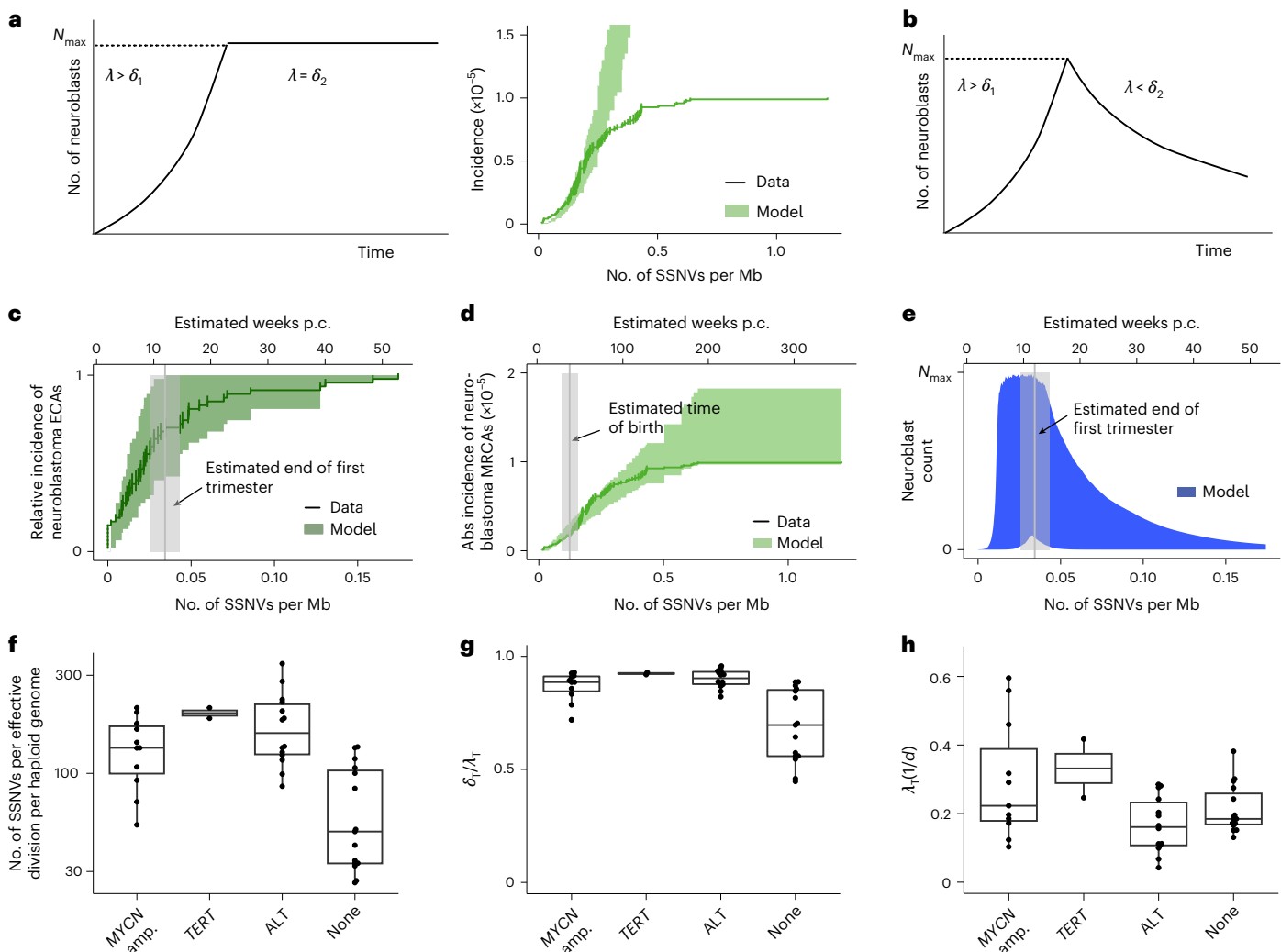

**Fig. 8 | Evolutionary dynamics of neuroblastoma initiation. a**, Assuming a constant putative population of origin following neuroblast expansion (left panel), the model fit does not capture the observed saturating incidence of high-risk neuroblastomas (right panel; shaded areas, 95% posterior probability of the fit; data, mutation densities of the MRCA from primary neuroblastomas with an acquired TMM, combining discovery and validation cohort; solid line, maximum-likelihood estimates of data with error bars representing s.d. estimated by bootstrapping). **b**, Transient putative population of origin. **c,d**, Model fits to ECA (**c**) and MRCA (**d**) with transient population of origin, as in **b** accounts for the experimental data [green shaded areas, 95% posterior probability bounds; vertical lines and shaded areas, mean and 95% CI of the estimated end of the first trimester (12 weeks p.c.) and of the time of birth (38 weeks p.c.); data, mutation densities of ECA (dark green, $n = 47$) and MRCA (light green, $n = 95$) from primary neuroblastomas (tumors/metastases) with acquired TMM; solid lines, maximum-likelihood estimates; error bars, s.d. estimated by bootstrapping]. **e**, Predicted transient expansion of putative cells of origin, agreeing with rapid proliferative

phase of sympathetic neuroblasts (shaded area, 95% posterior probability; vertical line and shaded area, mean and 95% CI of the estimated end of the first trimester (12 weeks p.c.)). **f**, Estimated mutation rate per effective cell division, computed from primary tumors/metastases with *MYCN* amplification (amp.) ($n = 11$), *TERT* rearrangement ($n = 2$), ALT ($n = 14$) or no acquired TMMs ($n = 15$), using cases with highly accurate subclonal VAF distribution due to high tumor purity. Boxes represent median and 25 and 75 percentiles; whiskers extend to the smallest and largest values within 1.5× interquartile range. **g,h**, Estimated loss rate (relative to cell division, **g**) and cell division rate (**h**) in primary tumors and metastases analyzed in **f** (shown are $n = 11$ primary tumors with *MYCN* amplification, $n = 2$ primary tumors with *TERT* rearrangement, $n = 14$ primary tumors with ALT and $n = 15$ primary tumors without acquired telomere maintenance). Boxes represent median and 25 and 75 percentiles; whiskers extend to the smallest and largest values within 1.5× interquartile range; $\lambda_T$, division rate during tumor growth; $\delta_T$, loss rate during tumor growth.

occurred within the first year of life (Fig. 8d). The model fit of Fig. 8 was based on all SSNVs, because our data (Extended Data Figs. 1e and 2d) suggest that mutational processes may not change markedly during neuroblastoma evolution. To test the robustness of our inference, we also performed all analyses with the subset of clock-like SSNVs as input. The inferred rate of clock-like SSNV acquisition ($\mu$) was 2.3 (1.2, 2.3; 80% credible interval) per division (corresponding to 2.2 ± 0.3 SSNVs per day). All other inferred parameters remained practically unchanged (Extended Data Fig. 5c,d). Hence, confining the analysis to clock-like mutations corroborates the real-time calibration of ECA and MRCA.

A further insight afforded by the population-genetic model is the extent of cell loss in the growing tumor. In general, only a subset of cell lineages will support growth by continued symmetric self-renewal of malignant cells whereas other lineages will terminate by either cell differentiation into nonproliferating states or cell death. We inferred the ratio of self-amplifying tumor cell divisions among all divisions from the subclonal tail of VAF distribution (Methods and ref. [37]), finding that only ~10% of tumor cell divisions result in growth of late-MRCA neuroblastomas that acquired TMMs (Fig. 8f,g). This inference is consistent with the clinical observation of extensive cell death in such

neuroblastomas. Moreover, the average cell division rate was lower in neuroblastomas with ALT than in those with *MYCN* amplification or *TERT* rearrangement (Fig. 8h), again in line with clinical observation; our estimated cell division rates agree quantitatively with those measured in neuroblastoma in vivo[38]. The fraction of self-renewing cell divisions supporting tumor growth was inferred from VAF distribution also for tumors without acquired TMMs, all falling within the early-MRCA class, yielding a higher value of ~30% (Fig. 8g). Hence, cell loss appears to be a stronger selective pressure for late-MRCA neuroblastomas, the majority of which gain TMMs, than for early-MRCA neuroblastomas, which rarely acquire such mechanisms.

## Discussion

In this paper we timed genetic events in the evolution of neuroblastoma using the molecular clock of SSNV accumulation and, inferring the rate of SSNV acquisition from the distribution of VAFs, related this clock to real time by factoring in the age at diagnosis. In two-thirds of cases we find that chromosomal gains implicated in the pathogenesis of the disease occurred early, and typically within the first trimester of pregnancy. With respect to further evolution, these cases fall into two distinct classes: in the early-MRCA class the early chromosomal gain event also marked clonal outgrowth of the resected tumor whereas in the late-MRCA class the tumors evolved further before clonal outgrowth commenced. Remarkably, in our cohort MRCA class is an accurate predictor of clinical outcome. This is true regardless of whether we could time an early chromosomal gain, and implies that neuroblastomas with a longer evolutionary history are more aggressive. Because the strong association between MRCA timing and outcome was also present with 30× WGS, the utility of this predictor for patient stratification may be tested in clinical trials.

Our real-time inference shows that neuroblastomas across the entire clinical spectrum acquired aneuploidy within the first trimester of pregnancy, when the adrenal medulla forms from sympathetic neuroblasts. Moreover, matching disease incidence with the population-genetic model suggests that the initial oncogenic event is limited to this time window. The transcriptomes of neuroblastomas most resemble those of sympathetic neuroblasts[15,36]. In this early window, neuroblasts are highly proliferative, which may make them vulnerable to aneuploidy. Finally, the observation that aneuploidy via near-triploidization is a temporally confined event is consistent with the long-standing hypothesis that this karyotype results from endoreduplication of the genome followed by tripolar cell division and selection of the fittest daughter cell[39,40].

The molecular nature of early aneuploidy is associated with whether the tumor continues to evolve: neuroblastomas with whole-chromosome aneuploidy typically did not evolve further and were overall associated with favorable outcomes; in contrast, most tumors with early genomic instability had unfavorable outcomes. Continued evolution of such tumors has also been noted in a recent study taking multiregion biopsies[41], emphasizing the potential of spatially resolved genetic and transcriptomic analysis[42,43]. We did not detect specific drivers of genomic instability: in particular, p53 function was not impaired genetically. However, with prevalent 17q gains, reduced expression of *TP53* (located on 17p) relative to driver genes expressed on 17q (for example, *BIRC5*, *IGF2-BP1* and *BRIP1*) may favor genomic instability[44,45].

Accurate risk stratification in neuroblastoma remains a major concern. Our data suggest a link between diverse criteria—age at diagnosis, segmental versus whole-chromosome gains and losses and acquisition of TMM[25,46]—based on how neuroblastomas evolve. We find that a greater age at diagnosis is often linked to longer evolution of the tumor rather than later origin. Paradoxically, this implies that low-risk tumors reach detectable size earlier than high-risk. Indeed, we infer that low-risk tumors have a substantially lower fraction of tumor cell loss than high-risk tumors and hence should grow faster.

Acquired TMMs should, consequently, provide a larger selective advantage in the high-risk, late-MRCA group where they indeed are enriched. Hence, late-MRCA tumors may grow more rapidly only after gaining telomere maintenance (similar to IDH-wild-type glioblastomas[7]) and hence are diagnosed later. Interestingly, we detected five neuroblastomas with amplified *MYCN* in the early-MRCA class, and, remarkably, all of these patients survived until the end of the observation periods (1,447–4,177 days), in contrast to the poor survival of patients with *MYCN*-amplified tumors in the late-MRCA group. Our findings suggest that MRCA timing may be worth considering as a parameter for patient stratification.

## Online content

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

## Methods

### Patient cohorts, tumor samples and ethical approval

Cohorts of primary and relapsed neuroblastoma tumors were retrospectively analyzed. Tumor material was collected as part of the diagnostic workflow of the German Neuroblastoma trial by the Society for Pediatric Oncology and Hematology and collected in the Neuroblastoma tumor bank. All trials were approved by the Ethics Committee of the Medical Faculty, University of Cologne, and collection and use of all tumor tissue material was approved (registry nos. NB97, NB2004 and NB2016). This study includes data of neuroblastoma tumors previously published in Hartlieb et al.[24] and Peifer et al.[26] The study by Hartlieb et al. also contains a subset of tumors from the St. Anna Kinderkrebsforschung at the Children's Cancer Research Institute in Vienna, Austria, as well as tumors analyzed in the registry trial INFORM[47]. The study office of the Neuroblastoma trial in Cologne provided clinical annotations and survival information. All patients or their legal guardian approved the use of tumor material by signed informed consent. For analysis, all resected tumors were divided into four quadrants, all of which were evaluated histologically. *MYCN* status was assessed as a routine clinical marker for all tumors using fluorescence in situ hybridization. A cross-sectional slice of one quadrant was used for DNA extraction for WGS; the same quadrant was used for ploidy analysis, measuring the DNA index. DNA was isolated using phenol chloroform extraction from fresh-frozen tumor material. Control DNA was isolated from whole blood using the NucleoSpin Blood DNA extraction kit (Macherey-Nagel) according to the manufacturers' instructions. Details of the included samples are provided in Supplementary Tables 1 and 6.

### WGS

WGS workflows for the previously published data are described in ref. [24]. For additional samples, high-coverage, WGS was performed on a patterned flowcell v.2.5 (150-bp, paired-end) with coverage of about 80× for the tumor and whole-blood control samples. All tumors had a histological tumor cell content of ≥60%. Sequencing libraries were prepared using the Truseq DNA Nano kit (Illumina) according to the manufacturers' instructions, and size selected using SPRI beads (Beckman Coulter Genomics). Alignment and variant calling was done using the One Touch Pipeline service of the German Cancer Research Center (DKFZ)[48]. Alignment was done using workflow v.1.2.73-1, available at Github (https://github.com/DKFZ-ODCF/AlignmentAndQCWorkflows). In brief, sequences were aligned to the 1000 Genomes project assembly with decoy and PhiX contigs using BWA-MEM v.0.7.15 with option '-T 0'. Merging and duplication marking were performed using Sambamba v.0.6.5, and bam files were filtered using Samtools v.0.1.19. Calling of SSNVs, somatic Indels, copy number variations and SVs was done using inhouse workflows, available at https://github.com/DKFZ-ODCF/SNVCallingWorkflow, https://github.com/DKFZ-ODCF/IndelCallingWorkflow, https://github.com/DKFZ-ODCF/ACEseqWorkflow and https://github.com/DKFZ-ODCF/SophiaWorkflow.

Estimates of tumor cell content were manually adjusted in one case (NBE40) following visual inspection of VAF distribution. Structural variants were excluded in the present study if they had a minimal event score of <5; focal amplifications were defined as regions with copy number ≥10; homozygous deletions were defined as regions with copy number <0.9. Deletions of (parts of) chromosomal arm 1p were defined as p-terminal regions lost relative to 1q and, moreover, with copy number ≤1 in near-diploid, ≤2 in near-triploid or ≤3 in near-tetraploid tumors. In analogy, we annotated deletions of chromosome 11q if the copy number was ≤1 in near-diploid, ≤2 in near-triploid or ≤3 in near-tetraploid tumors, and if 11q was lost relative to 11p. For gains on chromosomes 1, 2, 7 and 17 we required the copy number to be higher than the (rounded) basal ploidy of the tumor. Partial gains on 1q, 2p, 7q and 17q were defined as regions on the respective chromosomal arm that were gained relative to the other chromosomal arm.

**Mutational signatures.** Mutational signatures were learned de novo and thereafter decomposed into Cosmic mutational signatures v.3.1 (ref. [49]) (http://cancer.sanger.ac.uk/cosmic/signatures) using SigProfilerExtractor (v.1.1.1)[50]. Only signatures contributing to ≥5% of the mutations in at least one sample and, in addition, identified in at least 10% of samples, were considered. The contributions of these signatures to each sample were then re-estimated using the R package mmsig v.0.0.0.9000 (ref. [51]) (setting strandbias=F, bootstrap=F, cos_sim_threshold=0.01, force_include=c("SBS1", "SBS5")). For visualization, signatures SBS1, SBS5 and SBS40 were combined into a single, clock-like mutational signature.

Stratified analysis of clonal and subclonal mutations was performed by classifying mutations as subclonal if $\sum_{k=0}^{n_{var}} \binom{n_{var} + n_{ref}}{k} p^k (1-p)^{n_{var}+n_{ref}} < 0.05; p = \frac{\rho}{\rho CN + 2(1-\rho)}$, where $\rho$ is estimated tumor cell content, $n_{var}$ and $n_{ref}$ are the number of variant and reference reads, respectively, and CN is copy number.

**Mutation timing. Estimation of numbers of amplified and non-amplified clonal mutations.** We estimated mutation densities at partial and entire chromosomal gains, and at the MRCA of the tumor from the distribution of VAFs among clonal mutations. To this end we counted clonal mutations separately on each autosome, stratified by copy number (CN) state. Regions lacking a CN estimate were assigned to a specific CN state if the measured coverage ratio (CR) and B-allele frequency (BAF) matched the expected ratios within measurement error. Specifically, we required for each segment $i$, $\left[\frac{CN_i \rho + 2(1-\rho)}{\pi \rho + 2(1-\rho)} - 0.1\right] \leq CR_i \leq \left[\frac{CN_i \rho + 2(1-\rho)}{\pi \rho + 2(1-\rho)} + 0.1\right]$ and $\left[\frac{b_i}{CN_i} - 0.05\right] \leq BAF_i \leq \left[\frac{b_i}{CN_i} + 0.05\right]$, where $CN_i$ is the copy number of segment $i$, $\rho$ the tumor cell content, $\pi$ the average tumor ploidy and $b$ the number of B-alleles. States with copy number >4 or of size ≤$10^7$ bp were excluded from the analysis because the statistics become ambiguous for short pieces and high CN states.

On each retained segment we estimated the number of clonal mutations, distinguishing clonal mutations present on a single allele ('non-amplified clonal mutations') from those present on multiple alleles ('amplified clonal mutations'). To this end we applied a statistical framework[52], distinguishing amplified clonal mutations acquired before a clonal gain and thus present on all A-alleles, or on all B-alleles at a given CN state, from non-amplified clonal mutations acquired either on the non-amplified allele or after clonal gain but before the MRCA and hence present on a single copy; the remaining mutations are subclonal. Accordingly, we expect to find clonal mutations at VAFs $\in \left\{\frac{1}{CN}, \frac{CN-b}{CN}, \frac{b}{CN}\right\}$ in a pure sample, and at VAFs $\in \left\{\frac{\rho}{\zeta}, \frac{(CN-b)\rho}{\zeta}, \frac{b\rho}{\zeta}\right\}$ in an impure sample with tumor cell content $\rho$, where:

$$\zeta = \rho CN + 2(1-\rho) \tag{1}$$

is the average copy number of a given locus in the sample. Because measured VAFs are randomly distributed around their true values, we fit a binomial mixture model to the clonal mutations. To avoid misclassification of subclonal mutations as clonal, we neglected all variants with VAF $< \frac{\rho}{\zeta}$. By symmetry, this cutoff retains half of the non-amplified clonal mutations, introducing a subsequent correction factor of 2 (equation (4)). Defining the weights $w = (w_1, w_{CN-b}, w_b)$, for the non-amplified clonal and amplified clonal mutation peak, respectively, we computed the probability of measuring $n_{var,j}$ variant reads and $n_{ref,j}$ reference (nonvariant) reads at the position of the $j$th SSNV to:

$$P(n_{var,j}, n_{ref,j}|CN, \rho, w) = \sum_{k \in \{1, CN-b, b\}} B\left(n_{var,j}; n_{var,j} + n_{ref,j}, \frac{\rho k}{\zeta}\right) P(k|w), \tag{2}$$

where $B\left(n_{var,j}; n_{var,i} + n_{ref,j}, \frac{\rho k}{\zeta}\right)$ is the binomial probability for drawing $n_{var,j}$ variant reads at sequencing depth $n_{var,j} + n_{ref,j}$ from the peak

comprising mutations that are clonal on $k$ chromosomal copies. Defining a uniform prior probability, $P(w)$, for the weights, we computed, up to normalization, the posterior probability as:

$$P(w|\{n_{\text{var},j}, n_{\text{ref},j}\}_j^N, \text{CN}, \rho) \propto \prod_{j=1}^{N} P(n_{\text{var},j}, n_{\text{ref},j}|\text{CN}, \rho, w) P(w), \quad (3)$$

where $N$ is the total number of SSNVs under consideration. Clonal mutations were then assigned to distinct clonal peaks according to the weights at maximum a posteriori probability (MAP), yielding, on segment $l$, estimates for the number of clonal mutations on non-amplified chromosomes, $n_{1,l}$, on amplified $b$ alleles, $n_{b,l}$ (if $b > 1$) and on amplified $a$ alleles (with copy number CN − $b$), $n_{\text{CN}-b,l}$, according to:

$$n_{1,l} = 2\text{MAP}(w_{1,l}) N_l, n_{b,l} = 2\text{MAP}(w_{b,l}) N_l, \text{ and } n_{\text{CN}-b,l} = \text{MAP}(w_{\text{CN}-b,l}) N_l. \quad (4)$$

**Mutation timing. Timing MRCA and ECA.** Mutation densities (SSNVs per bp) at the MRCA were estimated from the number of non-amplified clonal mutations and total size of the analyzed genome, $g = \sum_l g_l$ (ref. [52]). The number of mutations per copy that were acquired up to the MRCA is

$$n_l = \frac{n_{1,l} + n_{\text{CN}-b,l}(\text{CN}_l - b_l) + n_{b,l}b_l}{\text{CN}_l}. \quad (5)$$

If tumor samples were well mixed, or tumors completely sampled, mutation densities at the MRCA could be directly estimated as $n_l/g$. In practice, however, $n_l$ may consist of a set of true clonal mutations, acquired before tumor growth, and an additional set of mutations that appear as clonal in the tumor sample due to incomplete sampling. To correct for the latter, false-positive (FP), clonal mutations we compared primary and relapse samples from two such pairs available in our dataset (NBE11/NBE66, NBE51/NBE78). The fraction of conservative clonal mutations in the primary sample that remained undetected in the relapse sample was 15 ± 5% and was taken as the fraction of FP. With this correction, the mutation count, $m_{\text{MRCA}}$, and mutation density, $\tilde{m}_{\text{MRCA}}$, at MRCA were estimated as:

$$m_{\text{MRCA}} = \sum_l n_l (1 - \text{FP}) \text{ and } \tilde{m}_{\text{MRCA}} = \frac{m_{\text{MRCA}}}{g}, \quad (6)$$

respectively. Lower and upper 95% confidence bounds for $\tilde{m}_{\text{MRCA}}$ were estimated by bootstrapping the genomic segments 1,000 times.

Next, we tested for each gained segment whether amplified clonal mutations were significantly less frequent than expected at the MRCA and, accordingly, assigned the segment to either the MRCA or an earlier time point (in the majority of cases the ECA; here we excluded a small number of segments in seven tumors with a higher density of amplified clonal mutations than the estimated mutation density at the MRCA, because such gains may be subclonal CN alterations erroneously classified as clonal). To this end, we modeled the number of mutations falling on each genomic segment with a negative binomial distribution, which accounts for overdispersion caused by heterogeneous mutation rates along the genome[53]. The probability that the gain of genomic segment $l$ coincided with the MRCA is then computed as:

$$P(n_{k,l;k\in\{b,\text{CN}-b\}}|g_l, g, m_{\text{MRCA}}) = \sum_{r=0}^{n_{k,l}} \binom{m_{\text{MRCA}} + r - 1}{m_{\text{MRCA}}} p^r (1-p)^{m_{\text{MRCA}}}, \quad (7)$$

where:

$$p = \frac{m_{\text{MRCA}}}{m_{\text{MRCA}}\left(1 + \frac{g_l}{g}\right)}.$$

Here, $n_{k,l;k\in\{b,\text{CN}-b\}}$ is the number of amplified clonal mutations on segment $l$ and $g_l$ is the respective segment size. We corrected the

$P$ values obtained with equation (7) for multiple testing using Holm correction (false discovery rate ≤ 0.01) and, accordingly, assigned each segment to either the MRCA or an earlier time point.

Finally, we computed the mutation densities at ECAs from the segments with significance level $\alpha = 0.01$ to:

$$\tilde{m}_{\text{ECA}} = \frac{\sum_{l,p_{\text{adj},l}\leq 0.01} n_{b,l} + n_{\text{CN}-b,l}}{\sum_{l,p_{\text{adj},l}\leq 0.01} g_{l,b} + g_{l,\text{CN}-b}}. \quad (8)$$

We then tested for each contributing segment whether its mutation load conformed to the ECA according to a negative binomial distribution (in analogy to equation (7)) and computed lower and upper 95% confidence bounds by bootstrapping, as before for the MRCA.

**Mutation timing. Translation of mutation densities into estimated weeks p.c.** We related mutation densities per haploid genome into weeks post conception (p.c.) by inferring SSNV rates per diploid genome and embryonic day ($\mu\lambda$) and mutation and division rates per day, using the measured VAF distributions and age at diagnosis as outlined in Real-time estimation of cell division rate (with similar results based on all SSNVs or only clock-like SSNVs). Because mutation calling was performed by comparing tumors against a matched blood control, mutation densities correlate with the time post gastrulation (at approximately 2 weeks p.c.). Thus, the mutation density per haploid genome, $\tilde{m}$, relates to the time p.c., $t$, according to $\tilde{m}(t) = \frac{\mu\lambda}{d} \frac{1}{3.3\times10^9}(t - 14 \text{ days})$.

The estimated time of birth was taken as 38 weeks after gastrulation (40 weeks p.c.).

## Survival analysis

Survival analysis was performed using the R package survival v.3.1.12 (ref. [54]). We detected a clear bimodality in the histogram of SSNV density at the MRCA in the discovery cohort and took the upper bin border just before the minimum (0.05 SSNVs per Mb) as threshold to split tumors into groups of early or late MRCA. The same value was found appropriate in the validation cohort.

## Modeling neuroblastoma initiation

**Modeling emergence of the MRCA.** We modeled neuroblastoma initiation by sequential acquisition of driver mutations in two oncogenic events, assuming that both events are infrequent and occurring in early neuroblasts with low probabilities, $\mu_1$ and $\mu_2$, during cell divisions. We denote the selective advantage conferred by the driver mutations acquired during the two events by $r$ and $s$. Specifically, we assume that one or more first events generate a precancerous cell population in which a second event creates the MRCA. On this basis, we compute the time-dependent probability of the emergence of the MRCA. Neuroblasts initially expand rapidly, but how this cell population behaves subsequently, because sympathetic neurons differentiate from these precursors, is not known precisely. Hence, two possible scenarios were considered: (1) an initial phase of exponential growth of the neuroblast population is followed by a subsequent phase of differentiation, modeled by exponential decay; and (2) an initial phase of exponential growth is followed by a subsequent phase of precursor homeostasis (for a related model of a two-step process of tumorigenesis, but in a homeostatic tissue, see ref. [55]). The two phases are associated with distinct rates of cell division, $\lambda_1$ and $\lambda_2$, and loss, $\delta_1$ and $\delta_2$, where $\lambda_1 > \delta_1$ and $\lambda_2 \leq \delta_2$ (with equality in the case of precursor homeostasis). Thus the population dynamics of neural precursor cells, $N(t)$, are described by:

$$N(t) = \begin{cases} e^{(\lambda_1 - \delta_1)t}, & 0 \leq t \leq T \\ e^{(\lambda_1 - \delta_1)T}e^{(\lambda_2 - \delta_2)(t-T)}, & t > T, \end{cases} \quad (9)$$

where $T$ denotes the time point at which the cell population peaks. Cells undergoing the first oncogenic event are generated at rates:

$$\mu_1\lambda_1 N(t)\mathrm{d}t,\ 0 \le t \le T$$
$$\mu_1\lambda_2 N(t)\mathrm{d}t,\ t > T. \tag{10}$$

For simplicity, we take the selective advantage associated with the first oncogenic event into account during the contraction (or homeostasis) phase and neglect it during the initial rapid expansion; this approximation is appropriate when the selective advantage of the first event is comparatively small, which is subsequently borne out by the parameter inference. Thus, $M_1$ cells harboring the first mutation grow at rate $\lambda_1 - \delta_1$ during tissue expansion and at rate $\lambda_2 - \delta_2/r$ during tissue contraction or homeostasis. Hence, depending on the actual value of $r > 1$, a clone harboring the first event may slowly shrink or expand for $t > T$. We now ask whether in this clone the second oncogenic event takes place that defines the MRCA of the tumor. The second event occurs at rate:

$$\mu_2\lambda_1 M_1(t)\mathrm{d}t,\ 0 \le t \le T$$
$$\mu_2\lambda_2 M_1(t)\mathrm{d}t,\ t > T. \tag{11}$$

Using the survival probability of the supercritical birth–death process, we have a cell undergoing the second oncogenic event survive with probability:

$$v_{2,\mathrm{E}} = 1 - \frac{\delta_1}{s\lambda_1} \tag{12a}$$

during the expansion phase (E) and, provided that $\frac{\delta_2}{s\lambda_2} < 1$, with probability:

$$v_{2,\mathrm{D}} = 1 - \frac{\delta_2}{s\lambda_2} \tag{12b}$$

during the decay or homeostatic phase (D). We are interested in the probability that at least one surviving cell underwent both oncogenic events, $P_{\mathrm{MRCA}}$. There are three possible cases: (1) both oncogenic events occur during precursor expansion, associated with probability $P_{\mathrm{MRCA,1}}$; (2) the first oncogenic event occurs during precursor expansion, the second during precursor contraction or homeostasis, associated with probability $P_{\mathrm{MRCA,2}}$; and (3) both oncogenic events occur during precursor contraction or homeostasis, associated with probability $P_{\mathrm{MRCA,3}}$. For each case we assumed that the number of cells with only the first event, $M_1$, is small compared with the number of normal cells. We have:

$$P_{\mathrm{MRCA}} = \begin{cases} P_{\mathrm{MRCA,1}}, t < T \\ 1 - (1 - P_{\mathrm{MRCA,1}})(1 - P_{\mathrm{MRCA,2}})(1 - P_{\mathrm{MRCA,3}}), t \ge T \end{cases}. \tag{13}$$

We derive expressions for $P_{\mathrm{MRCA,1}}$, $P_{\mathrm{MRCA,2}}$ and $P_{\mathrm{MRCA,3}}$ in Supplementary Note 1a for the case of precursor expansion and decay, and in Supplementary Note 2a for the case of precursor expansion and homeostasis. If precursor expansion is followed by decay, we find:

$$P_{\mathrm{MRCA,1}}(t) = \sum_{x=1}^{N(t)-1} e^{-\mu_1(x-1)}(1 - e^{-\mu_1})\left(1 - \exp\left\{-\frac{\mu_1\mu_2\lambda_1 TN(t)F}{1 - \frac{\delta_1}{\lambda_1}}\right\}\right), \tag{14a}$$

$$P_{\mathrm{MRCA,2}}(t) = 1 - \exp\left(-\frac{\mu_1\mu_2\lambda_1\lambda_2 v_{2,\mathrm{D}}T}{\lambda_2 - \delta_2/r}N(T)\left\{e^{(\lambda_2-\delta_2/r)(t-T)} - 1\right\}\right), \tag{14b}$$

$$P_{\mathrm{MRCA,3}}(t) = 1 - \exp\left(-\frac{\mu_1\mu_2\lambda_2^2 v_{2,\mathrm{D}}N(T)}{\delta_2\left(\frac{1}{r}-1\right)}\left\{\frac{e^{(\lambda_2-\delta_2)(t-T)}-1}{\lambda_2-\delta_2} - \frac{e^{(\lambda_2-\frac{\delta_2}{r})(t-T)}-1}{\lambda_2-\frac{\delta_2}{r}}\right\}\right), \tag{14c}$$

where $F = \int_0^1 v_{2,I}/(v_{2,I}z^\alpha)\,\mathrm{d}z$ and $\alpha = \frac{\delta_1-s\lambda_1}{\delta_1-\lambda_1}$. Alternatively, if precursor expansion is followed by homeostasis, $P_{\mathrm{MRCA,2}}$ and $P_{\mathrm{MRCA,3}}$ take the form (Supplementary Note 2a):

$$P_{\mathrm{MRCA,2}}(t) = 1 - \exp\left(-\frac{\mu_1\mu_2\lambda_1 v_{2,\mathrm{D}}T}{1 - \frac{1}{r}}N(T)\left\{e^{\lambda_2\left(1-\frac{1}{r}\right)(t-T)} - 1\right\}\right), \tag{14d}$$

$$P_{\mathrm{MRCA,3}}(t) = 1 - \exp\left(\frac{\mu_1\mu_2\lambda_2 v_{2,\mathrm{D}}}{1 - \frac{1}{r}}N(T)\left\{\frac{1 - e^{\lambda_2\left(1-\frac{1}{r}\right)(t-T)}}{\lambda_2\left(1-\frac{1}{r}\right)} + t - T\right\}\right). \tag{14e}$$

Equations (14a–e) give the probability that a growing tumor clone has emerged up to time $t$ for the distinct cases.

**Modeling the ECA.** The ECA is associated with the first oncogenic event, which may have occurred at variable time points before the MRCA. To time the ECA, we computed the conditional probability of having undergone the first oncogenic event before $t_1$, given that the second oncogenic event occurred at $t_2$, denoted by $P(t_1|t_2)$. The acquisition of the second oncogenic event is proportional to the number of cells with the first event $M_1$ (equation (11)). As denoted by $M_1(t_2|\tau)$, the number of cells at $t_2$ resulting from a first event at $\tau$. $P(t_1|t_2)$ can hence be expressed as:

$$P(t_1|t_2) = \frac{\int_0^{t_1} M_1(t_2|\tau)\,\mathrm{d}\tau}{\int_0^{t_2} M_1(t_2|\tau)\,\mathrm{d}\tau} = \frac{\int_0^{t_1} M_1(t_2|\tau)\,\mathrm{d}\tau}{M_1(t_2)}. \tag{15}$$

Distinguishing the three cases for the timing of the second event relative to the first, as before, we find for precursor expansion followed by decay for two cases (Supplementary Note 1b). When the second event occurs in the exponential growth phase, then:

$$P(t_1|t_2) = \frac{t_1}{t_2}; t_1 < t_2 \le T. \tag{16a}$$

This corresponds to the classical Luria–Delbrück model. Recalling that $t = 0$ marks the beginning of neuroblast expansion, the first mutation happens, as expected, with uniform probability during exponential growth. When the second event occurs during the decay phase, the probability for the first event is uniform in the exponential phase and decreases in the decay phase according to:

$$P(t_1|t_2) =$$
$$\frac{\Theta(T-t_1)\lambda_1\delta_2\left(1-\frac{1}{r}\right)t_1 + \Theta(t_1-T)[\lambda_2+\lambda_1 T\delta_2(1-1/r)-\lambda_2 e^{\delta_2(1-1/r)(T-t_1)}]}{\lambda_2+\lambda_1 T\delta_2(1-1/r)-\lambda_2 e^{\delta_2(1-1/r)(T-t_2)}}; \tag{16b}$$
$$t_2 > T$$

where $\Theta(\cdot)$ is the Heaviside step function and, of course, $0 \le t_1 < t_2$. For precursor expansion followed by homeostasis, $P(t_1|t_2)$ reads (Supplementary Note 2b):

$$P(t_1|t_2) = \frac{t_1}{t_2}; t_1 < t_2 \le T \tag{17a}$$

and:

$$P(t_1|t_2) =$$
$$\frac{\Theta(T-t_1)\lambda_1 t_1(1-1/r)e^{\lambda_2\left(1-\frac{1}{r}\right)(t_2-T)} + \Theta(t_1-T)\left[\left(\lambda_1 T\left(1-\frac{1}{r}\right)+1\right)e^{\lambda_2\left(1-\frac{1}{r}\right)(t_2-T)} - e^{\lambda_2\left(1-\frac{1}{r}\right)(t_2-t_1)}\right]}{\left(\lambda_1 T\left(1-\frac{1}{r}\right)+1\right)e^{\lambda_2\left(1-\frac{1}{r}\right)(t_2-T)}-1}; \tag{17b}$$
$$t_2 > T.$$

**Relating model and data.** To relate the model to the measured data, we translate time into mutation counts. Assuming that SSNVs

are accumulated at a constant rate, $\mu$, per cell division, the expected mutation count per cell is Poisson-distributed with rate $\mu\lambda t$.

**Parameter estimation.** We estimated the following parameters: peak size of the neuroblast population ($N$), relative loss rates in the growth and decay phases ($\delta_1/\lambda_1$ and $\delta_2/\lambda_2$, respectively), rate of SSNVs ($\mu$) and rates of first and second oncogenic events ($\mu_1$ and $\mu_2$) and their selective advantages (expressed as $r$ and $v_2$), using equations (9–17) and Approximate Bayesian Computation with Sequential Monte Carlo sampling (ABC–SMC) as implemented in pyABC[56]. We used a population size of 1,000 parameter samples and prior probabilities as outlined in Supplementary Table 11. Evaluation was abrogated after 25 SMC generations, or if $\varepsilon \leq 0.05$. We used mutation density estimates at MRCA and ECA of high-risk tumors (primary tumors and metastases) as determined by equations (6) and (8) as input data. Tetraploidization in tetraploid tumors was not included as ECA because there were probably earlier events, such as Chr. 17q gains. The experimental incidences were computed as $I_{MRCA,ev,i} = \sum \tilde{m}_{MRCA,ev} < i; i \in \tilde{m}_{MRCA,ev}$, where the subscript 'ev' denotes the experimentally determined value. Uncertainties were estimated to $\Delta I_{MRCA,ev,i} = \sum \tilde{m}_{MRCA,ev,l} < i - \sum \tilde{m}_{MRCA,ev,u} < i; i \in \tilde{m}_{MRCA,ev}$, where $\tilde{m}_{MRCA,ev,l}$ and $\tilde{m}_{MRCA,ev,u}$ denote the lower and upper bounds of the 95% confidence interval of $\tilde{m}_{MRCA,ev}$, respectively. Incidences and uncertainties of the ECA, $I_{ECA,ev,i}$ were computed in analogy.

For each sampled parameter set we performed the following steps:

(1) Sample for each tumor a time point of the MRCA, $t_2$. To this end, sample a uniform number $x$ between 0 and $10^{-5}$, thus accounting for the overall incidence of $10^{-5}$. Then, from equation (9), determine the time point at which $P_t = x$. To facilitate numerical computation, we approximated the sum in equation (13a) with an integral. To exclude second hits that do not confer a selective advantage during expansion, we required $s = \max\left(1, \frac{\delta_1}{\lambda_1(1-v_{2,E})}\right)$ and adjusted $v_2$ accordingly.

(2) Sample for each sampled time point $t_2$ a neutral mutation count from a Poisson distribution with mean $\mu t_2$ and determine mutation density by dividing with the haploid genome length of $3.3 \times 10^9$, yielding $\tilde{m}_{MRCA,sim}$.

(3) Determine the simulated incidence of the MRCA at the experimentally determined mutation loads:
$I_{MRCA,sim,i} = \sum \tilde{m}_{MRCA,sim} < i; i \in \tilde{m}_{MRCA,ev}$.

(4) Sample for each sampled $t_2$ a time point of the ECA, $t_1$. To this end, sample a uniform number $x$ between 0 and 1; then, from equation (16) or (17) determine the time point at which $P(t_1|t_2) = x$.

(5) Sample for each sampled $t_2$ a neutral mutation count from a Poisson distribution with mean $\mu t_2$ and divide by the haploid genome length of $3.3 \times 10^9$, yielding $\tilde{m}_{ECA,sim}$.

(6) Determine the simulated incidence of the ECA at the experimentally determined mutation loads:
$I_{ECA,sim,i} = \sum \tilde{m}_{ECA,sim} < i; i \in \tilde{m}_{ECA,ev}$.

(7) Determine the simulated incidence of the MRCA at age 10 years, $I_{MRCA,sim,ten years}$ using equation (9). This step was introduced to contrast the incidence at old ages with the clinically observed overall incidence of the order of $10^{-5}$, which we weighted by assuming an error of $10^{-4}$.

We computed the cost function, $d$, as

$$d = \sum_{i \in m_{MRCA,ev}} w_i \left( \frac{(I_{MRCA,sim,i} - I_{MRCA,ev,i})^2}{\Delta I_{MRCA,ev,i}^2} \right) + \frac{(I_{ECA,sim,i} - I_{ECA,ev,i})^2}{\Delta I_{ECA,ev,i}^2} + \frac{I_{MRCA,sim,ten years} - 10^{-5}}{(10^{-4})^2}. \quad (18)$$

To enforce good fits to the initial phase of the incidence curve for better comparison of the contraction and homeostasis models of neural precursor dynamics, we chose weights

$$w_i = \begin{cases} 10, & \text{if } \tilde{m}_{MRCA} \leq 0.2/10^6 \\ 1, & \text{if } \tilde{m}_{MRCA} > 0.2/10^6 \end{cases}.$$

Ninety-five per cent posterior probability bounds for the model fits were estimated by simulating the model at each sampled parameter set and cutting off 2.5% at each end of the simulated distribution.

To assess the robustness of the model, we performed an additional parameter estimation on clock-like mutations only. To this end, we multiplied mutation densities at MRCA and ECA by the fraction of mutations generated by clock-like mutational signatures (SBS1, SBS5 and SBS40) and fit the model to the clock-like mutation densities.

## Modeling mutation accumulation during tumor growth

**Model.** Denoting the rate at which tumor cells divide by $\lambda_T$ and the loss rate by $\delta_T$, we modeled the number of tumor cells, $N_T$, with an exponential growth model, $N_T(t) = e^{(\lambda_T - \delta_T)t}$. We assumed that some SSNVs are already present in the founder cell of the tumor and denote their number by $n_{clonal}$; these mutations are clonally propagated to the entire tumor and will thus be found at frequency $f = 1$. Additional SSNVs are acquired during tumor growth, and we denote their number by $n_{subclonal}$; these mutations are present in a subset of the tumor only and will thus be found at $f < 1$. The VAF distribution of a neuroblastoma is hence a superposition of clonal and subclonal mutations accumulated before and during tumor growth, respectively.

To model the number of subclonal mutations, we used a model of neutrally evolving tumors that accounts for the stochastic expansion of neutral subclones while assuming exponential expansion of the tumor mass overall[57]. This model assumes that neutral mutations are acquired at all times at rate $\mu\lambda_T N_T(t)$ and drift stochastically according to a supercritical birth–death process, where[58]:

$$P_{1,i}(\lambda_T, \delta_T, t) = \begin{cases} \alpha(t), i = 0 \\ (1 - \alpha(t))(1 - \beta(t))\beta(t)^{i-1}, i \geq 1 \end{cases} \quad (19)$$

where:

$$\alpha(t) = \frac{\delta_T\left(e^{(\lambda_T - \delta_T)t} - 1\right)}{\lambda_T e^{(\lambda_T - \delta_T)t} - \delta_T}, \beta(t) = \frac{\lambda_T\left(e^{(\lambda_T - \delta_T)t} - 1\right)}{\lambda_T e^{(\lambda_T - \delta_T)t} - \delta_T}.$$

Together, this yields the site frequency spectrum, $S(i, \mu)$, of subclonal mutations:

$$S(i, \mu) = \int_0^{t_{end}} P_{1,i}(\lambda_T, \delta_T, t_{end} - t)\mu\lambda_T N_T(t)\, dt. \quad (20)$$

The total number of subclonal mutations in a tumor is computed as;

$$n_{subclonal} = \sum_{i=1}^{N_T(t_{end})} S(i, \mu) = \sum_{i=1}^{N_T(t_{end})} \int_0^{t_{end}} P_{1,i}(\lambda_T, \delta_T, t_{end} - t)\mu\lambda_T N_T(t)\, dt, \quad (21)$$

where $N_T(t_{end})$ is the number of tumor cells at diagnosis. The number of mutations present in subclones of at least $a$ and at most $b$ cells is, in analogy to equation (21):

$$\sum_{i=a}^{b} S(i, \mu) = \sum_{i=a}^{b} \int_0^{t_{end}} P_{1,i}(\lambda_T, \delta_T, t_{end} - t)\mu\lambda_T N_T(t)\, dt. \quad (22)$$

Because clone sizes $a$ and $b$ are large, we approximate the sum in equation (22) by an integral, yielding:

$$\sum_{i=a}^{b} S(i,\mu) \approx \int_{a}^{b} \int_{0}^{t_{end}} P_{1,i}(\lambda_T, \delta_T, t_{end}-t)\mu\lambda_T N_T(t)\,dt\,di$$

$$= \int_{0}^{t_{end}} \mu\lambda_T N_T(t)\, \frac{P_{1,b}(\lambda_T, \delta_T, t_{end}-t) - P_{1,a}(\lambda_T, \delta_T, t_{end}-t)}{\log\beta(t_{end}-t)}\,dt. \qquad (23)$$

**Relating model and data.** To relate the model of mutation accumulation to the measured VAF distribution, we modeled mutations on each copy number state separately. To this end, we denote by $n_{f,k}$ the number of mutations at frequency $f$ on genomic segments with copy number state $k$. Note that $f$ reports the fraction of mutated cells among all tumor cells whereas VAFs report the fraction of mutated alleles in the tissue sample. The two quantities can be converted into each other using the following relation:

$$\text{VAF} = \frac{f\rho}{\zeta}. \qquad (24)$$

where $\zeta$ is the average copy number of the sample as defined in equation (1). Clonal mutations are associated with frequency $f = 1$. The number of clonal mutations falling within genomic segments of copy number $k$, $n_{1,k}$, is expected to scale with $g_k$, the fraction of the genome at copy number $k$, according to

$$n_{1,k} = n_{clonal}\frac{kg_k}{\sum_k kg_k}. \qquad (25)$$

To relate true with measured VAFs, we assumed that the latter are binomially distributed around the former according to

$$B\left(\frac{f\rho}{\zeta}, C_k\right),$$

where $C_k \propto \text{Pois}(\widehat{C_k})$ is the sequencing coverage on a segment with copy number $k$, and $\rho$ is tumor cell content.

**Data selection.** Because we model mutation accumulation with neutral tumor expansion, we included all tumors with well-defined subclonal tails and absence of subclonal selection. To this end, we visually inspected the VAF histograms and excluded 29 cases with poorly resolved subclonal tails. To identify subclonal selection we ran Mobster[12], excluding mutations on sex chromosomes and computing pseudoheterozygous VAFs, in the following termed $\widetilde{\text{VAF}}$ and defined as 50% of the mutated sample fraction, SF. The measured VAFs relate to SF according to $\text{VAF} = \frac{k}{\zeta}\text{SF}$, where $k$ is the number of alleles carrying the mutation. It follows that $\widetilde{\text{VAF}} = \frac{\zeta}{2k}\text{VAF}$. We excluded mutations with $\widetilde{\text{VAF}} < 0.1$ from the fit and ran the Mobster setting autosetup = "FAST". This resulted in an additional exclusion of nine cases for which Mobster suggested subclonal selection. Thus we selected 62 tumors with well-resolved subclonal VAF histograms and no evidence of subclonal selection for parameter inference (Supplementary Table 1).

**Parameter estimation.** We estimated $n_{clonal}$, $\mu$ and $\delta_T/\lambda_T$ using ABC–SMC as implemented in pyABC[56]. We used a population size of 1,000 parameter samples and prior distributions as outlined in Supplementary Table 12. Termination criteria were the same as above (Modeling neuroblastoma initiation).

We stratified measured VAFs by copy number, excluding copy numbers >4 or present on, in total, <$10^8$ bp. For each copy number $k$, we merged mutations from the clonal VAF peaks constituted by amplified and non-amplified mutations (see Mutation timing for the definition of amplified and non-amplified clonal mutations). To this end, we first assessed the average coverage $\widehat{C_k}$ from all mutations falling on segments with copy number $k$. Then, we classified mutations at copy number $k$ as amplified clonal if $\frac{Q_{l-1}^{0.95}}{\widehat{C_k}} < \text{VAF}_k \le \frac{Q_l^{0.95}}{\widehat{C_k}}$, where $Q_l^{0.95}$ is the 95%

quantile of a binomial distribution with success probability $\frac{\rho l}{\zeta}$ and where $l$ is the B-allele count. These mutations were then merged with those of the non-amplified clonal peak by multiplying their frequencies by $\frac{l}{k}$ and adding them $l$ times.

Finally, we computed the cumulative mutation counts of the measured data, $F_{k,ev}(f) = \sum\text{VAF}_k > f$, where $f$ runs from 0.05 to 1.00 in steps of size 0.05, and extrapolated the cumulative mutation counts to the whole genome with multiplication by $\frac{\sum_k g_k}{g_k}$. At a sampled parameter set for $n_{clonal}$, $\mu$ and $\delta_T$, the following steps were performed for each copy number state $k$ included in the analysis:

(1) Sample for each clonal mutation a sequencing coverage $C_k$ according to $\text{Pois}(\widehat{C_k})$.
(2) Sample for each clonal mutation a VAF according to equation (16).
(3) Determine $n_{f,k;f\ne1}$ from equation (13), assuming a tumor size of $10^9$ cells at diagnosis, and evaluating equation (13) in bins of size 0.05 at the lower limit:

$$n_{f,k;f\ne1} \approx \sum_{i=f10^9}^{(f+0.05)10^9} S(i,\mu).$$

(4) Sample for each subclonal mutation a sequencing coverage $C_k$ according to $\text{Pois}(\widehat{C_k})$ and a VAF according to equation (16).
(5) Compute the cumulative mutation counts, $F_{k,sim}(f) = \sum\text{VAF}_k > f$, where $f$ runs from 0.05 to 1.00 in steps of size 0.05.

The cost function is

$$d = \sum_k \sum_f (F_{k,sim} - F_{k,ev})^2 \frac{g_k}{\sum_{k'} g_{k'}}. \qquad (26)$$

## Real-time estimation of cell division rate

In the previous sections we describe how we inferred the parameters for our models of neuroblastoma initiation and growth from mutation data. This yielded estimates for the rates at which cells are lost, as well as the rates at which neutral and oncogenic mutations are acquired in units of cell divisions. To convert these estimates to real time we estimated the actual cell division rate, using the age distribution at diagnosis and the approximate tumor size at diagnosis.

We reasoned that the time span between gastrulation and tumor diagnosis ($t_D$) consists of two phases: premalignancy up to the formation of the MRCA and malignant growth of the tumor thereafter. Hence,

$$t_D = \frac{\tilde{m}_{MRCA}}{\lambda\mu} + \frac{\log N_T(t_D)}{\lambda\left(1 - \frac{\delta}{s\lambda}\right)}, \qquad (27)$$

where we assume exponential tumor growth until diagnosis. Thus the cell division rate, $\lambda$, can be expressed as

$$\lambda = \frac{1}{t_D}\left(\frac{\tilde{m}_{MRCA}}{\mu} + \frac{\log N_T(t_D)}{1 - \frac{\delta}{s\lambda}}\right). \qquad (28)$$

To estimate $\lambda$ with equation (28), we combined the clinical information with parameter estimates from our models as follows: we know $t_D$ (the age at diagnosis, $A$, plus approximately 250 days of embryogenesis after gastrulation) from the clinical data, and estimate that a tumor of a few cubic centimeters consists of the order of $N_T(t_D) = 10^9$ cells. As described below (Mutation timing and Modeling neuroblastoma initiation), we also have estimates for mutation density at the MRCA, $\tilde{m}_{MRCA}$, and for the SSNV rate per actual cell division and haploid genome, $\mu$. Finally, we obtained an estimate for the effective rate of acquisition of

(neutral) SSNVs during tumor growth from the subclonal VAF histograms of 62 tumors (compare with Modeling mutation accumulation during tumor growth), which is related to $\delta$ and $s$ as

$$\mu_{\mathrm{eff}} = \frac{\mu}{1 - \delta_{\mathrm{T}}/\lambda_{\mathrm{T}}} = \frac{\mu}{1 - \frac{\delta}{s\lambda}}. \tag{29}$$

Substituting equation (29) in equation (28), we obtain for each of the 62 tumors (labeled with the index $i$) an estimate for the division rate with mean, $\langle \lambda_i \rangle = \frac{1}{2\langle\mu\rangle(A_i+250\,\text{days})} (\langle 2\tilde{m}_{\mathrm{MRCA,i}} \rangle + \log 10^9 \langle \mu_{\mathrm{eff,i}} \rangle)$, and s.d. (standard deviation), $\sigma(\lambda_i) = \frac{1}{2\langle\mu\rangle(A_i+250\,\text{days})} \left( \frac{2\langle\tilde{m}_{\mathrm{MRCA,i}}\rangle + \log 10^9 \langle \mu_{\mathrm{eff,i}}\rangle}{2\mu} \sigma \langle 2\mu \rangle + \sigma(2\tilde{m}_{\mathrm{MRCA,i}}) + \sigma(\mu_{\mathrm{eff,i}}) \right)$, in actual time. Note that factor 2 accounts for the fact that $\mu$ and $\tilde{m}_{\mathrm{MRCA,i}}$ measure mutation rate and density, respectively, per haploid genome. From equation (29), we also get an estimate of $\delta_{\mathrm{T,i}}/\lambda_{\mathrm{T,i}}$ with expectation $\left\langle \frac{\delta_{\mathrm{T,i}}}{\lambda_{\mathrm{T,i}}} \right\rangle = 1 - \frac{2\langle\mu\rangle}{\langle\mu_{\mathrm{eff,i}}\rangle}$ and s.d. $\sigma(\delta_{\mathrm{T,i}}/\lambda_{\mathrm{T,i}}) = \frac{\sigma\langle 2\mu\rangle}{\langle\mu_{\mathrm{eff,i}}\rangle} + \frac{2\langle\mu\rangle}{(\langle\mu_{\mathrm{eff,i}}\rangle)^2} \sigma(\mu_{\mathrm{eff,i}})$. Fitting effective mutation rates to the VAF histograms of each of the selected 62 tumors individually hence yields tumor-specific division rates along with relative death rates, which we stratify by TMM.

Finally, one also obtains an estimate for the daily mutation rate during tumor initiation by computing $\mu\lambda_{\mathrm{T,i}}$ with associated uncertainty $\mu\Delta\lambda_{\mathrm{T,i}} + \lambda_{\mathrm{T,i}}\sigma(\mu)$, which relates molecular clock to real time. For this purpose, we average across the inferences from the 44 primary tumors/metastasis, excluding 18 relapsed tumors among the 62 submitted for analysis.

## Statistics and reproducibility
All statistical tests were computed with R (v.3.6.0 and v.4.0.0), and details (statistical tests and whether one- or two-sided, exact sample size, $P$ values and test statistics) are specified in the respective figures and accompanying Source data. This is a retrospective analysis of tumor material that was collected as part of the diagnostic workflow of the German Neuroblastoma trial by the Society for Pediatric Oncology and Hematology and collected in the Neuroblastoma tumor bank. Hence, no statistical method was used to predetermine sample size. All analyzed tumors had a clear copy number profile, tumor cell content ≥25% and no hypermutation genotype; no data were excluded from the analyses. The experiments were not randomized. The investigators were not blinded to allocation during experiments and outcome assessment.

## Software and packages
Analysis was performed with R (v.3.6.0 and v.4.0.0) and python v.3.6.1. We used the following R packages: openxlsx v.4.1.5, ggsignif v.0.6.0, ggbeeswarm v.0.6.0, gridExtra v.2.3, RColorBrewer v.1.1-2, HDInterval v.0.2.2, cdata v.1.1.8, moments v.0.14, Hmisc v.4.4-0, scales v.1.1.1, bedr v.1.0.7, circlize v.0.4.10, ggplot2 v.3.3.2 (ref. [59]), Bioconductor v.3.15 (ref. [60]), ggbio v.1.34.0 (ref. [61]), ggpubr v.0.4.0, pammtools v.0.2.2 (ref. [62]), ComplexHeatmap v.2.5.1 (ref. [63]), BSgenome.Hsapiens.UCSC. hg19 v.1.4.3, MASS v.7.3-51.6 (ref. [64]), GenomicRanges v.1.38.0 (ref. [65]), reshape2 v.1.4.4 (ref. [66]), mixtools v.1.2.0 (ref. [67]), dplyr v.1.0.0, survminer v.0.4.8, survival[68] v.3.1-12, wesanderson v.0.3.6, cowplot v.1.1.1, mobster[12] v.1.0.0, CNAqc v.1.0.0 and mmsig[51] v.0.0.0.9000; and the python packages SigProfilerMatrixGenerator[69] v.1.1.26, SigProfilerExtractor[50] v.1.1.1 and pyABC[56] v.0.9.13.

## Reporting summary
Further information on research design is available in the Nature Portfolio Reporting Summary linked to this article.

## Data availability
Subsets of WGS and RNA sequencing data were part of previously published studies[24,26]. Data from these studies are deposited at the European Genome-Phenome Archive (https://www.ebi.ac.uk/ega/) under accession nos. EGAS00001004349 and EGAS00001001308. Additional WGS data generated for this study are available at the European Genome-Phenome Archive under accession nos. EGAS00001004990 and EGAS00001006533. In accordance with the laws of data protection, data are deposited under controlled access. Access can be granted by contacting Frank Westermann (f.westermann@kitz-heidelberg.de) and requires a data access agreement; requests will be replied to within 4 weeks. Variant calls (SNVs, indels, SVs and copy number variations), mutational signatures, model fits and a summary of the mutation profile and modeling results for each tumor can be accessed at Mendeley (https://doi.org/10.17632/m9pwjbm7c8.1)[70]. All remaining data are available in the Supplementary information. Source data are provided with this paper.

## Code availability
All code developed for this study is available at https://github.com/hoefer-lab/Neuroblastoma_evolution[71].

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

## Acknowledgements

We thank the members of the Höfer and Westermann groups for discussions. T.H. and F.W. are supported through ERA-CoSysmed consortium INFER-NB (Federal Ministry for Education and Research, grant agreement no. 031L0238) and DKFZ core funding. We thank the NCT Molecular Precision Oncology Program for technical support and funding through HIPO2-K09R.

## Author contributions

V.K., F.W. and T.H. conceived the project. S.A.S., K.-O.H., B.H. and F.W. provided primary sequencing data and integrated these with clinical information. V.K. analyzed the data, developed mathematical models and performed all computations. B.B. and R.K. contributed to survival analysis. T.H. supervised the study. T.H. and V.K. wrote the manuscript, with input from all authors.

## Funding

## Competing interests

The authors declare no competing interests.

## Additional information

**Extended data** is available for this paper at https://doi.org/10.1038/s41588-023-01332-y.

**Correspondence and requests for materials** should be addressed to Frank Westermann or Thomas Höfer.

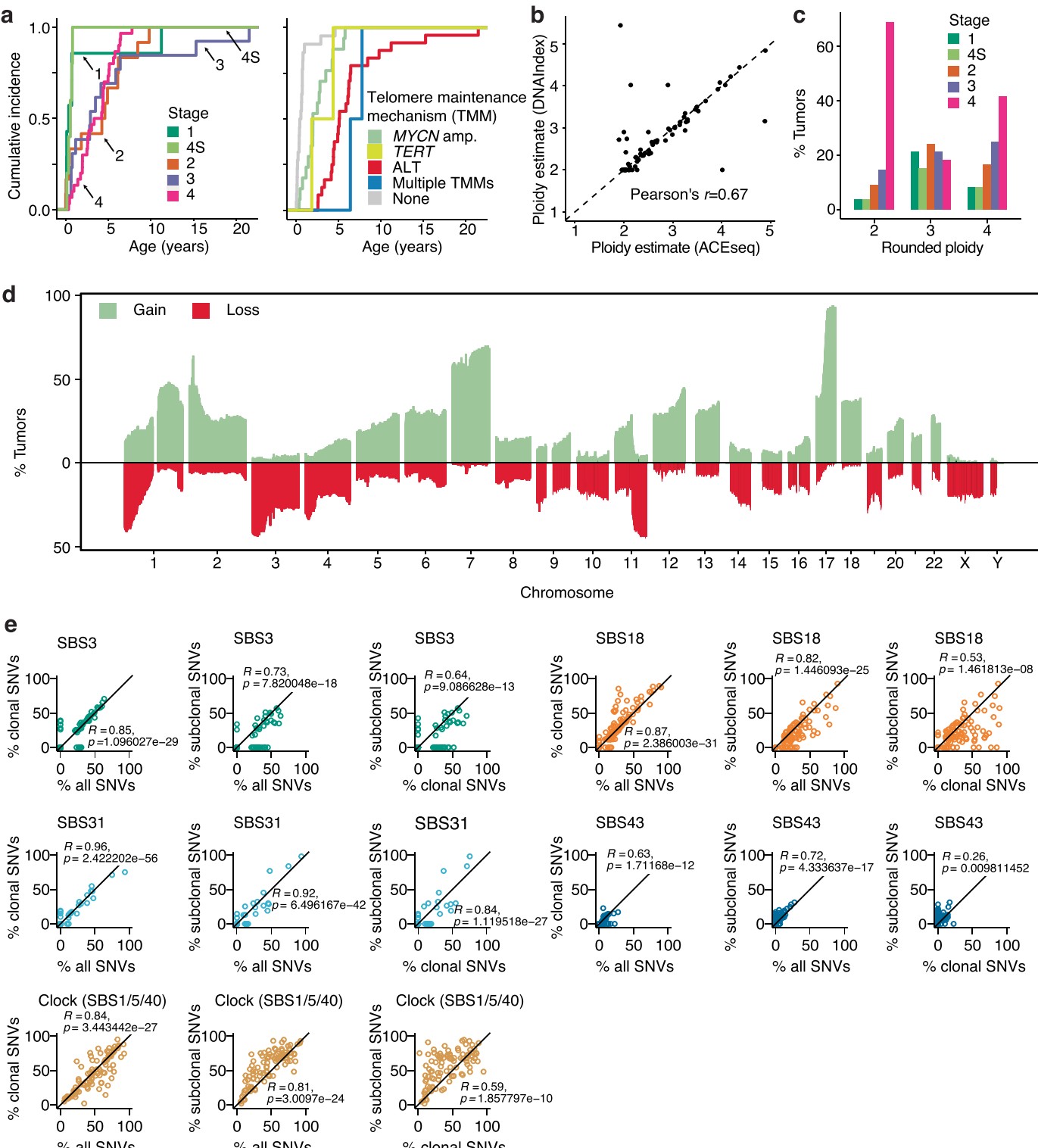

**Extended Data Fig. 1 | Copy number changes and mutational signatures in primary and relapsed neuroblastomas of the discovery cohort. a**, Age distribution at tumor diagnosis. **b**, Ploidy estimates based on WGS (inferred with ACEseq) and on DNA-index measurements (as measured by flow cytometry). Shown are 71 primary and relapsed tumors for which DNA-index was determined. **c**, Distribution of neuroblastoma stages among rounded ploidies. **d**, Overview of gains and losses across the tumor cohort. **e**, Comparison of mutational signatures (Cosmic v3) contributing to all SSNVs and to clonal and subclonal SSNVs (*n* = 100 primary and relapse tumors from the discovery cohort; Pearson's correlation coefficients and two-sided *P* values are shown for each comparison).

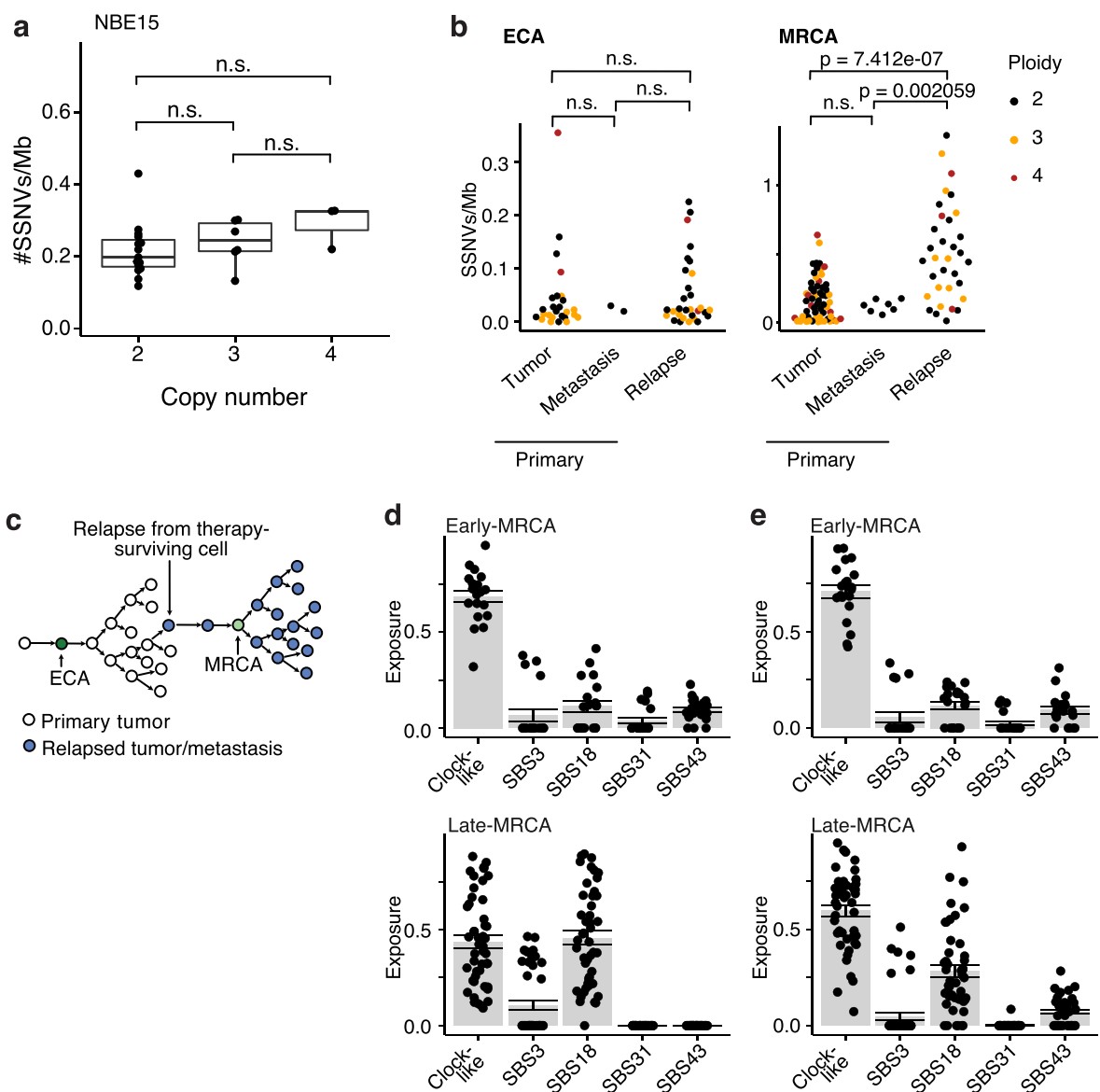

**Extended Data Fig. 2 | Mutation densities at chromosomal gains in the discovery cohort. a**, Mutation densities of non-amplified clonal mutations per genomic segment, stratified by copy number for tumor NBE15. Significance was tested with a two-sided Wilcoxon rank sum test/Mann–Whitney $U$-test (disomic: $n = 15$; trisomic: $n = 6$; tetrasomic: $n = 3$). Boxes show median, 25% and 75% percentiles, whiskers extend to the smallest and largest value within 1.5x interquartile range. **b**, Estimated mutation densities at ECA and MRCA of primary tumors (ECA: $n = 26$; MRCA: $n = 60$), primary metastases (ECA: $n = 2$; MRCA: $n = 7$) and relapsed tumors/metastases (ECA: $n = 30$; MRCA: $n = 33$). Significance

was tested with a two-sided Wilcoxon rank sum test/Mann–Whitney $U$-test and defining ** as $P < 0.01$ (exact $P$ values are given in Source Data). **c**, Model scheme for neuroblastoma relapse corresponding to data in (**b**). **d** and **e**, Exposures of mutational signatures among clonal (**d**) and subclonal SSNVs (**e**). Signatures SBS1, SBS5 and SBS40 were combined into a single clock-like mutational signature. Bar heights correspond to the average among early-MRCA (top; $n = 20$ primary tumors and metastases) and late-MRCA tumors (bottom; $n = 47$ primary tumors and metastases); error bars show standard error of the mean.

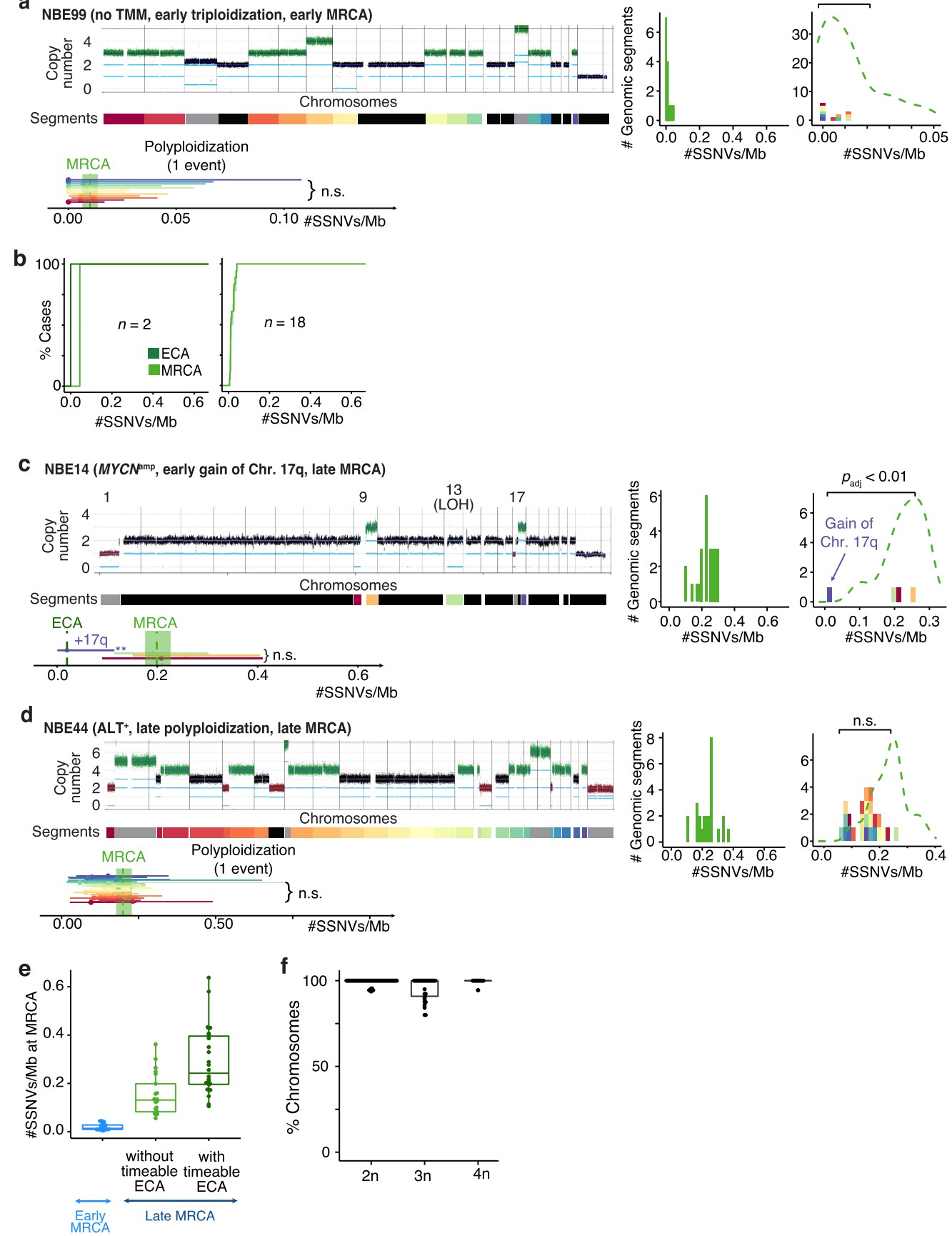

**Extended Data Fig. 3 | See next page for caption.**

**Extended Data Fig. 3 | Examples of early evolution. a**, Copy number profiles of an example tumor with near-triploid genome and without a timeable ECA. The color panels annotate genomic segments of equal copy number. Next to the copy number profile, the densities of non-amplified (green) and amplified (color-encoded) clonal mutations on segment of length $\geq 10^7$ bp are shown, with the dashed line showing the kernel-density estimate of the distribution of non-amplified clonal mutations. Chromosomal segments of equal copy number were merged and Holm-corrected one-sided $P$ values were computed based on a negative binomial distribution (**, $p_{adj} < 0.01$, exact $P$ values are provided in Source Data). The bottom panel shows mutation densities at ECA and MRCA with 95% confidence bounds estimated by bootstrapping. The horizontal lines show mutation densities on gained segments with 95% confidence bounds computed from $\chi^2$ distributions. **b**, Mutation densities at ECA and MRCA of early-MRCA tumors with (left) and without (right) timeable ECA. Solid lines represent maximum likelihood estimates and shaded areas represent 95% confidence intervals, as obtained by bootstrapping. **c** and **d**, As in (**a**) but for a near-diploid tumor with timeable ECA (**c**) and for a near-tetraploid tumor without timeable ECA (**d**). Holm-corrected one-sided $P$ values were computed based on a negative binomial distribution (**, $p_{adj} < 0.01$, exact $P$ values are provided in Source Data). **e**, Mutation densities at MRCA in early-MRCA ($n = 20$) and late-MRCA primary tumors ($n = 47$, thereof $n = 26$ with ECA and $n = 21$ without ECA). **f**, Fraction of polysomic chromosomes in near-triploid ($n = 33$) and near-tetraploid ($n = 12$) tumors that were generated in a single oncogenic event. As a control, the fraction of disomic chromosome whose mutation density matched with the MRCA is shown for near-diploid tumors ($n = 55$). In (**e**) and (**f**), boxes show median, 25 and 75% percentiles, whiskers extend to the smallest and largest value within 1.5x interquartile range.

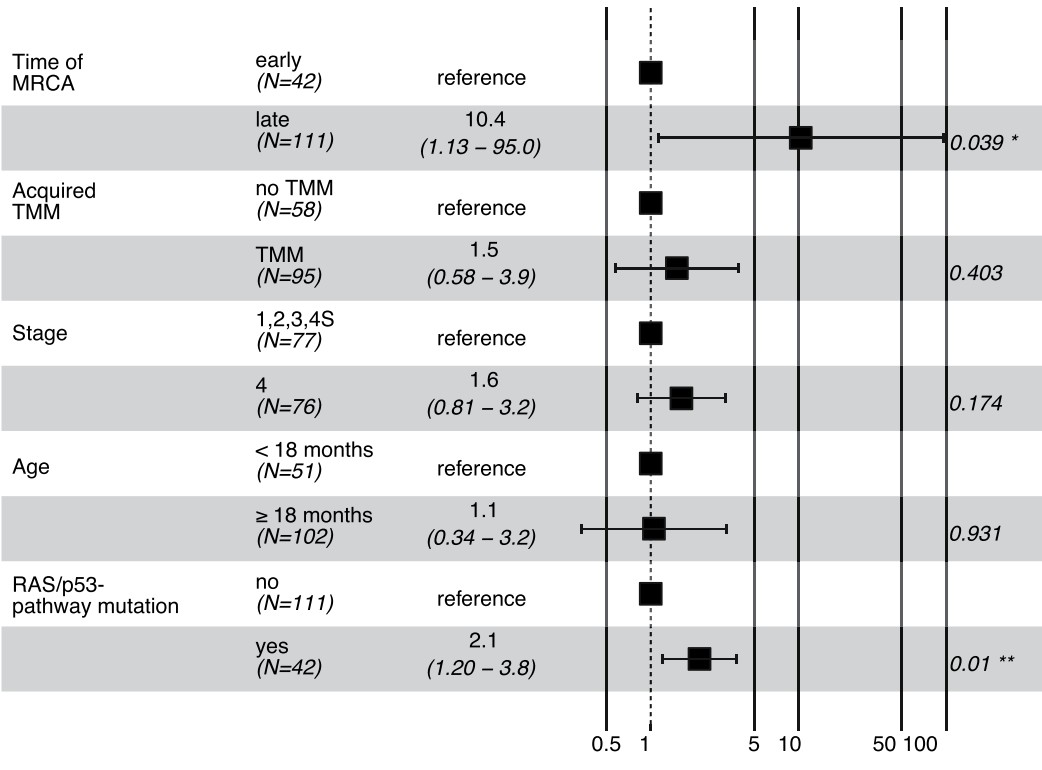

**Extended Data Fig. 4 | Analysis of overall survival.** Multivariate analysis of overall survival with Cox-regression considering MRCA timing, acquired mechanisms of telomere maintenance, stage, age at diagnosis and mutation status in the RAS/P53 pathway. Shown are mean hazard ratio, 95% confidence intervals and $p$-values for each variable (two-sided Wald test).

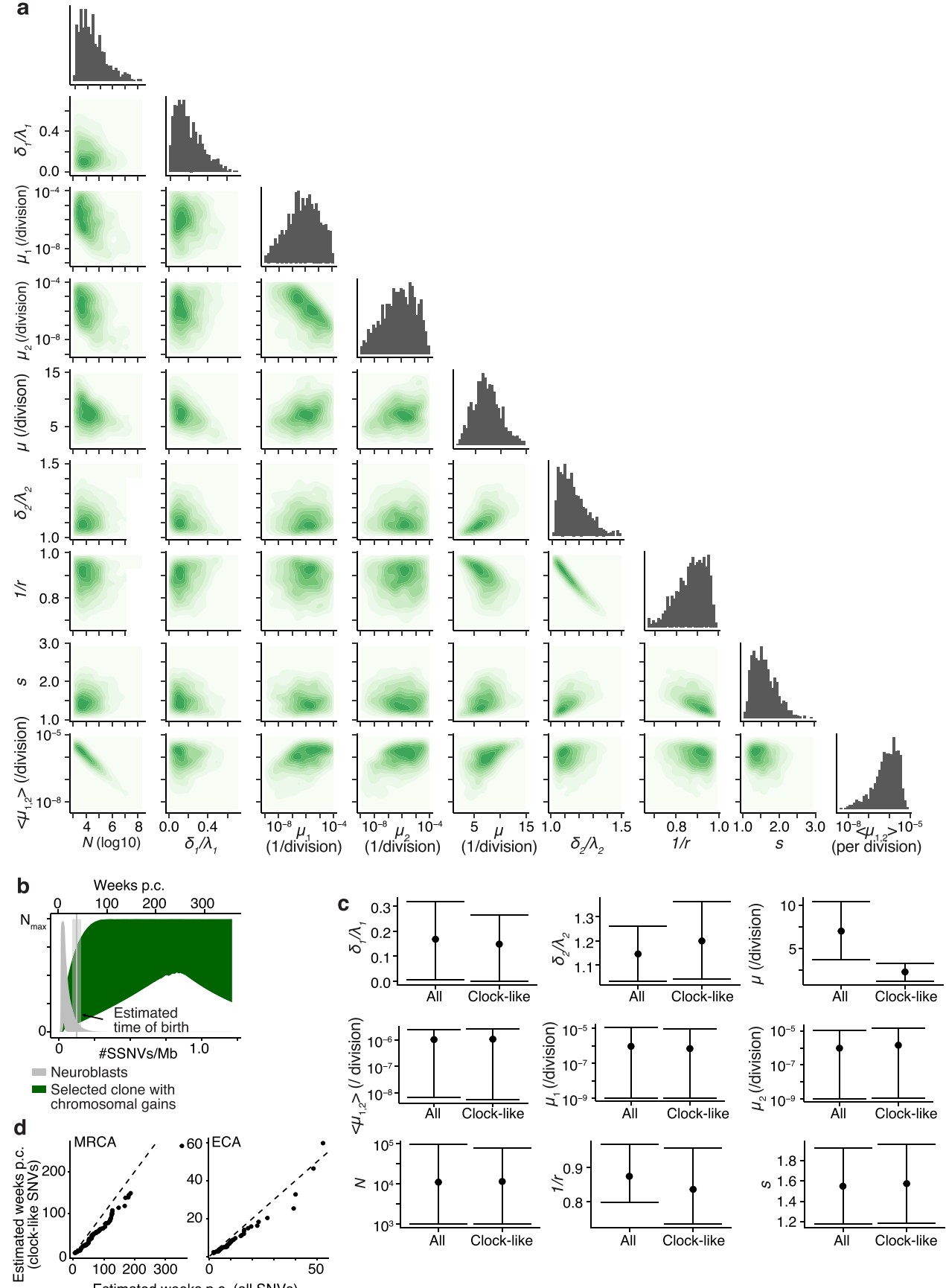

**Extended Data Fig. 5 | See next page for caption.**

**Extended Data Fig. 5 | Neuroblastoma initiation during progenitor expansion. a**, Two-dimensional projections of the posterior probability distribution of the model parameters (considering neuroblastoma initiation in a transient population of early neuroblasts). **b**, Predicted expansion of neuroblasts (grey) and the selected subclone upon acquisition of the first oncogenic event (dark green) in a model of transient neuroblast expansion. Colored areas show the 95% posterior probability bounds (estimated from simulations using 1,000 samples from the posterior probability distribution). Vertical line and shaded area give mean and 95% CI for the estimated time of birth (computed from $n = 62$ primary tumors). **c**, Comparison of parameter estimates if fitting the model to all SSNVs or to SSNVs generated by a clock-like mutational process only. For the latter, mutant densities were adjusted by the fraction of SSNVs explained by SBS1, SBS5 or SBS40. For each parameter, median and 80% credible intervals are shown (estimated from $n = 1000$ samples of the posterior distribution). **d**, Comparison between the estimated time point at which the MRCA (left, $n = 95$ primary tumors/metastases with TMM) and the ECA (right, $n = 47$ primary tumors/metastases with TMM) emerged if using all SSNVs or if using SSNVs that were generated by a clock-like process only. Time is measured in weeks post conceptionem (p.c.).

# Reporting Summary

## Statistics

For all statistical analyses, confirm that the following items are present in the figure legend, table legend, main text, or Methods section.

| n/a | Confirmed | |
|---|---|---|
| ☐ | ☒ | The exact sample size (*n*) for each experimental group/condition, given as a discrete number and unit of measurement |
| ☒ | ☐ | A statement on whether measurements were taken from distinct samples or whether the same sample was measured repeatedly |
| ☐ | ☒ | The statistical test(s) used AND whether they are one- or two-sided *Only common tests should be described solely by name; describe more complex techniques in the Methods section.* |
| ☐ | ☒ | A description of all covariates tested |
| ☐ | ☒ | A description of any assumptions or corrections, such as tests of normality and adjustment for multiple comparisons |
| ☐ | ☒ | A full description of the statistical parameters including central tendency (e.g. means) or other basic estimates (e.g. regression coefficient) AND variation (e.g. standard deviation) or associated estimates of uncertainty (e.g. confidence intervals) |
| ☐ | ☒ | For null hypothesis testing, the test statistic (e.g. $F$, $t$, $r$) with confidence intervals, effect sizes, degrees of freedom and $P$ value noted *Give P values as exact values whenever suitable.* |
| ☐ | ☒ | For Bayesian analysis, information on the choice of priors and Markov chain Monte Carlo settings |
| ☒ | ☐ | For hierarchical and complex designs, identification of the appropriate level for tests and full reporting of outcomes |
| ☐ | ☒ | Estimates of effect sizes (e.g. Cohen's *d*, Pearson's *r*), indicating how they were calculated |

*Our web collection on statistics for biologists contains articles on many of the points above.*

## Software and code

Policy information about availability of computer code

| Data collection | No software was used for data collection. |
|---|---|
| Data analysis | Analysis was performed with R (v3.6.0 and v4.0.0) and python v 3.6.1.<br><br>R packages:<br>openxlsx v4.1.5<br>ggsignif v0.6.0<br>ggbeeswarm v0.6.0<br>RColorBrewer v1.1-2<br>HDInterval v0.2.2<br>cdata v1.1.8<br>moments v0.14<br>Hmisc v4.4-0<br>scales v1.1.1<br>bedr v1.0.7<br>ggplot2 v3.3.2<br>ggbio v1.34.0<br>ggpubr v0.4.0<br>gridExtra v2.3<br>circlize v0.4.10<br>pammtools v0.2.2 |

ComplexHeatmap v2.5.1
BSgenome.Hsapiens.UCSC.hg19 v1.4.3
MASS v7.3-51.6
GenomicRanges v1.38.0
Bioonductor v3.15
reshape2 v1.4.4
mixtools v1.2.0
dplyr v1.0.0
survminer v0.4.8
survival v3.1-12
wesanderson v0.3.6
cowplot v1.1.1
mobster v1.0.0
CNAqc v1.0.0
mmsig v0.0.0.9000

python packages:
SigProfilerMatrixGenerator v1.1.26
SigProfilerExtractor v1.1.1
pyABC v0.9.13

Custom code is available on github (https://github.com/hoefer-lab).

For manuscripts utilizing custom algorithms or software that are central to the research but not yet described in published literature, software must be made available to editors and reviewers. We strongly encourage code deposition in a community repository (e.g. GitHub). See the Nature Portfolio guidelines for submitting code & software for further information.

# Data

Policy information about availability of data

All manuscripts must include a data availability statement. This statement should provide the following information, where applicable:
- Accession codes, unique identifiers, or web links for publicly available datasets
- A description of any restrictions on data availability
- For clinical datasets or third party data, please ensure that the statement adheres to our policy

Subsets of the whole genome sequencing and RNA sequencing data were part of previously published studies24,26. Data of these studies is deposited at the European Genome-Phenome Archive (https://www.ebi.ac.uk/ega/) under accession numbers EGAS00001004349 and EGAS00001001308. Additional whole genome sequencing data generated for this study is available at the European Genome-phenome Archive under the accession numbers EGAS00001004990 and EGAS00001006533. In accordance with the laws of data protection, data is deposited under controlled access. Access can be granted by contacting Frank Westermann (f.westermann@kitz-heidelberg.de) and requires a data access agreement. Requests will be replied to within four weeks. Variant calls (SNVs, Indels, SVs and CNVs), mutational signatures, model fits and a summary on the mutation profile and the modeling results for each tumor can be accessed on Mendeley (http://dx.doi.org/10.17632/m9pwjbm7c8.1)70. All remaining data is available in the Supplementary information.

# Human research participants

Policy information about studies involving human research participants and Sex and Gender in Research.

| Reporting on sex and gender | Patients' sex is reported in Supplementary Table 1. |
| --- | --- |
| Population characteristics | See Supplementary Table 1 for age at diagnosis and survival information. |
| Recruitment | Cohorts of primary and relapsed neuroblastoma tumors were retrospectively analyzed. Tumor material was collected as part of the diagnostic workflow of the German Neuroblastoma trial by the Society for Pediatric Oncology and Hematology (GPOH) and collected in the Neuroblastoma tumor bank. |
| Ethics oversight | All trials were approved by the Ethics Committee of the Medical Faculty, University of Cologne and collection and use of all tumor tissue material was approved (NB97, NB2004, NB2016 registry). All patients or their parents signed an informed consent. |

Note that full information on the approval of the study protocol must also be provided in the manuscript.

# Field-specific reporting

Please select the one below that is the best fit for your research. If you are not sure, read the appropriate sections before making your selection.

☒ Life sciences          ☐ Behavioural & social sciences          ☐ Ecological, evolutionary & environmental sciences

For a reference copy of the document with all sections, see nature.com/documents/nr-reporting-summary-flat.pdf

# Life sciences study design

All studies must disclose on these points even when the disclosure is negative.

| | |
|---|---|
| Sample size | A unique cohort of ultra-deep WG sequenced neuroblastomas was analyzed, primarily limited by availability of appropriate samples. Available sample size allowed meaningful statistical analysis. |
| Data exclusions | No data were excluded from the analyses |
| Replication | Individual tumor samples were analyzed and compared by statistical analysis. The analysis was validated in an independent cohort. |
| Randomization | n/a. This is a retrospective analysis of tumor material that was collected as part of the diagnostic workflow of the German Neuroblastoma trial by the Society for Pediatric Oncology and Hematology (GPOH) and collected in the Neuroblastoma tumor bank. No case-control design was involved. |
| Blinding | n/a. This is a retrospective analysis of tumor material that was collected as part of the diagnostic workflow of the German Neuroblastoma trial by the Society for Pediatric Oncology and Hematology (GPOH) and collected in the Neuroblastoma tumor bank. No case-control design was involved. |

# Reporting for specific materials, systems and methods

We require information from authors about some types of materials, experimental systems and methods used in many studies. Here, indicate whether each material, system or method listed is relevant to your study. If you are not sure if a list item applies to your research, read the appropriate section before selecting a response.

## Materials & experimental systems

| n/a | Involved in the study |
|---|---|
| ☒ | ☐ Antibodies |
| ☒ | ☐ Eukaryotic cell lines |
| ☒ | ☐ Palaeontology and archaeology |
| ☒ | ☐ Animals and other organisms |
| ☐ | ☒ Clinical data |
| ☒ | ☐ Dual use research of concern |

## Methods

| n/a | Involved in the study |
|---|---|
| ☒ | ☐ ChIP-seq |
| ☒ | ☐ Flow cytometry |
| ☒ | ☐ MRI-based neuroimaging |

## Clinical data

Policy information about clinical studies

All manuscripts should comply with the ICMJE guidelines for publication of clinical research and a completed CONSORT checklist must be included with all submissions.

| | |
|---|---|
| Clinical trial registration | This is a retrospective analysis of tumor material from patients being enrolled in the neuroblastoma trials of the GPOH and/or INFORM registry trial. |
| Study protocol | n/a. This is a retrospective analysis of tumor material from patients being enrolled in the neuroblastoma trials of the GPOH and/or INFORM registry trial. |
| Data collection | n/a. This is a retrospective analysis of tumor material from patients being enrolled in the neuroblastoma trials of the GPOH and/or INFORM registry trial. |
| Outcomes | n/a. This is a retrospective analysis of tumor material from patients being enrolled in the neuroblastoma trials of the GPOH and/or INFORM registry trial. |

