## [Peer review file. · Nature Genetics]

Peer Review Information

Manuscript Title: Neuroblastoma arises in early fetal development and its evolutionary duration predicts outcome

Corresponding author name(s): Dr Frank Westermann, Professor Thomas Höfer

Editorial Notes:

Redactions – unpublished data Parts of this Peer Review File have been redacted as indicated to maintain the confidentiality of unpublished data.

Reviewer Comments & Decisions:

Decision Letter, initial version:
--

15th Aug 2022

Dear Thomas,

Your Article, "Genetic origin of neuroblastoma in early fetal development" has now been seen by 3 referees. You will see from their comments below that while they find your work of interest, some important points are raised. We are interested in the possibility of publishing your study in Nature Genetics, but would like to consider your response to these concerns in the form of a revised manuscript before we make a final decision on publication.

In brief, the three reviewers sound positive and supportive for your work at this stage.

Reviewer #1, in line with their expertise, finds the most interesting part of your work to be the technical advance in cancer evolution analysis methods. However, they also think that the presentation of this needs to be majorly clarified and improved.

Reviewer #2 is very supportive and says their suggestions amount to a minor revision. The most important comment they make is on the robustness of your SCNA clonality calling, which underlies your analytic approach.

Reviewer #3 sounds perhaps the most skeptical of the three, but still quite positive. Their major concern is regarding the potential for early/late MRCA to be used as a predictive biomarker in the clinic, where they say further work is needed to fully substantiate the reliability and superiority over

current prognostic schema.

In our reading of these reports, there is a clear path to publication; none of the requests made by the reviewers seems impractical or impossible. We thought that the comments regarding the technical/methodological aspects (Reviewer #1) and the clinical utility (Reviewer #3) are especially important and we strongly encourage you to respond to these as fully as possible.

To guide the scope of the revisions, the editors discuss the referee reports in detail within the team, including with the chief editor, with a view to identifying key priorities that should be addressed in revision and sometimes overruling referee requests that are deemed beyond the scope of the current study. We hope that you will find the prioritized set of referee points to be useful when revising your study. Please do not hesitate to get in touch if you would like to discuss these issues further.

We therefore invite you to revise your manuscript taking into account all reviewer and editor comments. Please highlight all changes in the manuscript text file. At this stage we will need you to upload a copy of the manuscript in MS Word .docx or similar editable format.

*2) If you have not done so already please begin to revise your manuscript so that it conforms to our Article format instructions, available

http://www.nature.com/ng/authors/article_types/index.html here.

*3) Include a revised version of any required Reporting Summary:

Please be aware of our <https://www.nature.com/nature-research/editorial-policies/image-integrity> guidelines on digital image standards.

[redacted]

Sincerely,

Michael Fletcher, PhD
Senior Editor, Nature Genetics

ORCID: 0000-0003-1589-7087

Referee expertise:

Referee #1: cancer evolution

Referee #2: paediatric cancers/neuroblastoma, evolution

Referee #3: neuroblastoma, cancer genomics

Reviewers' Comments:

Reviewer #1:
Remarks to the Author:
A

This paper presents a computational analysis of Neuroblastoma (NB) formation carried out from whole-genome sequencing data. I think the main result is to find 2 distinct subgroups of NB, with distinct MRCA-timing features and associated cytogenetic events. The paper uses mathematical modelling and inference to justify its claims. In general, I feel the work is interesting but its presentation is not completely clear and needs to be improved, as much as results presentation and the code accompanying the paper. I would suggest the authors to work on these aspects - especially the modelling part - which to me might be the more interesting technical contribution of the paper, besides its biological implications. I discuss this in point F.

B

I feel this contribution is novel, but I am not a NB expert even though I have extensive experience in cancer evolutionary genomics.

C

The approach is valid. Data quality cannot be easily assessed since the authors do not show, e.g., read-counts distributions for all the 100 samples (see below, point F).

E

Conclusions are logical and follow the evidence of the data. Clarifications of the presentation will however improve the paper.

F

Major observations mostly from the main text:

- [Timing] I think some of your terminology could be clarified. e.g., you should find multiple clonal peaks for aneuploidy configurations, not a single clonal peak (page 4-6 line 26 onward etc). Second, the MRCA cannot be characterised by just single-copy SSNVs, rather by both single- and double-copy SSNVs. In fact, fig 2c contains mutations at ~66% VAF, which are from double-copy SSNVs. Moreover, from Fig 2d I see that you refer to the density of heterozygous SSNVs as those "defining the MRCA". In general, all mutations define the MRCA, even those on more complex copy states; it's just that you want to compare clocks from diploid copy states and from aneuploid ones. I have the feeling that this is because mutations pre-aneuploidy were accumulated at the same growth rate (i.e., with no selected drivers); if this was the case, I would make it explicit.

- [Timing] can you define some confidence in the predictions obtained from molecular clocks based on mutation densities? For instance, can you define a p-value (bootstrapping?) that compares the densities in heterozygous diploid regions from single-copy gains etc and shows that Chr9 and 20 are not significantly different from reference (your MRCA), while the others do?

- [Evolutionary classes] I feel biopsy size/ number of sequenced cells might be a confounder to time an MRCA. If I extract DNA from fewer cells I will sample less of the tumour, if I have taken genetically-close cells I will tend to have a more recent MRCA compared to a large sample where I get many cells, and some of them might be invariably far away. Are here data/ sampling/ DNA libraries comparable across patients? Can you comment also in light using sequencing data from 2 distinct cohorts (85+15)?

- [Popgen] I am not sure I understood how you did convert tumour doubling times to actual days. In the model μ and δ are rates per cell division, so your simulation with ABC counts cell divisions; btw it seem you get 9.6 mutations/day in the section that you refer to as "Simulating VAF distributions", even though that section has a different name in the paper. Also, is this measurement patient-specific or the same across the whole cohort? In general, I am a bit confused by the pop-gen description of the models you have implemented. First of all, you have a section called "Modeling neuroblastoma initiation" and then another "Modeling mutation accumulation during tumor growth"

but it is not 100% clear to me which is used and when to carry out simulations. I have the feeling that one refers to pre-cancerous states (so constant growth rates), while the others refers to expansions with selective advantage. The presentation should better clarify what is what. For instance, in line 10-20 pag 7 and lines 9-11 pag 8: are you suggesting constant growth rate for these groups of samples? But then when/ what model did you apply? I would suggest you to make a cartoon exemplifying your findings (eg lines ~20) in the form of a graphical abstract, also clarifying what models you have implemented in the paper (pre-tumour, tumour). Intuitions about some of the equations might also be given in the Methods, and I would report the main equations also in the main text (this is a computational paper), at least those associated to some of the plots in the main text figures. I did not check in detail all derivations but could sense some of the Durrett-alike results in the formulas presented.

- [Popgen] Are the geometric means of μ_1/μ_2 realistic? If the events happen once per million divisions does this mean that with a tumour with 10M cells that divide you get 10 such events? What is the predicted tumour size (number of cells) of your model?

----- General methodological questions

- the classification of clonal-subclonal (pag 17) relies on a p-value from a Bin density with max 80 reads with the variant. This is a p-values that does not account for sequencing overdispersion, and indeed DP is taken as an average across measures. First, you could test for the coverage of each and every mutation, or justify why 80 is a good number. Second, why don't you just carry out tumour subclonal deconvolution with standard methods (pyClone, MOBSTER, etc) to define the clonality status? You also use some of these methods below...

- pag 18 line 25: what is weighted Binomial clustering? Is a standard Binomial Dirichlet mixture? If yes, why didn't you use standard subclonal deconvolution methods? In general, I think you want to show for every sample which mutations are assigned to each one of the peaks (add this a supplementary data; see comments below); this is the only way to assess the reliability of your estimates. In general why a CCF-computation method (DPclust, CNAqc, CCube, etc) cannot be used to phase mutations to multiplicities?

- Mutation counts can be related to time with a Poisson process, but where is the growth rate of the tumour in the expression (pag 25)? In general, one thing was not completely clear to me is what growth rate are you assuming (see also the popgen comment above), if this changes over subclonal expansions and its relation to copy number events. I think these aspects should be much better clarified in the current paper, distinguishing growth in presence of a tumour fully-formed, in the initial steps etc. A growth-model-cartoon should be part of the main figures of the paper imho.

- In modelling mutation accumulation during tumor growth you carried out subclonal deconvolution with MOBSTER. What are the results obtained by that analyses? Why are these not shown?

----- General results reporting, code and data availability

Your cohort is rich and I am sure that in the field it might become popular among NB-folks.

However, at the moment results from all the patients are not reported, and only a few examples are presented. I feel that each one of the samples should be reported with its most relevant information:

clinical trajectory (as inferred), NGS assesemnt including coverage/VAF histogram/ deconvolution results, copy number profiles, mutation timing inference from density, ABC inferences and quality control, etc. These should be summarised in a 1-page PDF (1 per sample) to be released as Supplementary Data, or at Zenodo etc.

In general, I also feel that the paper falls short in terms of code organisation (a fundamental point for reproducibility) and, this being a computational paper, this should be better curated. The GitHub repository

https://github.com/hoefler-lab/Neuroblastoma_evolution

is a collection of scripts that only marginally suggest a certain logical organisation. I attempted to use/replicate some of the analyses but could not succeed in a reasonable time, and so I dropped. I think higher standards should be kept by using vignettes, tutorials in the code analysis, input files processing, visualisation etc. Right now, a collection of R scripts is not a sufficient effort to ensure reproducibility of these analyses.

For instance, you could release the code to analyse one or more samples from the cohort in a well-documented, organised and commented way, and explain how to analyse the rest of the cohort by arranging data in a ready-to-go way. Arranging a website/ package is also an option that you should consider.

A further aspect is to describe precisely what data you are releasing to replicate your analyses. I can see from one table the set of calls split as somatic mutation type, but I could not easily find allele-specific CNAs (locations, major/minor allele copies) to compute phasing and check densities etc. Your R code should clarify all these aspects in my opinion. Consider releasing these data via Zenodo or other portals.

----- Minor:

- [Timing] As a remark: if mutation densities in single and double copies are similar for distinct CNAs, then you can conclude that all CNAs happened pretty much at the same instant. You might want to relate this to the ideas of punctuated evolution or hopefull monsters.

- [Timing] mutation rate: I think you are assuming that the mutation rate is locally constant. This might be appropriate but you might want to show that if you take random chunks of genome with the same copy state, for a certain size of genome, then you find statistiacly indistinguishable densities.

- [Timing] I think you can deduce the fact that the ECA preceeds the MRCA by just the fact that the CNAs are clonal (page 6 line 13), but it seems you make this sound as less straightforward. I am not sure I understand why.

- [Timing] Fig 2e and f, what are the colors?

- [Timing] Please clarify how you did detect subclonal CNAs and why they do not confound your inference;

- [Timing] Mutational signatures: I think the AC-naming convention is from an older COSMIC version, can you lift it to the latest SBS-one? Also, can you report the amount of signature per sample? Are these changing or do they stay with similar proportions across all samples?
- [Evolutionary classes] The detection of MRCA's closer to sampling (in relapses) makes sense; please provide some details for non-NB expert of what is the therapy adopted here.
- [Evolutionary classes] How is the cut early/late MRCA's defined in Fig 3a/4c?
- [Telomere] In some cases you identify telomere maintenance gained between the ECA and the MRCA; would you think this could cause an increased growth rate or better a decreased death rate in the tumour? Can you comment? In general, can these events be acquired with any temporal order relative to CNAs? If yes, do you have evidence that orders are commutative?

G

Appropriate credit to previous work is given even though some things are re-implemented while I feel they could have been done with standard methods (e.g, Binomial deconvolution, multiplicity phasing etc). The authors might clarify why published methods could not be used.

H. See F.

Reviewer #2:

Remarks to the Author:

Körber et al perform an in depth analysis on the somatic evolution of neuroblastoma by analysing a discovery cohort of cases

With deep WGS data (~80x), n=100. 67 of the samples were from initial diagnosis (7 of these from metastatic lesions). The rest (n=33) from

relapse tumors. They integrate data on somatic copy number alterations and somatic SNVs to estimate when specific genetic alterations occurred, and use mathematical models of neuroblastoma tumorigenesis and approximate bayesian computation to infer time points during the patients life when genetic alterations should have arisen. They then validate their findings in an additional cohort of patients, sequenced to an average of 30x. These findings are novel and adds to our understanding of neuroblastoma evolution.

The authors should be commended for this very clear article and the work presented should be of broad interest to researchers working in both

Cancer genomics, tumor evolution and also to clinicians treating patients with Neuroblastoma.

Furthermore the code is published on GitHub and the data deposited in repositories for access by other researchers. Especially the result that late MRCA tumors tend to have a poorer prognosis is intriguing and while Körber et al note that this group is also enriched for tumors with segmental copy number alterations and thus might be more genomically unstable, one wonders whether there are other factors (microenvironmental?) that predispose these tumors to evolve for a longer time.

All in all, the paper and the results presented in it are well described and backed up by the data and

careful analysis, and my recommendation is minor revision.

Questions/Comments:

* While Körber et al describe how they guarded against misclassification of subclonal somatic SNVs as clonal ones in my mind there is still some concern about whether the classification of SCNAs as clonal or sub clonal is robust. As it has been shown that analysis using only a single tumor sample may miscall sub-clonal SCNAs as clonal (as we and others have shown: c.f PMID: 29867221, PMID: 32041836 and PMID: 34815394). The authors use of experimentally derived ploidy is helpful here (constraining the number of (purity, ploidy)-pairs that fit the data.) In my mind, this should be further discussed/analyzed, for instance, for the subset of primary/relapse-pairs, how did the SCNAs differ between the samples? In addition, how far apart in the biopsy was the material used for WGS from the material that was used for ploidy-analysis?

* It is unclear from the manuscript whether they used all somatic SNVs in their timing analysis, or just the ones inferred to be due to 'Molecular clock'-like signatures (AC1 and AC2 using the nomenclature from the authors). This should be clarified, and if all sSNVs were used, how did the authors test the assumption of ~constant mutation rate for AC3 (Cosmic signature #18 - where the proposed causal mechanism is reactive oxygen species?).

* In their survival analysis the authors dichotomise the sSNV/mb parameter (to define early vs late MRCA) and refer to the bimodal distribution of sSNVs/Mb. This analysis should be repeated with sSNV/mb as a continuous variable, as dichotomising typically leads to loss of information (see <https://www.ncbi.nlm.nih.gov/pmc/articles/PMC1458573/>)

* Was there any pattern in when specific genetic alterations (e.g. +17q) typically occurred?

Anders Valind
MD, PhD
Department of Clinical Genetics Lund University

Reviewer #3:

Remarks to the Author:

Körber and colleagues from the Höfer and Westermann groups report about somatic evolution of the childhood tumor neuroblastoma in early fetal development calculated and quantified by an innovative bioinformatics/systems biology approach consisting of deep whole-genome sequencing, molecular clock analysis and population-genetic modeling in a discovery cohort of 67 primary neuroblastomas and a validation cohort of 86 tumors. Their interesting major new findings include

a) start of neuroblastoma development occurs via aberrant mitoses between week 4 and 12 of pregnancy (this has important implications for future prevention strategies and epidemiological/environmental studies!)

b) neuroblastomas with favorable prognosis cease to evolve early, whereas aggressive neuroblastomas show prolonged evolution during which they acquire telomere maintenance mechanisms and

c) duration of evolution is an accurate predictor of outcome

These data will indeed be of high interest and very useful to the neuroblastoma research community by significantly advancing current insights into neuroblastoma development and evolution. It provides the bioinformatical explanation and rationale of biological and clinical phenomena previously observed in patients and their tumor material. In addition, it harbors a high potential for clinical relevance as a novel predictive biomarker.

However, the claim that the findings may prospectively guide treatment decisions with MRCA timing as a reliable and superior outcome predictor needs some further substantiation (see below). I have the following concerns:

1. How do the authors explain the discrepancy that MRCA timing was a significant predictor of overall survival in the validation cohort (86 samples), but not in the discovery cohort (67 samples)?
 2. To justify a future effort of calculating MRCA timing for individual patient tumors as a new outcome predictor, the superiority and independence of MRCA timing should be shown in comparison to more recently identified and meanwhile clinically used molecular outcome predictors such as established mRNA classifiers (Oberthuer et al, JCO 2010 and Clin Can Res 2015; Rosswog Neoplasia 2017; Volland Genome Biol 2015) or RAS/p53 pathway mutations (Ackermann et al, Science 2018), preferably for the whole patient cohort (if these data are available) or, if this is not feasible, for a subcohort of patients. The comparison with the old clinical classification (age, stage, MYCN amplification) in the trial NB2004 and the telomere maintenance status alone is interesting, but less informative. If these comparative data are not available at all, the need of this validation step has at least to be included into the discussion.
 3. The mutation rate in late MRCA/high-risk/segmental gain neuroblastomas might be much higher than in low-risk tumors. Indeed high-risk tumors have a much higher expression of DNA repair genes. The presence of oncogenic gene defects and segmental defects causes an entirely different type of tumor cell, with potential effects on mutation rate. This would inflate the number of mutations between ECA and MRCA and place the ECA erroneously early in development.
 4. High risk neuroblastoma, correlating with late MRCA/early ECA tumors, are diagnosed at a much higher age than low-risk tumors. If they have a longer distance in time to the ECA, the ECA of high risk tumors may still be much later than the ECA of low risk tumors. I did not notice that the effect of the difference in age at diagnosis was included in the analyses, modelling or discussion.
 5. Furthermore, the authors use gained chromosomal areas to infer ECA and MRCA. That implies that in triploid cases, they use the entire genome to count SSNVs, while in near-diploid cases, they necessarily use the typically gained regions, probably #7 and #17q. A uniform analysis, focussing on the q arm of #17 which is gained in almost all neuroblastoma, would avoid this possible bias.
 6. The authors have validated their data in the context of temporal evolution (first diagnosis versus relapse samples), but not in the context of potential spatial evolution. This has to be discussed in light of the future potential/need of spatially resolved transcriptomics technologies.
- Minor: A tumor cell content of 88% in average seems surprisingly high considering the fact that all tumor stages have been included. Can the authors explain and prove how this was assessed?

Author Rebuttal to Initial comments

Point-by-point response to reviewers' comments

Reviewers' comments in blue, our responses in black

As explained in the reply to Reviewer #1, we will make the following additional data available on

Mendeley and, for the review process, also provide them under download links that can be accessed without a Mendeley account:

[redacted]

The new tutorials on the computational analysis are available at https://github.com/hoeferlab/Neuroblastoma_evolution.

Reviewer #1:

Remarks to the Author:

A

This paper presents a computational analysis of Neuroblastoma (NB) formation carried out from whole-genome sequencing data. I think the main result is to find 2 distinct subgroups of NB, with distinct MRCA-timing features and associated cytogenetic events. The paper uses mathematical modelling and inference to justify its claims. In general, I feel the work is interesting but its presentation is not completely clear and needs to be improved, as much as results presentation and the code accompanying the paper. I would suggest the authors to work on these aspects - especially the modelling part - which to me might be the more interesting technical contribution of the paper, besides its biological implications. I discuss this in point F.

B

I feel this contribution is novel, but I am not a NB expert even though I have extensive experience in cancer evolutionary genomics.

We thank the reviewer for the careful reading of our manuscript and the very insightful and constructive suggestions for improving it. In particular, the comments on the mathematical and computational analyses helped us clarify the logic of our arguments, examine these arguments further, and make the computational framework, as well as all important data, more readily accessible.

C

The approach is valid. Data quality cannot be easily assessed since the authors do not show, e.g., read-counts distributions for all the 100 samples (see below, point F).

We now make the relevant data for the individual samples accessible. Please see our response to Point F.

E

Conclusions are logical and follow the evidence of the data. Clarifications of the presentation will however improve the paper.

F

Major observations mostly from the main text:

1. [Timing] I think some of your terminology could be clarified. e.g., you should find multiple clonal peaks for aneuploidy configurations, not a single clonal peak (page 4-6 line 26 onward etc). Second, the MRCA cannot be characterised by just single-copy SSNVs, rather by both single- and double-copy SSNVs. In fact, fig 2c contains mutations at ~66% VAF, which are from double-copy SSNVs. Moreover, from Fig 2d I see that you refer to the density of heterozygous SSNVs as those "defining the MRCA". In general, all mutations define the MRCA, even those on more complex copy states; it's just that you want to compare clocks from diploid copy states and from aneuploid ones. I have the feeling that this is because mutations pre-aneuploidy were accumulated at the same growth rate (i.e., with no selected drivers); if this was the case, I would make it explicit.

We completely agree that all clonal mutations define the MRCA. Indeed, this is how we did our analyses. Thanks to your comment we noticed that this was not apparent in the main text when explaining MRCA timing (page 5). Figure 2b shows that there are multiple clonal peaks, which were ordered by copy number states (diploid and aneuploid, including the mentioned ~66% VAF SSNVs). For computing the SSNV density of the MRCA, we used all these clonal peaks – precisely as you argue. We have now explained this in more detail in the main text, writing on page 5: "In addition, on copy numbers larger than 2, we detected clonal mutations present on the two (and in some cases more) copies of a multiplied chromosome. Collectively, both single-copy and multiple-copy clonal SSNVs characterize the MRCA (Fig. 2c)." We also tested whether the clonal SSNVs on a given segment belonged to the SSNV distribution defined by all segments collectively (see next point).

2. [Timing] can you define some confidence in the predictions obtained from molecular clocks based on mutation densities? For instance, can you define a p-value (bootstrapping?) that compares the densities in heterozygous diploid regions from single-copy gains etc and shows

that Chr9 and 20 are not significantly different from reference (your MRCA), while the others do?

This is an excellent point. Indeed, we tested whether the SSNV density of each genomic segment belongs to the SSNV density distribution of the MRCA. First, we ascertained that the SSNVs characterizing the MRCA across the genome can be modelled by a common negative binomial distribution (accounting for overdispersion due to locally varying mutation rates, e.g., Martincorena et al. Cell 2017). Second, for each individual segment, we computed the p value that its clonal SSNVs belong to the MRCA SSNV density defined by all segments collectively (corrected for multiple testing, Holm correction; significance level of 0.01). Over all samples, we found that the clonal SSNVs on 99% of the segments belonged to the common distribution (consistent with our significance level).

Third, to address the timing of copy number gains, we asked whether amplified clonal SSNVs of each amplified segment belong to the same distribution; this was done for each segment individually. As suggested, we thus obtain a p value for each segment with a copy number gain to decide whether the density of amplified clonal SSNV density is drawn from the MRCA distribution; we use 0.01 as significance level. The null hypothesis will be rejected if the density of amplified clonal SSNVs is significantly smaller than the MRCA density, this defines an MRCA (note that gains occurring after the MRCA will be subclonal; as we were interested in neuroblastoma initiation, we did not analyse this small fraction – see Fig. 1c – of gains further). In addition to p-values in relation to the MRCA SSNV density distribution, we also computed confidence bounds for the SSNV density point estimate for each segment. We now provide all this information in the figures and amended the main text, as follows:

You refer specifically to the example shown in Fig. 2f, and this provides a good illustration of the above. The densities of amplified clonal SSNVs on all segments except for Chr9 and 20q were significantly smaller than the SSNV density of the MRCA and, moreover, are consistent with originating from the same distribution; hence they define an ECA associated with near-triploidization of the genome. The amplified clonal SSNVs on Chr9 and 20q were not significantly different from the MRCA density. We have now indicated the results of the statistical tests in the figure; similarly, we added this information to Extended Data Fig. 3a, c and d. The confidence bounds on the density estimates had already been shown in these figures in the initial version. Evidently, shorter segments (e.g., Ch20 in Fig. 2f) have a larger confidence interval. We amended the text on page 6: “Next, we asked whether clonal chromosomal gains occurred around the time when the MRCA arose or ancestral to the MRCA. To this end, we quantified SSNVs that were clonal on two copies on trisomic and tetrasomic segments (termed amplified clonal SSNVs⁸; Fig. 2b). These mutations were acquired prior to a gain on the respective allele (Fig. 2c, dark

green), and their density thus scales with the time at which the chromosomal gain occurred^{8,34}. We compared these densities to the densities at MRCA based on a negative binomial distribution of clonal SSNVs across the genome (Methods). In the example tumor, nearly all gains, except for Chr. 9 and 20q, had a mean density of 1 amplified clonal SSNV per 100 Mbp, which is significantly smaller (adjusted p value < 0.01) than the mutation density of the MRCA (Fig. 2e). Hence, the molecular clock places these gains ancestral to the MRCA, in an early common ancestor (ECA) of the tumor. The only exceptions were gains of Chr. 9 and 20q, which had a mutation density consistent with that of the MRCA and hence occurred later than the ECA.”

3. [Evolutionary classes] I feel biopsy size/ number of sequenced cells might be a confounder to time an MRCA. If I extract DNA from fewer cells I will sample less of the tumour, if I have taken genetically-close cells I will tend to have a more recent MRCA compared to a large sample where I get many cells, and some of them might be invariably far away. Are here data/ sampling/ DNA libraries comparable across patients? Can you comment also in light using sequencing data from 2 distinct cohorts (85+15)?

We agree that these are important considerations. We address in turn the experimental sampling and the computational analyses we did to look for potential biasing effects of sampling in the data. Briefly, from all tumors DNA was sampled using a standard protocol. We did not detect strong effects of sampling on MRCA density.

In more detail, all resected tumors, from both cohorts, were processed according to a standard protocol (German NB study protocol). The tumor is divided into 4 quadrants, all of which are evaluated histologically. The cross-section of one quadrant is used for WGS; the same quadrant is used for ploidy analysis. We added this information now to the Methods (page 17): “For analysis, all resected tumors were divided into 4 quadrants, all of which were evaluated histologically. MYCN status was assessed as routine clinical marker for all tumors using FISH (fluorescence in situ hybridization). A cross-sectional slice of one quadrant was used for DNA extraction for WGS; the same quadrant was used for ploidy analysis, measuring the DNA index.”

Despite the standardized sampling across the cohort, the question whether subsampling of the entire tumor for WGS biases MRCA timing is pertinent, as subclonal mutations that happened to be present in all cells of the sample would erroneously be classified as clonal. We performed two types of analyses to get information on the size of this biasing effect. First, we analyzed pairs of primary and relapse samples and required clonal SSNVs to be present in both. This was the case for 85±5% of SSNVs. To not overestimate the time of the MRCA, we corrected our estimates of clonal SSNV densities for all tumors by this factor. We write on pages 5 and 6: “To evaluate a potentially biasing effect of partial sampling of the tumor, which may erroneously cause some

subclonal mutations to be classified as clonal in the sample, we analyzed pairs of primary and relapse samples and found that the vast majority (85±5%) of clonal SSNVs were present in both samples, indicating that sampling did not induce a strong bias. Nevertheless, to estimate MRCA density conservatively, we performed all subsequent computations with the measured densities corrected by the factor 0.85.” Second, we analysed somatic copy number gains. A gained segment that was subclonal but erroneously identified as clonal in the tumor sample would have a clonal SSNV density higher than that of the MRCA. Across the entire cohort (N = 186), we found a very small number of such segments in only seven tumors (five primary tumors and 2 relapse tumors), and we excluded these putative subclonal segments from further analysis. To clarify this, we have now added to the Methods, on page 22: “here we excluded a small number of segments in seven tumors with a larger density of amplified clonal mutations than the estimated mutation density at the MRCA, because such gains may be subclonal CNAs that were erroneously classified as clonal”. In sum, we did not detect pervasive effects of subsampling on the mutation density of the MRCA, and corrected for very small such effects to guard against a (small) overestimation of the mutation density of the MRCA.

4. [Popgen] I am not sure I understood how you did convert tumour doubling times to actual days. In the model μ and δ are rates per cell division, so your simulation with ABC counts cell divisions; btw it seem you get 9.6 mutations/day in the section that you refer to as “Simulating VAF distributions”, even though that section has a different name in the paper. Also, is this measurement patient-specific or the same across the whole cohort?

We agree with you that this point deserves better explanation, which we now provide in the new Figure 6, in the accompanying text, and in the revised part of the Methods, “Real-time estimates of cell division rate” (previously being mislabeled “Simulating VAF distributions”). We also thank you for picking up our mistake in mislabeling this section.

First, we developed a population genetics model of neuroblastoma initiation for late-MRCA neuroblastomas, which we fitted to the SSNV densities of ECA and MRCA, yielding, among other parameters, the mutation rate μ and the loss rate δ , measured per division, as well as the selective advantage s of the neuroblastoma cells over the sympathetic neuroblasts. The cell division rate λ remains unknown in this framework. Intuitively, λ scales real time whereas the data on ECA and MRCA timing are in terms of SSNV densities – hence we have no real-time information in this dataset. Second, real time is introduced by requiring the primary tumors to grow from one cell (= the MRCA) to detectable size $N_T(t_D)$ up to the time point of diagnosis t_D . Assuming a uniform division rate of neuroblasts and tumor cells, we have

$$t_D = \frac{\tilde{m}_{\text{MRCA}}}{\lambda\mu} + \frac{\log N_T(t_D)}{\lambda\left(1 - \frac{\delta}{s\lambda}\right)},$$

Where \tilde{m}_{MRCA} denotes the SSNV count of the MRCA (which is measured). The first term converts the SSNVs accumulated up to the MRCA into real time while the second term results from assuming exponential growth from the MRCA to the detected tumor. Third, we assumed that detected tumors have around 10^9 cells, corresponding to the cell count in one cubic centimeter of neural tissue (Del Monte, Cell Cycle, 2009; DeVita et al., Cancer, 1975; Milo et al., Nucleic Acid Res., 2010); as the time to detection depends logarithmically on this size estimate, the latter does not need to be known with precision. Forth, we noticed that we can improve the credible interval for the estimate of $1 - \frac{\delta}{s\lambda}$ beyond the result obtained by fitting to the population data by using the VAF histograms of individual tumors. Here, the effective mutation rate $\mu_{\text{eff}} = \frac{\mu}{1 - \frac{\delta}{s\lambda}}$ can be estimated from the subclonal tails of the VAF histograms, which, using the existing estimate of μ from the population-level analysis, yields $1 - \frac{\delta}{s\lambda}$ for the individual tumors. With all the available data ($\tilde{m}_{\text{MRCA}}, \mu, \log N_T(t_D), 1 - \frac{\delta}{s\lambda}$), the above equation yields λ , allowing us to convert SSNV densities into approximate real-time estimates. For this purpose, we averaged across the inferences from the 44 primary tumors/metastasis, excluding 18 relapsed tumors among the 62 tumors that went into the analysis of subclonal tails of the VAF histograms. This is now explained more clearly in the Methods and, schematically, in the new Figure 6. While re-checking all computations, we also detected an error that unfortunately occurred when copying data files. We sincerely apologize for this inadvertent error. Rectifying it, the correct inference of the SSNV rate is 3.2 ± 0.4 SSNVs per day for all SSNVs (page 12): “The inferred rate of SSNV acquisition (μ) was 3.2 ± 0.4 SSNVs per day, which falls in the same range as a recent direct measurement of somatic mutation rate in the developing central nervous system [5.1 (1.5, 9; 95% CI) SSNVs per day]³⁹.” As a further result, we obtain tumor-specific division rates along with relative death rates, which we stratify by telomere maintenance mechanism (Figure 7 g,h). Of note, our inferred cell-cycle times of 2 to 5 days were consistent with experiments determining the division rates of neuroblastoma cells *in vivo*, yielding an average of 4.2 days (done by giving the patients Colcemid to arrest cells in mitosis before resecting the tumor; Aherne and Buck, Br J Cancer 1971). We added on page 13: “our estimated cell division rates agree quantitatively with rates measured in neuroblastoma *in vivo*.⁴²”

5a. In general, I am a bit confused by the pop-gen description of the models you have implemented. First of all, you have a section called "Modeling neuroblastoma initiation" and then another "Modeling mutation accumulation during tumor growth" but it is not 100% clear to me which is used and when to carry out simulations. I have the feeling that one refers to pre-

cancerous states (so constant growth rates), while the others refers to expansions with selective advantage. The presentation should better clarify what is what.

This point is well-taken. We have done the following to present the modeling in a more readily accessible way: First, we have added an overview figure, now Figure 6, as you suggested (below). This figure introduces the principal ideas for our population-based and individual tumor-based inferences. Second, while keeping the overall structure of the Methods sections on modeling and inference, we have thoroughly revised the text to increase readability.

5b. For instance, in line 10-20 pag 7 and lines 9-11 pag 8: are you suggesting constant growth rate for these groups of samples?

Here, growth rate does not factor in. The analysis is based on SSNV densities. To clarify, we added on page 7: “These data raise the question of whether late-MRCA tumors began to develop later, or developed early and evolved for a longer period of time. To address this question in terms of the molecular clock, we analyzed the SSNV densities on chromosomal/segmental gains in both tumor classes, these may define an ECA.”

5c. But then when/ what model did you apply? I would suggest you to make a cartoon exemplifying your findings (eg lines ~20) in the form of a graphical abstract, also clarifying what models you have implemented in the paper (pre-tumour, tumour).

We thank you for this suggestion and now summarize our modeling and inference approaches in the new Figure 6 (see comment above).

5d. Intuitions about some of the equations might also be given in the Methods, and I would report the main equations also in the main text (this is a computational paper), at least those associated to some of the plots in the main text figures. I did not check in detail all derivations but could sense some of the Durett-alike results in the formulas presented.

We now added key equations in the new Figure 6 and revised the Methods (see above), also giving more intuition on key equations. We also checked Durrett, Branching Process Models of Cancer and more recent papers. In Durrett, A waiting time problem arising from the study of multi-stage carcinogenesis, the authors model a two-step process of tumor initiation in a homeostatic tissue, which we now cite on p. 24. While similar to our approach in spirit, our model involves different growth regimes: the basic cell dynamics during which the driver mutations happen are exponential expansion of precursors followed by contraction or homeostasis.

6. [Popgen] Are the geometric means of μ_1/μ_2 realistic? If the events happen once per million divisions does this mean that with a tumour with 10M cells that divide you get 10 such events? What is the predicted tumour size (number of cells) of your model?

We agree that the rate of acquiring driver mutations is a pertinent question of tumor evolution that we believe is not yet well understood. Model-based inference from sequencing data, as we perform here, is one source of such information. As we detail in the following, our estimate is reasonable in the light of what is known about neuroblastoma biology.

Mean driver mutation rate: Our estimates of μ_1 and μ_2 include all driver events (SSNVs, indels, CNAs). We identified 28 distinct driver events in our cohort (oncogenes and gains/losses that are known to be highly recurrent in neuroblastoma). As an order-of-magnitude calculation, a rate of the order of 1–10 SSNV per cell division and a driver rate of 10^{-6} per division amounts to about 300 positions in a haploid genome that, when mutated, cause a selective advantage. At least for tumor suppressors, several positions per gene may cause loss of function, so that the number of distinct oncogenes may well be an order of magnitude smaller. Of course, the rate of SSNVs leading to drivers will be smaller (and hence there will be fewer SSNV-caused drivers than our rough calculation suggests), as prominent driver events are CNAs (and so there will be CNA-based drivers); overall, this simple calculation suggests that our estimated rate can be realized with several tens of distinct drivers. A more detailed estimate of this kind, based on 286 tumor suppressors and 91 oncogenes in the human genome, suggests a driver mutation rate of 3.4×10^{-5} per cell division (Bozic et al., PNAS, 2010). It is re-assuring that our estimate is an order of magnitude lower, as only a subset of these events will be relevant for neuroblastoma. Hence, our estimated driver mutation rate is not incompatible with the overall number of distinct oncogenic events in neuroblastoma.

Number of events in a tumor: The typical size of the resected tumor is 10^9 cells (see above). Our estimate of the driver mutation rate holds up to the emergence of the MRCA, long before such a size is reached. At this point in time, the rapid divisions of fetal sympathetic neuroblasts might cause a higher error rate than what would occur in more slowly proliferating tumor cells. In particular, the cell division rate in developing tissues in the fetus can be as high as 3 times per day, whereas we estimate that tumor cells divide once every 2 to 5 days, which agrees with a measurement in the literature (Ref. 42; see Point 4 above). One might speculate that the high division rate may make chromosome segregation less reliable, hence the early aneuploidization events; the SSNV rate, however, may not be affected.

As you write, 10 M divisions would create 10 driver events with our rate, but subsequently such an event may be lost again or reach fixation; hence, the number of surviving drivers will be lower and depend on their selective advantage. We use the survival probability of a linear birth-death process (see page 24, Equations 12a, b), which is of order of 10^{-1} with our inferences of the ratio of death to birth rates in the tumor and selective advantage. Hence only a minority of drivers will survive and shape clonal growth.

To address your point, we amended the text on page 12 as follows: “The model further inferred that oncogenic driver events occurred on average once per one million cell divisions (geometric mean of μ_1 and μ_2), which is consistent with a global estimate of a driver mutation rate of 3.4×10^{-5} per division in the human genome¹⁶ given that only a subset of all potential drivers will cause neuroblastoma.”

----- General methodological questions

7. the classification of clonal-subclonal (pag 17) relies on a p-value from a Bin density with max 80 reads with the variant. This is a p-values that does not account for sequencing overdispersion, and indeed DP is taken as an average across measures. First, you could test for the coverage of each and every mutation, or justify why 80 is a good number. Second, why don't you just carry out tumour subclonal deconvolution with standard methods (pyClone, MOBSTER, etc) to define the clonality status? You also use some of these methods below...

This point relates to the classification of individual SSNVs as clonal or subclonal to subsequently determine mutational signatures separately for these types of SSNVs. To improve this classification, we followed your suggestion and used the actual coverage at each mutation.

We also updated the calling of the signatures to COSMIC v3 (see also below, page 9), obtaining very similar results as before with respect to comparable signatures (clock-like, SBS18/AC18 – known to be frequent in neuroblastoma – and SBS3/AC3 dominated). In new Extended Data Figure 2d and e, we show the comparison of signatures of clonal and subclonal SSNVs, which we now additionally separated into early-MRCA and late-MRCA cases. This updated analysis confirms our earlier result that clonal and subclonal signatures were overall very similar. Interestingly, however, SBS18 was more prominent, at both clonal and subclonal level, in late-MRCA tumors. On page 7, we added: “Of note, overall the majority of SSNVs in both early-MRCA and late-MRCA tumors were clock-like, while SBS18, characteristic of neuroblastoma, was more abundant in late-MRCA tumors (Extended Data Fig. 2d, e).”

8. pag 18 line 25: what is weighted Binomial clustering? Is a standard Binomial Dirichlet mixture? If yes, why didn't you use standard subclonal deconvolution methods? In general, I think you want to show for every sample which mutations are assigned to each one of the peaks (add this a supplementary data; see comments below); this is the only way to assess the reliability of your estimates. In general why a CCF-computation method (DPclust, CNAqc, CCube, etc) cannot be used to phase mutations to multiplicities?

Indeed, we use a standard binomial mixture and revised the wording in the Methods section; we now also show the fitted distributions in the sample sheets for each tumor (available on Mendeley). Following your suggestion, we compared peak deconvolution with an alternative method, CNAqc, and obtained very similar results (Figure for the reviewer below). As we had specific requirements which were not implemented in CNAqc, we stayed with our method. Specifically, CNAqc does not allow quantification of tetrasomic chromosomal segments with a 3:1 configuration, which we find in our data and use in the analyses. The compute_CCF function in CNAqc does not classify SNVs in the middle between two peaks (as its main purpose appears to be classifying individual variants as clonal or subclonal), causing an underestimation of mutation densities as can be seen in the Figure for the reviewer.

Figure for the reviewer: Comparison of mutations densities on individual segments computed with our method (x axis) and CNAqc (y axis). Segments were defined by copy number (see Figure 2a in the manuscript); all tumor samples are shown. Densities are positively correlated with Pearson's $r=0.84$ (0.83, 0.84).

9. Mutation counts can be related to time with a Poisson process, but where is the growth rate of the tumour in the expression (pag 25)?

To clarify, Equation (12) – now Equation 16 on page 27 – relates to the first oncogenic event, and hence there is no tumor growth to consider. We revised the text describing the modeling in the Methods to make this and other aspects (mentioned in several other points) clearer.

10. In general, one thing was not completely clear to me is what growth rate are you assuming (see also the popgen comment above), if this changes over subclonal expansions and its relation to copy number events. I think these aspects should be much better clarified in the current paper, distinguishing growth in presence of a tumour fully-formed, in the initial steps etc. A growth-model-cartoon should be part of the main figures of the paper imho.

We apologize that this has not become clear from the manuscript. We have now included the new Figure 6 to sketch to provide a cartoon of the modeling, including the growth model underlying the two-event scenario of initiating late-MRCA tumors. Briefly, for all inferences on tumor initiation, the cell division rate per real time, λ , cannot be determined, as all timing information is in terms of SSNV counts, associated with ECA and MRCA. We estimate λ in a separate step that uses the approximate tumor size at diagnosis as well as additional data from the tumors, namely the subclonal tail of the VAF histogram. This is explained in more detail in Point 4 above, where we also outline the actions taken to clarify this in the manuscript. The difference in net growth rate between the initiation phase and the fully formed tumor is due to a reduction in the cell loss rate (due to the selective advantage of the second hit). This specific choice was motivated by the fact that late-MRCA tumors gain telomere maintenance mechanisms, very likely to stabilize cell survival.

10. In modelling mutation accumulation during tumor growth you carried out subclonal deconvolution with MOBSTER. What are the results obtained by that analyses? Why are these not shown?

Initially, we did not show subclonal deconvolution of the resected tumors as the manuscript focuses on tumor evolution up to the MRCA. However, we agree with you that subclonal deconvolution is interesting information about our cohort and include the Mobster results for each individual tumor now in the tumor data sheets (available on Mendeley).

----- *General results reporting, code and data availability*

11. Your cohort is rich and I am sure that in the field it might become popular among NB-folks. However, at the moment results from all the patients are not reported, and only a few examples are presented. I feel that each one of the samples should be reported with its most relevant information: clinical trajectory (as inferred), NGS assessemnt including coverage/VAF histogram/ deconvolution results, copy number profiles, mutation timing inference from density,

ABC inferences and quality control, etc. These should be summarised in a 1-page PDF (1 per sample) to be released as Supplementary Data, or at Zenodo etc.

We thank you for this excellent suggestion. We now prepared tumor data sheets and uploaded them to Mendeley, where they are available for preview by the reviewers and will be made public upon publication of our study. As suggested, the data sheets contain (a) copy number profile, (b) ploidy estimate, (c) VAF histograms for each copy number (up to $CN = 4$), (d) mutational signatures, (e) timing of MRCA, ECA and all gained segments via mutations densities, (f) subclonal deconvolution (Mobster), (g) ABC fit of model for SSNV accumulation during tumor growth and (h) estimated effective mutation rate. We note that clinical classification and outcome are recorded in Supplementary Table 1.

12. In general, I also feel that the paper falls short in terms of code organisation (a fundamental point for reproducibility) and, this being a computational paper, this should be better curated. The GitHub repository https://github.com/hoefer-lab/Neuroblastoma_evolution is a collection of scripts that only marginally suggest a certain logical organisation. I attempted to use/replicate some of the analyses but could not succeed in a reasonable time, and so I dropped. I think higher standards should be kept by using vignettes, tutorials in the code analysis, input files processing, visualisation etc. Right now, a collection of R scripts is not a sufficient effort to ensure reproducibility of these analyses. For instance, you could release the code to analyse one or more samples from the cohort in a well-documented, organised and commented way, and explain how to analyse the rest of the cohort by arranging data in a ready-to-go way. Arranging a website/ package is also an option that you should consider.

We thank you for taking the time to try our code and for making very useful suggestions to improve code availability and user friendliness. To address your request, we prepared two tutorials. The first tutorial (Example_code_mutation_density.pdf) explains how to compute the SSNV densities at copy numbers 1-4 for a specific example tumor and visualize the results. The second tutorial (Dynamics_of_neuroblastoma_initiation.pdf) outlines how to perform ABC inference from the SSNV density data of the respective cohort of tumors (late-MRCA cases) and visualize the results. We tested the tutorials with group members unfamiliar with popgen analyses of tumors and found that they reproduced the results for the example tumor within a working day.

To reproduce all our results more readily, we made the input from the WGS analysis pipelines available. These comprise SSNVs, indels, structural variants and CNVs. For the example tumor addressed in the tutorials, these data files are available on the github page. The data files for all

other tumors are available on Mendeley (upon making code and data publicly available, we will provide mutual links on the Mendeley and github pages).

Thus, our results can now be reproduced by following step-by-step tutorials and using the data for the entire cohort. Moreover, to reduce the number of R scripts, we combined all user-defined functions into the R package NBevolution. Installing this package makes all functions available. Finally, we extensively revised and extended README.MD on github.

13. A further aspect is to describe precisely what data you are releasing to replicate your analyses. I can see from one table the set of calls split as somatic mutation type, but I could not easily find allele-specific CNAs (locations, major/minor allele copies) to compute phasing and check densities etc. Your R code should clarify all these aspects in my opinion. Consider releasing these data via Zenodo or other portals.

As described under the previous point, we made the input from the WGS analysis pipelines available. These comprise SSNVs, indels, structural variants and CNVs, which are available for all tumors on Mendeley. Of course, all original sequencing data (BAM) files will also be made available, and the EGA links are in the manuscript.

----- Minor:

- [Timing] As a remark: if mutation densities in single and double copies are similar for distinct CNAs, then you can conclude that all CNAs happened pretty much at the same instant. You might want to relate this to the ideas of punctuated evolution or hopeful monsters.

Thank you for this suggestion. We added on page 14 (Discussion) “This scenario provides a specific mechanism for generating hopeful monsters in the evolution of embryonal tumors – grossly altered clones that may be selected (Turajlic et al. 2019).”

- [Timing] mutation rate: I think you are assuming that the mutation rate is locally constant. This might be appropriate but you might want to show that if you take random chunks of genome with the same copy state, for a certain size of genome, then you find statistically indistinguishable densities.

We agree, and we have done this. Please see our response to your second major point (page 2).

- [Timing] I think you can deduce the fact that the ECA precedes the MRCA by just the fact that the CNAs are clonal (page 6 line 13), but it seems you make this sound as less straightforward. I am not sure I understand why.

This comment refers to the passage: “Next, we asked whether clonal chromosomal gains occurred around the time when the MRCA arose or ancestral to the MRCA. To this end, we quantified SSNVs that were identical and clonal on two copies on trisomic and tetrasomic segments (termed amplified clonal SSNVs⁸; Fig. 2b).” To make the second sentence clearer, we added “identical”. There are two possibilities: Either these amplified clonal SSNVs have a lower density than the MRCA SSNV density, in which case the gain defines an ECA preceding the MRCA, or they have the same density, in which case they occurred at the onset of clonal tumor outgrowth (note that we did not consider subclonal segments, which exist in a small number of samples, as our analyses focus on the onset of neuroblastoma). In the same paragraph, we introduce the term ECA for the former case. Hence, clonality of amplified SSNVs does not imply that they are coincident with the MRCA.

- [Timing] Fig 2e and f, what are the colors?

These are the segments of a given copy number that we define in Fig. 2e. We have made this now clearer in the figure legend as follows: “Each segment of equal copy number is marked by a color in the bar at the bottom; these colors are used below to mark the segments in e and f.”

- [Timing] Please clarify how you did detect subclonal CNAs and why they do not confound your inference;

Copy numbers were called using ACEseq, and overall ploidy was confirmed with a subset of tumors by flow cytometry. The subclonal segments were not used for inference. To clarify, we added on page 4: “Allele-specific copy numbers were computed with ACEseq. Median tumor purity was high (88%) and allowed for reliable estimation of tumor ploidy (Fig. 1b), which we confirmed by direct measurement of the DNA index (Extended Data Fig. 1b).” We emphasized throughout the text on SSNV density inference that we considered only clonal variants.

- [Timing] Mutational signatures: I think the AC-naming convention is from an older COSMIC version, can you lift it to the latest SBS-one? Also, can you report the amount of signature per sample? Are these changing or do they stay with similar proportions across all samples?

We lifted the calling of mutational signatures to COSMIC v3. The signatures are now shown for each sample in Figure 1e; we found the major signatures in all tumors, albeit with somewhat different weights. We revised the text accordingly. On page 5: “Analysis of mutational signatures assigned the majority of SSNVs to clock-like signatures (SBS1, SBS5, SBS40). Overall the next abundant signature was SBS18, associated with reactive oxygen species³² and frequently found in neuroblastoma³³, followed by SBS3, associated with failure of homologous recombination in dividing cells (Fig. 1e). Moreover, we inferred mutational signatures separately for clonal and subclonal mutations, classified according to their variant allele frequencies (VAFs), finding that the above signatures overall occur at similar frequencies clonally and in subclones (Extended Data Fig. 1e).”

To check the robustness of our inference results, we now performed all inferences in addition with the SSNV count of clock-like signatures only, finding agreement with the results obtained by using all SSNVs (also in response to a comment by Reviewer #2). The parameter inference based on the subset of clock-like mutations yields practically identical results, now shown in new Extended Data Figure 5c and d. We added (page 12): “The model fit of Fig. 7 was based on all SSNV, which assumes that the corresponding mutational processes do not change strongly during neuroblastoma evolution (as suggested by Extended Data Fig. 1e and 2d). To test the robustness of our inference, we also performed all analyses with the subset of clock-like SSNVs as input. The inferred rate of clock-like SSNV acquisition (μ) was 2.3 (1.2, 2.3; 80% credible interval) per division (corresponding to 2.2 ± 0.3 SSNVs per day). All other inferred parameters remained practically unchanged (Extended Data Fig. 5c, d). Hence, confining the analysis to clock-like mutations corroborates the real time calibration of ECA and MRCA.”

It could be worthwhile to probe more deeply into the differences of mutation frequencies between individual samples, taking into account recent insights on how to assign proper weights (e.g., Jin/Park et al. BioRxiv, <https://doi.org/10.1101/2022.04.21.489082>). We feel that this would overload the manuscript and may be a separate study.

- [Evolutionary classes] The detection of MRCAs closer to sampling (in relapses) makes sense; please provide some details for non-NB expert of what is the therapy adopted here.

All patients were treated according to the German NB2004 protocol. Briefly, post resection patients are assigned to one of three groups: observation, medium-risk and high-risk (noted in Supplementary Table 1 as observation, MRG and HR). Patients in the first group are simply observed after the surgery. Medium-risk patients receive 6 cycles of chemotherapy, followed by radiotherapy if residual tumor is detected and, for all patients, treatment with retinoic acid. High-risk patients are treated with a different protocol that involves very intensive multimodal chemotherapy, followed by autologous stem cell transplantation. Relapses occurred from all three groups. As the tumor will grow back after resection from a small subset of tumor cells that remained in the patient, the MRCA will be dated later. As the number of relapse samples is limited, and the focus of the manuscript is on the primary disease, we did not go into further detail here. We edited the corresponding text (page 6/7): “By contrast, the timing of the MRCA, defining the origin of a tumor sample, was significantly later in relapsed tumors – consistent with a bottleneck imposed by incomplete tumor resection, in some cases further cytotoxic therapy (Supplementary Table 1), and eventually regrowth of the tumor from a small number of surviving cells (Extended Data Fig. 2b and c).”

- [Evolutionary classes] How is the cut early/late MRCAs defined in Fig 3a/4c?

We detected a clear bimodality in the histogram of the SSNV density at MRCA in the discovery cohort (Fig. 3a) and took the upper bin border just before the minimum as the cut between early and late MRCA (0.05 SSNVs per Mb). As an alternative criterion for choosing the cut point, we asked at which value we best discriminate the high-risk cases and found the same value (two significant digits). When we used this value in the validation cohort (Fig. 4c), we found that it also marked the end of a clear peak of tumors with early MRCA. We added in Methods, Survival analysis (page 24): “Survival analysis was performed using the R-package survival v 3.1.12⁶¹. We detected a clear bimodality in the histogram of the SSNV density at MRCA in the discovery cohort and took the upper bin border just before the minimum, 0.05 SSNVs/Mb, as threshold to split tumors into groups with early and late MRCA. The same value was found appropriate in the validation cohort.”

We will be conducting further work addressing the clinical utility of MRCA timing as an additional parameter for patient stratification; here we will look at thresholding in a larger cohort which we are currently assembling. This larger cohort will give us better statistical power to choose a threshold that discriminates best the high-risk cases.

- [Telomere] In some cases you identify telomere maintenance gained between the ECA and the MRCA; would you think this could cause an increased growth rate or better a decreased

death rate in the tumour? Can you comment? In general, can these events be acquired with any temporal order relative to CNAs? If yes, do you have evidence that orders are commutative?

We agree, gained telomere maintenance should cause a decrease in tumor cell death rate. This is precisely how we implemented the selective advantage of the oncogenic events.

The effect on net growth rate, and especially the comparison with early-MRCA tumors that typically have no gained telomere maintenance mechanism, may not be straightforward. We estimate that the death rate relative to the proliferation rate is lower in early-MRCA neuroblastomas (Fig. 7h). Hence, the longer evolving, late-MRCA tumors may intrinsically be more prone to cell death. Speculative reasons for this could be that early-MRCA tumors better preserve the cell survival mechanisms characteristic of highly proliferative progenitors (including superior telomere maintenance), or that additional oncogenic hits in late MRCA tumors make them also vulnerable to cell death. Therefore, late-MRCA tumors may also be under higher selective pressure to acquire additional mechanisms to stabilize their telomeres. With the current data, we cannot accurately deduce the temporal order of acquiring telomere maintenance mechanisms, stating in the manuscript (page 8): “Hence, for these tumors, we cannot time an early genetic event which, however, leaves open the possibility that small-scale mutations, chromosomal losses or high-level amplifications (>4 copies), neither of which can be timed reliably, preceded the MRCA of the tumor.”

G

Appropriate credit to previous work is given even though some things are re-implemented while I feel they could have been done with standard methods (e.g, Binomial deconvolution, multiplicity phasing etc). The authors might clarify why published methods could not be used.

Please refer to Point 8 above; published methods we have used for comparison or further analysis (CNAqc, Mobster) have been cited.

H. See F.

Reviewer #2:*Remarks to the Author:*

Körber et al perform an in depth analysis on the somatic evolution of neuroblastoma by analysing a discovery cohort of cases

With deep WGS data (~80x), n=100. 67 of the samples where from initial diagnosis (7 of these from metastatic lesions). The rest (n=33) from relapse tumors. They integrate data on somatic copy number alterations and somatic SNVs to estimate when specific genetic alterations occurred, and use mathematical models of neuroblastoma tumorigenesis and approximate bayesian computation to infer time points during the patients life when genetic alterations should have arisen. They then validate their findings in an additional cohort of patients, sequenced to an average of 30x. These findings are novel and adds to our understanding of neuroblastoma evolution.

The authors should be commended for this very clear article and the work presented should be of broad interest to researchers working in both Cancer genomics, tumor evolution and also to clinicians treating patients with Neuroblastoma. Furthermore the code is published on GitHub and the data deposited in repositories for access by other researchers. Especially the result that late MRCA tumors tend to have a poorer prognosis is intriguing and while Körber et al note that this group is also enriched for tumors with segmental copy number alterations and thus might be more genomically unstable, one wonders whether there are other factors (microenvironmental?) that predispose these tumors to evolve for a longer time.

All in all, the paper and the results presented in it are well described and backed up by the data and careful analysis, and my recommendation is minor revision.

Anders Valind MD, PhD

Department of Clinical Genetics Lund University

We thank Dr. Valind for the careful reading of our manuscript, his very supportive evaluation and the excellent suggestions. We have addressed the reviewer's concerns as detailed in the following.

Questions/Comments:

** While Körber et al describe how they guarded against misclassification of subclonal somatic*

SNVs as clonal ones in my mind there is still some concern about whether the classification of SCNAs as clonal or sub clonal is robust. As it has been shown that analysis using only a single tumor sample may miscall sub-clonal SCNAs as clonal (as we and others have shown: c.f PMID: 29867221, PMID: 32041836 and PMID: 34815394). The authors use of experimentally derived ploidy is helpful here (constraining the number of (purity, ploidy)-pairs that fit the data.) In my mind, this should be further discussed/analyzed, for instance, for the subset of primary/relapse-pairs, how did the SCNAs differ between the samples?

Dr. Valind makes the excellent point that subsampling may lead to mistaking subclonal somatic mutations as clonal. He acknowledges that the corrections for SNVs we make to account for this subsampling effect are appropriate and enquires about SCNAs. Regarding CNAs, differences between primary and relapse samples could occur when in the relapsed tumor these regions have further evolved from the CNA in the primary tumor. In this case, the CNA could have been truly clonal in the primary tumor (e.g., a trisomic chromosome) but further evolution (e.g., another chromosomal gain) could have rendered the original CNA subclonal or even made it disappear if the additional event is selected. Alternatively, a subclonal CNA in the primary tumor could have erroneously been classified as clonal by ACEseq, mainly due to subsampling. We have further analysed our data in this regard.

In the two primary-relapse sample pairs available in our cohort, we observe broad agreement of the copy number profile but also a few differences (top panels of Figures I and II for the reviewer, below). The copy numbers of regions with differences are different whole numbers in the two samples of a pair (regions marked by boxes with capital letters in Figures I and II), except for a small region of Chr. 7 that appears to be partially lost in the relapse sample.

Figure I for the Reviewer. Evolution of SCNAs between primary and relapsed tumor. Shown are the copy number profiles of the primary-relapse pair NBE11/NBE66 (top panel). Capital letters highlight differences between the two samples of the pair (focusing on the segments with copy number ≤ 4 considered by our timing approach). For each region, a two-dimensional scatter plot compares the VAF distribution of SNVs in primary and relapsed tumors (lower panels; dashed lines indicate clonal VAFs on different copy numbers and circles highlight amplified clonal SNVs). Cartoons below each scatter plot illustrate the temporal evolution of the respective SCNA suggested by the timing analysis on the respective samples.

Figure II for the Reviewer. Evolution of SCNAs between primary and relapsed tumor. Shown are the copy number profiles of the primary-relapse pair NBE51/NBE78 (top panel). Capital letters highlight differences between the two samples of the pair (focusing on the segments with copy number ≤ 4 considered by our timing approach). For each region, a two-dimensional scatter plot compares the VAF distribution of SNVs in primary and relapsed tumors (lower panels; dashed lines indicate clonal VAFs on different copy numbers and circles highlight amplified clonal SNVs). Cartoons below each scatter plot illustrate the temporal evolution of the respective SCNA suggested by the timing analysis on the respective samples.

This observation is compatible with clonality of these genomic regions in both primary and relapse tumor, with mutation and selection leading from the CNA in the primary tumor to the altered CNA in the relapse. To ask whether this could be the case, we visualized the two-dimensional VAF distribution for each diverging copy number variant in these regions (lower panels A-C in Figure I and A to D in Figure II); we focused on the segments with copy number ≤ 4 considered by our timing approach. We observed that amplified clonal SSNVs were shared between primary and relapse but occurred at different frequencies (marked by circles in the respective panels). As shown in the corresponding cartoons, this finding can be readily explained by further alleles/segments gained or lost on the trajectory to the relapse tumor, consistent with the presence of genomic instability in both samples. We conclude that the copy number changes between primary and relapse tumor do not show an indication of subclonality being mistaken as clonality in a limited sample but could result from ongoing evolution.

As there were no relapse tumors for the majority of the samples, we of course cannot exclude that some subclonal SCNAs in the cohort that were erroneously classified as clonal. The SSNV density on such CNAs would be larger than that of the MRCA. As a safeguard, we tested for each segment whether its SSNV density is compatible with the SSNV density of the MRCA. Across the cohort, we found a very small number of segments, in five primary tumors (out of 153) and two relapse tumors (which did not enter timing analysis of the MRCA), that had apparently clonal SSNV densities significantly larger than that of the MRCA. We excluded these putative subclonal segments from further analysis. This step was not clear from the Methods and we now added on page 22: “here we excluded a small number of segments in seven tumors with a larger density of amplified clonal mutations than the estimated mutation density at the MRCA, because such gains may be subclonal CNAs that were erroneously classified as clonal.” To emphasize the relevance of multiregion sampling, we added to the Discussion on page 14, where we now cite PMID: 29867221 and PMID: 32041836 in addition to the already cited PMID: 34815394: “Continued evolution of such tumors has also been noted in a recent study taking multi-region biopsies⁴⁵, emphasizing the potential of spatially resolved genetic and transcriptomic analysis^{46,47}.”

In addition, how far apart in the biopsy was the material used for WGS from the material that was used for ploidy-analysis?

All resected tumors were processed according to a standard protocol (German NB study protocol). The tumor is divided into 4 quadrants, all of which are evaluated histologically. The cross-section of one quadrant is used for WGS; the same quadrant is used for ploidy analysis. We added this information now to the Methods (page 17): “For analysis, all resected tumors were divided into 4 quadrants, all of which were evaluated histologically. MYCN status was assessed as routine clinical marker for all tumors using FISH (Fluorescence in situ hybridization). A cross-sectional slice of one quadrant was used for DNA extraction for WGS; the same quadrant was used for ploidy analysis, measuring the DNA index.”

** It is unclear from the manuscript whether they used all somatic SNVs in their timing analysis, or just the ones inferred to be due to ‘Molecular clock’-like signatures (AC1 and AC2 using the nomenclature from the authors). This should be clarified, and if all sSNVs were used, how did the authors test the assumption of ~constant mutation rate for AC3 (Cosmic signature #18 - where the proposed causal mechanism is reactive oxygen species?).*

The reviewer asks for clarification on which SNVs were used for timing analysis. He also raises the important point that non-clock-like mutational signatures may have been acquired during tumor evolution and hence distort the real time calibration. In the original version of the model-based inference, we used all SSNVs as input. This was based on the observation that clonal and subclonal mutational signatures are very similar (see below). Following the above suggestion, we have now repeated all computational analysis with only clock-like SSNVs as input. The parameter inference based on this basal subset of mutations yields practically identical results, now shown in new Extended Data Figure 5c and d. We added the requested clarification and described our new results as follows (page 12): “The model fit of Fig. 7 was based on all SSNV, which assumes that the corresponding mutational processes do not change strongly during neuroblastoma evolution (as suggested by Extended Data Fig. 1e and 2d). To test the robustness of our inference, we also performed all analyses with the subset of clock-like SSNVs as input. The inferred rate of clock-like SSNV acquisition (μ) was 2.3 (1.2, 2.3; 80% credible interval) per division (corresponding to 2.2 ± 0.3 SSNVs per day). All other inferred parameters remained practically unchanged (Extended Data Fig. 5c, d). Hence, confining the analysis to clock-like mutations corroborates the real time calibration of ECA and MRCA.”

** In their survival analysis the authors dichotomise the sSNV/mb parameter (to define early vs late MRCA) and refer to the bimodal distribution of sSNVs/Mb. This analysis should be repeated with sSNV/mb as a continuous variable, as dichotomising typically leads to loss of information (see <https://www.ncbi.nlm.nih.gov/pmc/articles/PMC1458573/>)*

We agree that this is an interesting suggestion. After careful discussion with clinical partners, we decided to stay with implementing MRCA timing as a binary variable (early versus late) for the following reasons:

- Our results suggest that MRCA timing is worth following up as a new parameter for patient stratification in clinical trials. Hence individual cases need to be classified by a standard protocol. We believe that this will be more easily realized with a robust binary variable.
- In particular, we tested binary MRCA timing in the context of other novel predictors of outcome (suggested by Reviewer 3) and found the former to be the leading predictor of both event-free and overall survival (new Figure 4j, new Extended Data Fig. 4).
- The distribution of MRCA times is clearly bimodal. Hence there is a natural threshold for classification. This situation is different from imposing a threshold onto an essentially unimodal distribution. The interesting article referred to by the reviewer cites examples such as hypertension, obesity and blood cholesterol, which do not show pronounced bimodality across populations of interest.

** Was there any pattern in when specific genetic alterations (e.g. +17q) typically occurred?*

We summarized the timing of characteristic genetic alterations in Fig. 3c and 4e. Indeed, if timeable, segmental gains occurred early, coincident with the ECA, in the vast majority of cases where we could recognize and ECA. We added (page 10): “In the late-MRCA tumors with timeable ECA, the vast majority of the segmental gains that we could time were coincident with the ECA (Fig. 3c and Fig. 4e, dark green squares in the annotations of gains and losses).” In a small number of cases, the gain occurred in the MRCA (light green squares), while the remaining cases it could not be uniquely timed, e.g., when the gain occurred at copy numbers > 4 (grey squares).

Reviewer #3

Körber and colleagues from the Höfer and Westermann groups report about somatic evolution of the childhood tumor neuroblastoma in early fetal development calculated and quantified by an innovative bioinformatics/systems biology approach consisting of deep whole-genome sequencing, molecular clock analysis and population-genetic modeling in a discovery cohort of 67 primary neuroblastomas and a validation cohort of 86 tumors. Their interesting major new findings include

a) start of neuroblastoma development occurs via aberrant mitoses between week 4 and 12 of pregnancy (this has important implications for future prevention strategies and epidemiological/environmental studies!)

b) neuroblastomas with favorable prognosis cease to evolve early, whereas aggressive neuroblastomas show prolonged evolution during which they acquire telomere maintenance mechanisms and

c) duration of evolution is an accurate predictor of outcome

These data will indeed be of high interest and very useful to the neuroblastoma research community by significantly advancing current insights into neuroblastoma development and evolution. It provides the bioinformatical explanation and rationale of biological and clinical phenomena previously observed in patients and their tumor material. In addition, it harbors a high potential for clinical relevance as a novel predictive biomarker.

However, the claim that the findings may prospectively guide treatment decisions with MRCA timing as a reliable and superior outcome predictor needs some further substantiation (see below).

We thank the reviewer for the careful reading of our manuscript, their very supportive evaluation and the excellent suggestions, especially to further scrutinize the potential for clinical application. We have addressed the reviewer's concerns as detailed in the following.

1. How do the authors explain the discrepancy that MRCA timing was a significant predictor of overall survival in the validation cohort (86 samples), but not in the discovery cohort (67 samples)?

The reviewer asks why early versus late MRCA separated cases with, respectively, long and short, overall survival in the validation cohort but did not do so significantly in the discovery cohort. While revising the manuscript, we detected an unfortunate, inadvertent numerical error in the clinical data file for the discovery cohort. After correcting this error, the above-mentioned discrepancy between discovery and validation cohort does no longer exist – the MRCA is a

significant predictor of overall survival in both cohorts. We sincerely apologize for this inadvertent error and thank the reviewer for their question that helped us detect it.

In more detail, we re-checked all the primary data on which our analyses were based and discovered an unfortunate error that was caused by copying the clinical annotation from the original Excel file into an input file for computational survival analysis. There the clinical annotation for the discovery cohort was shifted by one row for a subset of 88 tumors (as tumors were roughly sorted by survival, this error did not cause a gross distortion and went undetected in the initial version of the manuscript). We now corrected this error. As a result, the survival predictions based on the MRCA improve both for event-free survival (shown in corrected Figure 4a) and for overall survival (shown in corrected Figure 4b). We changed the corresponding text on page 8 as follows: “Remarkably, early MRCA timing clearly identified cases with long event-free survival (Fig. 4a) and long overall survival (Fig. 4b), suggesting that MRCA timing in primary neuroblastoma samples may help predict outcome. ~~The prediction for overall survival was not significant (Fig. 4b).~~”

2. To justify a future effort of calculating MRCA timing for individual patient tumors as a new outcome predictor, the superiority and independence of MRCA timing should be shown in comparison to more recently identified and meanwhile clinically used molecular outcome predictors such as established mRNA classifiers (Oberthuer et al, JCO 2010 and Clin Can Res 2015; Rosswog Neoplasia 2017; Volland Genome Biol 2015) or RAS/p53 pathway mutations (Ackermann et al, Science 2018), preferably for the whole patient cohort (if these data are available) or, if this is not feasible, for a subcohort of patients. The comparison with the old clinical classification (age, stage, MYCN amplification) in the trial NB2004 and the telomere maintenance status alone is interesting, but less informative. If these comparative data are not available at all, the need of this validation step has at least to be included into the discussion.

The reviewer proposes to further test the performance of MRCA timing as a potentially informative variable for outcome prediction. Specifically, the reviewer asks us to consider, beyond clinical variables, recent molecular classifiers (mRNA classifier and RAS/p53 pathway mutations). We agree that this is a highly relevant question and performed the requested new survival analyses. The key result is that MRCA timing is an informative predictor of both event-free survival and overall survival. Specifically, inclusion of MRCA timing renders age, stage and gain of telomere maintenance (including *MYCN* amplification) non-significant as predictors of both event-free survival (new Figure 4j) and overall survival (Extended Data Fig. 4) in our cohort. This finding indicates that MRCA timing contains information that is not present in

current classifier variables. Moreover, MRCA timing was the sole significant predictor of event-free survival when the mRNA classifier was included (Figure 1a for the Reviewer). Overall survival was significantly predicted by the mRNA classifier while MRCA timing had the highest mean Hazard ratio but was not significant (Figure 1b for the Reviewer). These findings suggest that MRCA timing could even be informative if the mRNA classifier were included in patient stratification.

In more detail:

- As the standard in the field, we used Cox regression analysis to compare individual variables with respect to predicting outcome for the entire cohort. The cohort size allows for comparison of up to 5 variables. We chose the following layout: MRCA timing + age and stage (present worldwide in classification schemes) + gain of telomere maintenance mechanisms (TMM, which improves on the currently used variable, *MYCN* amplification, in classification schemes worldwide) + one of the more recent molecular variables, either RAS/p53 pathway mutations or mRNA classifier. We note that in the first version, we included NB2004 – a multivariate classifier specific to Germany – as a single variable in Cox regression. Upon consultation with statisticians, we now compare only single variables, including key variables used in NB2004 and many other classification schemes worldwide.
- Cox regression ranks the different explanatory variable according to the information they contain on survival. MRCA timing emerges as the best predictor of event-free and overall survival in our cohort when compared with age, stage, TMM, and RAS/p53 pathway mutation. now shown in new Figure 4j and Extended Data Fig. 4. We emphasize that the other variables are not as such poor predictors but, as single variables, contain less information on survival than MRCA timing.
- When including the mRNA classifier in place of gain of telomere maintenance and RAS/p53 pathway mutations, MRCA timing was the sole significant predictor of event-free survival (Figure 1 for the reviewer). Overall survival was significantly predicted by the mRNA classifier (while MRCA timing had the highest mean Hazard ratio but was not significant). We analysed the mRNA classifier in the most current version that is being prepared for entering clinical trial in Germany (with the help of Roma Kurilov and Benedikt Brors at the DKFZ, who are leading the bioinformatics of the classifier and have now been included as co-authors; Oberthuer et al., *Clinical Cancer Research*, 2015). As mRNA data were available only for a subset (131 of 153) of primary tumors, this could be done only on these tumors. The data available to us were bulk RNA-seq, whereas the classifier was developed for microarray data. It is not clear to what extent this may have compromised the performance of the classifier. For these reasons, we show these data here as a Figure for the reviewer. Alternatively, however, we would be

happy to include them in the manuscript upon advise by the reviewer.

Taken together, these new analyses substantiate that MRCA timing is worth being considered as a parameter for patient stratification. We edited the corresponding paragraph in Results as follows (page 9): “To compare MRCA timing quantitatively with other predictors of survival, we considered clinical variables used worldwide (stage, age), gain of a telomere maintenance mechanism (which improves on the clinically used criterion, MYCN amplification³¹), and a more recently proposed molecular predictor, the mutation status of the RAS/p53 pathway³⁰. Early MRCA timing emerged as the most informative predictor of event-free survival (Fig. 4j); overall survival was best explained by both MRCA timing and mutations in the RAS/p53 pathway (Extended Data Fig. 4a).

We also added to the Discussion (page 15): “In our combined discovery and validation cohort, MRCA timing was an informative predictor of both event-free and overall survival when tested in combination with a comprehensive set of predictor variables, including age, stage, gain of telomere maintenance³¹, and mutation status of the Ras/p53 pathway³⁰. Hence, MRCA timing, which can be robustly obtained from standard WGS, may be worth considering as a further parameter for patient stratification.”

Figure 1 for the Reviewer. Multivariate survival analysis with Cox-regression considering MRCA timing, acquired mechanisms of telomere maintenance, stage, age at diagnosis and mRNA classifier. Survival analysis was run on a subset of 131 tumors for which RNAseq data was available. Shown are mean hazard ratio, 95% confidence intervals and p-values for each variable (Wald test) for event-free survival (a) or overall survival (b).

3. The mutation rate in late MRCA/high-risk/segmental gain neuroblastomas might be much higher than in low-risk tumors. Indeed high-risk tumors have a much higher expression of DNA repair genes. The presence of oncogenic gene defects and segmental defects causes an entirely different type of tumor cell, with potential effects on mutation rate. This would inflate the number of mutations between ECA and MRCA and place the ECA erroneously early in development.

The reviewer asks whether potential differences in mutation rate between high-risk and low-risk neuroblastomas could affect ECA timing. We agree that this is an important question and performed further analyses. These additional analyses support our finding that neuroblastomas, both with early and late MRCA, can begin to develop as early as in the first trimester of pregnancy, as detailed in the following:

There are two aspects of timing: molecular clock and relating the clock to real time. We consider both aspects in turn.

Molecular clock: The timing of the ECA by the molecular clock is independent of the mutations accumulated between ECA and MRCA. The ECA is timed by SSNVs that have occurred before the copy number gains and losses happen (most likely during normal neurogenesis). Hence, the conclusion that late-MRCA (high-risk) tumors start to develop in the same time window as early-MRCA (low-risk) tumors is independent of the mutation rate in high-risk tumors. Theoretically, it is possible that even prior to the ECA, the rates of SSNV accumulation differ between cells that will give rise to late-MRCA tumors compared to cells giving rise to early MRCA tumors. However, a higher mutation rate in the former would cause the real-time estimate for the ECA of late-MRCA tumors to be earlier than the ECA of early-MRCA tumors.

For relating the clock to real time, we estimate the rate at which SSNVs occur. In the manuscript, this rate is estimated from late-MRCA tumors. It is not clear whether the presence of segmental defects in these tumors causes the rate of acquiring SSNVs to increase. To test this, we confined our timing inference to clock-like SSNVs only. First, we note that correcting the data copying error (Point 1) also caused a slight shift of the ECA and MRCA real-time estimates (now corrected in Figure 7c, d); the conclusion that a substantial fraction of tumors has their ECA at the end of the first trimester of pregnancy remains unaltered. To test for robustness, we first updated the calling of mutation signatures and show the result in the new Figure 1e (requested by Reviewer #1). These new data show that in primary tumors – both high-risk and low-risk – clock-like mutations are abundant. The next most common signature is SBS18, which is typical for neuroblastoma and of not completely clear etiology (possibly damage caused by reactive oxygen species). A smaller number of tumors (among them many relapse tumors) also show evidence for signature SBS3, associated with defective homologous recombination-based DNA damage repair (which manifests itself predominantly as small indels and genome rearrangements rather than SSNVs). We find that the neuroblastoma related signature SBS18 is more prominent in late MRCA tumors, both at clonal and subclonal level (new Extended Data Figure 2d, e). Hence, focusing on clock-like SSNVs removes a potential bias by SBS18 mutations. Fitting the population-genetic model to the cumulative distributions of SSNV densities at ECA and MRCA, defined only by clock-like SSNVs, yielded practically identical results to our previous fit using all SSNVs (new Extended Data Fig. 5c; obviously, the single difference is in

the mutation rate μ , which now only counts clock-like SSNVs). As a result, we also obtained similar real time estimates of ECAs and MRCAs (new Extended Data Fig. 5d).

We revised the text accordingly. On page 5: “Analysis of mutational signatures assigned the majority of SSNVs to clock-like signatures (SBS1, SBS5, SBS40). Overall the next abundant signature was SBS18, associated with reactive oxygen species³² and frequently found in neuroblastoma³³, followed by SBS3, associated with failure of homologous recombination in dividing cells (Fig. 1e). Moreover, we inferred mutational signatures separately for clonal and subclonal mutations, classified according to their variant allele frequencies (VAFs), finding that the above signatures overall occur at similar frequencies clonally and in subclones (Extended Data Fig. 1e).”

On page 7, we added: “Of note, overall the majority of SSNVs in both early-MRCA and late-MRCA tumors were clock-like while SBS18, characteristic of neuroblastoma, was more abundant in late-MRCA tumors (Extended Data Fig. 2d, e).”

On page 12, we added: “The model fit of Fig. 7 was based on all SSNV, which assumes that the corresponding mutational processes do not change strongly during neuroblastoma evolution (as suggested by Extended Data Fig. 1e and 2d). To test the robustness of our inference, we also performed all analyses with the subset of clock-like SSNVs as input. The inferred rate of clock-like SSNV acquisition (μ) was 2.3 (1.2, 2.3; 80% credible interval) per division (corresponding to 2.2±0.3 SSNVs per day). All other inferred parameters remained practically unchanged (Extended Data Fig. 5c, d). Hence, confining the analysis to clock-like mutations corroborates the real time calibration of ECA and MRCA.”

We add that key parameters estimated with our population genetic model are consistent with independent experimental evidence. First, regarding the somatic SNV rate (page 12): “The inferred rate of SSNV acquisition (μ) was 3.2±0.4 SSNVs per day, which falls in the same range as a recent direct measurement of somatic mutation rate in the developing central nervous system [5.1 (1.5, 9; 95% CI) SSNVs per day]⁴¹.” Note that our inference was corrected compared to the previous version, as a result of correcting for the data copying error described in Point 1 above. Second, our inferred cell-cycle times of neuroblastoma cells of 2 to 5 days (Figure 7h), which rely on the correct inference of the mutation rate, were consistent with experiments determining the division rates of neuroblastoma cells *in vivo*, yielding an average of 4.2 days (done by giving the patients Colcemid to arrest cells in mitosis before resecting the tumor; Aherne and Buck, Br J Cancer 1971). We added on page 13: “our estimated cell division rates agree quantitatively with rates measured in neuroblastoma *in vivo*.⁴²”

4. High risk neuroblastoma, correlating with late MRCA/early ECA tumors, are diagnosed at a much higher age than low-risk tumors. If they have a longer distance in time to the ECA, the ECA of high risk tumors may still be much later than the ECA of low risk tumors. I did not notice that the effect of the difference in age at diagnosis was included in the analyses, modelling or discussion.

The reviewer asks about the impact of age of diagnosis on our analysis. We agree that this is a pertinent question. Our analysis based on somatic mutations and age at diagnosis are related, as expected by the reviewer, in that MRCA timing by mutation density is positively correlated with age at diagnosis (Figure II for the reviewer below). Here one also notices effects of specific oncogenes, as ALT neuroblastomas grow more slowly than MYCN-amplified ones.

Figure II for the reviewer. Relation between SNV density at MRCA (y-axis) and the age at diagnosis (x-axis) for

primary tumors/ metastases. Age at diagnosis and SNV density at MRCA are positively correlated ($r=0.82$, Spearman's correlation coefficient). TMM, gain of telomere maintenance mechanism.

Regarding the second point, ECA timing in tumors with early or late diagnosis, our method of ECA timing in terms of SSNV density is independent of the age at diagnosis. We find that the

SSNV density at ECA is indistinguishable between early and late MRCA neuroblastomas. As this density is determined by mutations occurring before the initial transforming event (whole-chromosome or segmental gains and/or losses), our ECA timing by mutation density is unaffected by the details of subsequent tumor evolution. Hence, we respectfully disagree with the reviewer: when we can time an ECA in late-MRCA/high-risk tumors (which is the case in 55% of late-MRCA tumors) it is overall not later than the ECA of early-MRCA/low-risk tumors. We clarified this point on page 6: “We timed the MRCAs for all tumor samples and the ECAs associated with chromosomal gains, via the densities of the SSNVs associated with the two types of cell.” Of course, this leaves 45% of late-MRCA/high-risk tumors, in which we cannot time an early event, and hence their time of origin is not clear. This is stated on page 8 of the manuscript.

For converting the SSNV density estimates into real time, the time periods during which the tumors grew until diagnosis is relevant, and at this specific point the age at diagnosis is used in the mathematical modeling. This is now clarified in new Figure 6. We also added in the Discussion on page 13: “In this paper, we timed genetic events in the evolution of neuroblastoma using the molecular clock of SSNV accumulation and, inferring the rate of SSNV acquisition from the distribution of variant allele frequencies, related this clock to real time by factoring in the age at diagnosis.”

5. Furthermore, the authors use gained chromosomal areas to infer ECA and MRCA. That implies that in triploid cases, they use the entire genome to count SSNVs, while in near-diploid cases, they necessarily use the typically gained regions, probably #7 and #17q. A uniform analysis, focussing on the q arm of #17 which is gained in almost all neuroblastoma, would avoid this possible bias.

The reviewer asks whether there is bias in comparing SSNV densities on genomic regions of different length. This is a key point when asking whether there is an ECA identifiable prior to the MRCA. Our analysis has no potential length bias: (1) We used all clonal SSNVs in the genome to time the MRCA. (2) To time the ECA, we used the entire gained parts of a given genome and determined the densities of amplified clonal mutations for each amplified segment. This is done to use all available data for getting the best statistical discrimination possible. (3) For each gained segment, we tested whether the count of amplified clonal mutations is consistent with being drawn from the mutation count of the MRCA according to a negative binomial distribution (the standard model for mutation density across the genome); this test factors in the length of each segment and hence removes length bias. The details are given in Methods (section Mutation timing. (1) Estimating the number of amplified and non-amplified clonal mutations) and

include correction for multiple testing. In this way, (i) we obtain a p-value for each amplified segment stating whether the timing of amplification is at MRCA (null hypothesis) or earlier (p value, corrected for multiple testing, < 0.01), and (ii) we also compute a confidence bound for the mutation density of each segment that depends on its length. In this way, we can decide for each clonal segment whether it was duplicated/amplified prior to the MRCA or at the MRCA. Specifically, this procedure allows us to also identify mixed cases where some segments were gained at an ECA and others later (for an example, see Extended Data Fig. 3c). To clarify, we amended the text on page 6: “Next, we asked whether clonal chromosomal gains occurred around the time when the MRCA arose or ancestral to the MRCA. To this end, we quantified SSNVs that were clonal on two copies on trisomic and tetrasomic segments (termed amplified clonal SSNVs⁸; Fig. 2b). These mutations were acquired prior to a gain on the respective allele (Fig. 2c, dark green), and their density thus scales with the time at which the chromosomal gain occurred^{8,34}. We compared these densities to the densities at MRCA based on a negative binomial distribution of clonal SSNVs across the genome (Methods). In the example tumor, nearly all gains, except for Chr. 9 and 20q, had a mean density of 1 amplified clonal SSNV per 100 Mbp, which is significantly smaller (adjusted p value < 0.01) than the mutation density of the MRCA (Fig. 2e). Hence, the molecular clock places these gains ancestral to the MRCA, in an early common ancestor (ECA) of the tumor.”

6. The authors have validated their data in the context of temporal evolution (first diagnosis versus relapse samples), but not in the context of potential spatial evolution. This has to be discussed in light of the future potential/need of spatially resolved transcriptomics technologies.

This point is well taken and now discussed on page 14: “Continued evolution of such tumors has also been noted in a recent study taking multi-region biopsies⁴⁵, emphasizing the potential of spatially resolved genetic and transcriptomic analysis^{46,47}.”

Minor: A tumor cell content of 88% in average seems surprisingly high considering the fact that all tumor stages have been included. Can the authors explain and prove how this was assessed?

Tumor cell content was estimated according to standard procedures for the analysis of neuroblastoma WGS data, using ACEseq. As a confirmation, the VAF distributions provide also an estimate of TCC, which agreed well with the ACEseq estimates. Only in one case, out of all 186 tumors, was it necessary to adjust the ACEseq estimate of TCC upon visual inspection of the VAF distribution (NBE40, as mentioned in Methods on page 17). The VAF distributions at all

copy numbers are now shown for each tumor in a separate datasheet available on Mendeley (available for preview by the reviewers).

Decision Letter, first revision:

29th Nov 2022

Dear Thomas,

Thank you for submitting your revised manuscript "Genetic origin of neuroblastoma in early fetal development" (NG-A60512R). It has now been seen by the original referees and their comments are below. The reviewers find that the paper has improved in revision, and therefore we'll be happy in principle to publish it in Nature Genetics, pending minor revisions to satisfy the referees' final requests and to comply with our editorial and formatting guidelines.

****As the current version of your manuscript is in a PDF format, please email us a copy of the file in an editable format (Microsoft Word or LaTeX)-- we can not proceed with PDFs at this stage.****

Sincerely,

Michael Fletcher, PhD
Senior Editor, Nature Genetics

ORCID: 0000-0003-1589-7087

Reviewer #1 (Remarks to the Author):

I was happy to read through this revised version because I think that the authors did a good work replying all the reviewer's comments.

Regarding my particular questions, I find that all the replies are indeed satisfactory. The overall paper I think improved largely, especially in clarifying the modelling approaches and the logic followed. The organisation of the GitHub is now very clear and I do believe that this will make it easy to adapt these analyses to other cohorts as well. I praise the authors for following this suggestion.

I have no other reservations and indeed suggest to accept the current version of this paper.

Giulio Caravagna
University of Trieste

Reviewer #2 (Remarks to the Author):

The authors have convincingly showed that on the level of SCNAs the genomes are stable between primary and relapses for the two cases in their cohort where they had paired samples (within the limitations inherent in that comparison - bottlenecks, selection etc). They are also quite correct in that when samples differ can be hard to really know whether (especially for whole chromosome gains) the SCNA is really subclonal or whether it was clonal and then lost in a subclone. They furthermore validate this by comparing the sSNV density for each segment within a tumor to the sSNV density for the MRCA, and reassuringly found densities compatible with subclonal SCNAs within a small fraction of samples. In total, I am now convinced that the vast majority of SCNAs reported by Körber et al are indeed clonal and any residual subclonal SCNAs should have at maximum minimal effect on the conclusions.

Körber et al have also clarified the relationship between the sample material used for WGS and the material used for measuring ploidy in a satisfying manner, reassuring this reviewer that they are indeed from closely located samples. In my mind this also adds another layer of security in the classification of clonal vs subclonal SCNAs (by constraining the parameter space for (purity, ploidy) solutions when calling absolute copy numbers).

Regarding my question on mutational signatures the authors have repeated the analysis using only clock-like signatures and this encouragingly shows concordant results with using all somatic SNVs. They are also quite right that dichotomising a bimodal distribution typically doesn't lead to a dramatic loss of information and I can also appreciate that a binary classification would facilitate more prompt implementation in clinical trials, so I am now satisfied with the use of the dichotomised variable in their survival analysis.

While Supplementary Table 2 contains all mutational data on the discovery cohort I was unable to find the mutational data for the validation cohort as well as the copy number data from ACEseq (for both discovery and validation). these should be part of the supplemental data as somatic mutations and SCNAs doesn't identify the patient while their availability increases the utility of the article also as a data resource. The revised article was a delight to read and I'm convinced that the findings will be important both to the neuroblastoma field but also more broadly to the tumor evolution field.

The authors have now answered all of my questions in a satisfying manner and when the datasets mentioned above are added to the supplement my recommendation is acceptance of the article.

Sincerely
Anders Valind
MD, PhD
Department of Clinical Genetics, Lund University, Sweden

Reviewer #3 (Remarks to the Author):

From my perspective all reviewer comments have been appropriately addressed and I recommend the paper for publication.

Author Rebuttal, first revision:

Point-by-point reponse

Reviewer #1 (Remarks to the Author):

I was happy to read through this revised version because I think that the authors did a good work replying all the reviewer's comments.

Regarding my particular questions, I find that all the replies are indeed satisfactory. The overall paper I think improved largely, especially in clarifying the modelling approaches and the logic followed. The organisation of the GitHub is now very clear and I do believe that this will make it easy to adapt these analyses to other cohorts as well. I praise the authors for following this suggestion.

I have no other reservations and indeed suggest to accept the current version of this paper.

Giulio Caravagna

We thank Reviewer #1.

Reviewer #2 (Remarks to the Author):

The authors have convincingly showed that on the level of SCNAs the genomes are stable between primary and relapses for the two cases in their cohort where they had paired samples (within the limitations inherent in that comparison - bottlenecks, selection etc). They are also quite correct in that when samples differ can be hard to really know whether (especially for whole chromosome gains) the SCNA is really subclonal or whether it was clonal and then lost in a subclone. They furthermore validate this by comparing the sSNV density for each segment within a tumor to the sSNV density for the MRCA, and reassuringly found densities compatible with subclonal SCNAs within a small fraction of samples. In total, I am now convinced that the

vast majority of SCNAs reported by Körber et al are indeed clonal and any residual subclonal SCNAs should have at maximum minimal effect on the conclusions.

Körber et al have also clarified the relationship between the sample material used for WGS and the material used for measuring ploidy in a satisfying manner, reassuring this reviewer that they are indeed from closely located samples. In my mind this also adds another layer of security in the classification of clonal vs subclonal SCNAs (by constraining the parameter space for (purity, ploidy) solutions when calling absolute copy numbers).

Regarding my question on mutational signatures the authors have repeated the analysis using only clock-like signatures and this encouragingly shows concordant results with using all somatic SNVs. They are also quite right that dichotomising a bimodal distribution typically doesn't lead to a dramatic loss of information and I can also appreciate that a binary classification would facilitate more prompt implementation in clinical trials, so I am now satisfied with the use of the dichotomised variable in their survival analysis.

While Supplementary Table 2 contains all mutational data on the discovery cohort I was unable to find the mutational data for the validation cohort as well as the copy number data from ACEseq (for both discovery and validation). these should be part of the supplemental data as somatic mutations and SCNAs doesn't identify the patient while their availability increases the utility of the article also as a data resource. The revised article was a delight to read and I'm convinced that the findings will be important both to the neuroblastoma field but also more broadly to the tumor evolution field.

The authors have now answered all of my questions in a satisfying manner and when the datasets mentioned above are added to the supplement my recommendation is acceptance of the article.

We thank Reviewer #2. We have added all the requested data; they are now in Supplementary Tables 6-10.

Reviewer #3 (Remarks to the Author):

From my perspective all reviewer comments have been appropriately addressed and I recommend the paper for publication.

We thank Reviewer #3.

Final Decision Letter:

6th Feb 2023

Dear Thomas,

I am delighted to say that your manuscript "Neuroblastoma arises in early fetal development and its evolutionary duration predicts outcome" has been accepted for publication in an upcoming issue of Nature Genetics.

Your paper will be published online after we receive your corrections and will appear in print in the next available issue. You can find out your date of online publication by contacting the Nature Press Office (press@nature.com) after sending your e-proof corrections. Now is the time to inform your Public Relations or Press Office about your paper, as they might be interested in promoting its publication. This will allow them time to prepare an accurate and satisfactory press release. Include your manuscript tracking number (NG-A60512R1) and the name of the journal, which they will need when they contact our Press Office.

Please note that *Nature Genetics* is a Transformative Journal (TJ). Authors may publish their research with us through the traditional subscription access route or make their paper immediately open access through payment of an article-processing charge (APC). Authors will not be required to make a final decision about access to their article until it has been accepted. [Find out more about Transformative Journals](https://www.springernature.com/gp/open-research/transformative-journals)

Authors may need to take specific actions to achieve [compliance with funder and institutional open access mandates](https://www.springernature.com/gp/open-research/funding/policy-compliance-faqs). If your research is supported by a funder that requires immediate open access (e.g. according to [Plan S principles](https://www.springernature.com/gp/open-research/plan-s-compliance)) then you should select the gold OA route, and we will direct you to the compliant route where possible. For authors selecting the subscription publication route, the journal's standard licensing terms will need to be accepted, including [self-archiving and license to publish](https://www.nature.com/nature-portfolio/editorial-policies/self-archiving-and-license-to-publish). Those licensing terms will supersede any other terms that the author or any third party may assert apply to any version of the manuscript.

Please note that Nature Portfolio offers an immediate open access option only for papers that were first submitted after 1 January, 2021.

If you have not already done so, we invite you to upload the step-by-step protocols used in this manuscript to the Protocols Exchange, part of our on-line web resource, natureprotocols.com. If you

complete the upload by the time you receive your manuscript proofs, we can insert links in your article that lead directly to the protocol details. Your protocol will be made freely available upon publication of your paper. By participating in natureprotocols.com, you are enabling researchers to more readily reproduce or adapt the methodology you use. Natureprotocols.com is fully searchable, providing your protocols and paper with increased utility and visibility. Please submit your protocol to <https://protocolexchange.researchsquare.com/>. After entering your nature.com username and password you will need to enter your manuscript number (NG-A60512R1). Further information can be found at <https://www.nature.com/nature-portfolio/editorial-policies/reporting-standards#protocols>

Sincerely,

Michael Fletcher, PhD
Senior Editor, Nature Genetics

ORCID: 0000-0003-1589-7087